# Wasserstein convergence of Čech persistence diagrams for samplings of submanifolds

**Charles Arnal**[*]
Université Paris-Saclay, Inria

**David Cohen-Steiner**[*]
Université Côte d'Azur, Inria

**Vincent Divol**[*]
CREST, ENSAE

## Abstract

Čech Persistence diagrams (PDs) are topological descriptors routinely used to capture the geometry of complex datasets. They are commonly compared using the Wasserstein distances $\mathrm{OT}_p$; however, the extent to which PDs are stable with respect to these metrics remains poorly understood. We partially close this gap by focusing on the case where datasets are sampled on an $m$-dimensional submanifold of $\mathbb{R}^d$. Under this manifold hypothesis, we show that convergence with respect to the $\mathrm{OT}_p$ metric happens exactly when $p > m$. We also provide improvements upon the bottleneck stability theorem in this case and prove new laws of large numbers for the total $\alpha$-persistence of PDs. Finally, we show how these theoretical findings shed new light on the behavior of the feature maps on the space of PDs that are used in ML-oriented applications of Topological Data Analysis.

## 1  Introduction

Topological Data Analysis (TDA) is a set of tools that aims at extracting relevant topological information from complex datasets, e.g. regarding connected components, loops, cavities, or higher dimensional features. These different notions are made formal through the use of *homology theory*, and in particular the *i-th homology group* $H_i(\mathsf{A})$ of a set $\mathsf{A}$, which captures the $i$-dimensional topological features of $\mathsf{A}$ for $i \geq 0$, see e.g. [46] or Appendix B. TDA has been successfully applied in a variety of domains, including material science [51, 63, 30, 17], biology [21, 68, 8], real algebraic geometry [29, 35, 28, 44] and neuroscience [67, 61, 18], to name a few. When used in conjunction with more traditional approaches such as neural networks, TDA-based methods have outperformed state of the arts methods for tasks such as graph classifications [19, 50].

The most prominent techniques in TDA rely on multiscale approaches, in particular through the use of *persistent homology* [22]. Given a compact set $\mathsf{A}$ in $\mathbb{R}^d$, persistent homology tracks the evolution of the homology groups $H_i(\mathsf{A}^t)$ of the $t$-offset $\mathsf{A}^t = \bigcup_{x \in \mathsf{A}} \overline{B}(x, t)$ of $\mathsf{A}$ as $t$ goes from $0$ to $+\infty$ (where $\overline{B}(x, t)$ is the closed ball of radius $t$ centered at $x$). The process is summarized by the *Čech persistence diagram (PD) of degree* $i$ of the set $\mathsf{A}$: the PD $\mathrm{dgm}_i(\mathsf{A})$ is a multiset[2] of points in the half-plane $\Omega := \{(u_1, u_2) \in \mathbb{R}^2 : u_1 < u_2\}$, where each point $(u_1, u_2)$ in the PD corresponds to a $i$-dimensional topological feature that appeared in $\mathsf{A}^t$ at scale $t = u_1$ (its birth time) and disappeared at scale $t = u_2$ (its death time), see Figure 1.[3]  Points close to the

---

[*]Equal contribution

[2]A multiset is a set where each element appears with some non-zero multiplicity.

[3]In general, PDs can have points with infinite coordinates. For Čech PDs, this will only be the case for a single point of the diagram for $i = 0$, of coordinate $(0, +\infty)$. We discard this point in the following.

38th Conference on Neural Information Processing Systems (NeurIPS 2024).

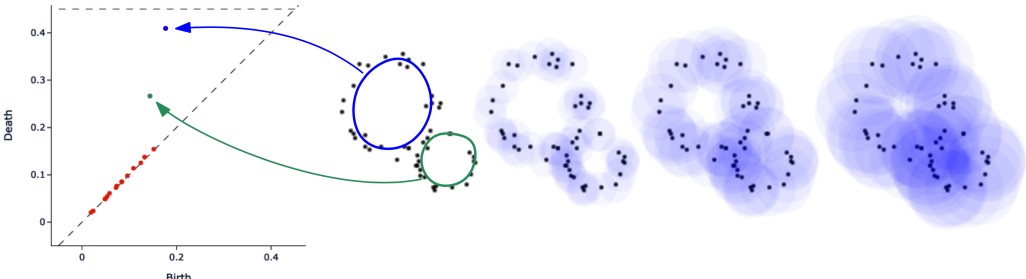

Figure 1: The Čech PD of a point cloud A in $\mathbb{R}^2$ for $i = 1$ and its $t$-offsets. The two points far from the diagonal $\partial\Omega$ in $\mathrm{dgm}_i(\mathsf{A})$ correspond to the two large cycles in the set A.

diagonal $\partial\Omega = \{(u_1, u_2) \in \mathbb{R}^2 : u_1 = u_2\}$ correspond to topological features of small *persistence* $\mathrm{pers}(u) = \frac{u_2 - u_1}{2}$, which have a short lifetime in the filtration $(\mathsf{A}^t)_{t \geq 0}$.

A key property of PDs is their robustness to small perturbation in the data, making them suitable for analyzing real-world datasets. This stability property is phrased in terms of the *bottleneck distance* between PDs [36]. Let $a$ and $b$ be two PDs. A *partial matching* between $a$ and $b$ is a bijection of multisets $\gamma : a \cup \partial\Omega \to b \cup \partial\Omega$, where each point $(u, u)$ of $\partial\Omega$ has infinite multiplicity. In other words, the points of $a$ are either paired with a single point of $b$, or mapped to the diagonal $\partial\Omega$ (and similarly for the points of $b$). Let $\Gamma(a, b)$ be the set of all partial matchings between $a$ and $b$. The bottleneck distance is defined as

$$\mathrm{OT}_\infty(a, b) = \inf_{\gamma \in \Gamma(a,b)} \max_{u \in a \cup \partial\Omega} \|u - \gamma(u)\|_\infty. \tag{1}$$

The Bottleneck Stability Theorem [26, 22] states that if $\mathsf{A}_1$ and $\mathsf{A}_2$ are two compact sets, then for any integer $i \geq 0$

$$\mathrm{OT}_\infty(\mathrm{dgm}_i(\mathsf{A}_1), \mathrm{dgm}_i(\mathsf{A}_2)) \leq \varepsilon, \tag{2}$$

where $\varepsilon$ is the Hausdorff distance between the sets $\mathsf{A}_1$ and $\mathsf{A}_2$, defined by $d_H(\mathsf{A}_1, \mathsf{A}_2) = \sup_{x \in \mathbb{R}^d} |d_{\mathsf{A}_1}(x) - d_{\mathsf{A}_2}(x)|$ and where $d_\mathsf{A}$ is the distance function to a set A. An important property of the bottleneck distance is that it is blind to small-persistence topological features: if $\mathrm{OT}_\infty(a, b) = \varepsilon$, then one can arbitrarily modify the PDs $a$ and $b$ on a slab of width $\varepsilon$ above the diagonal (for the $\ell_\infty$-metric) without changing the bottleneck distance between the two PDs.

Due to this phenomenon, the bottleneck distance turns out to be too weak in many situations of interest, where some topological features of small persistence can be as important as large-scale topological features in the PD (say, with a classification or a regression task in mind). For this reason, finer transport-like distances are often preferred to compare PDs. These distances, which we denote as $\mathrm{OT}_p$, are defined for $1 \leq p < \infty$ by

$$\mathrm{OT}_p(a, b) = \inf_{\gamma \in \Gamma(a,b)} \left( \sum_{u \in a \cup \partial\Omega} \|u - \gamma(u)\|_\infty^p \right)^{1/p}, \tag{3}$$

with $\mathrm{OT}_p \leq \mathrm{OT}_{p'}$ for $1 \leq p' \leq p < \infty$. They can be seen as modified versions of the Wasserstein distances used in optimal transport, with the diagonal $\partial\Omega$ playing the role of a landfill of infinite mass, see e.g. [33].

The increased sensitivity to small perturbations of the $\mathrm{OT}_p$ distances is of crucial importance in the standard TDA pipeline, which we briefly recall. Starting from a sample of sets $\mathsf{A}_1, \ldots, \mathsf{A}_n$, one computes a sample of PDs $a_j = \mathrm{dgm}_i(\mathsf{A}_j)$, $j = 1, \ldots, n$. Statistical methods to analyze this sample of PDs are typically awkward to define, due to the nonlinear geometry of the space of PDs [16, 75]. To overcome this issue, the space of PDs is mapped to a vector space through some map $\Phi$ called a feature map. Statistical method are then applied in the feature space on the transformed sample $\Phi(a_1), \ldots, \Phi(a_n)$. Various feature maps have been designed [23, 60, 25, 54, 20, 70, 53, 48], important examples including persistent images [4], PersLay [19], and PLLay [50]; in the latter two, the feature map is parametrized by a neural network. A good feature map should preserve as much as possible the geometry of the space of PDs [57]; in particular, Lipschitz (or Hölder) continuity of the feature map is a basic requirement. However, due to their (often desirable) sensitivity to small scale

features, most common feature maps are **not** regular with respect to the $\mathrm{OT}_\infty$ distance on the space of diagrams. Instead, they enjoy Lipschitz regularity with respect to either the finer $\mathrm{OT}_1$ distance (see e.g. [33, Proposition 5.2]), or to the $\mathrm{OT}_p$ distances (for $p > 1$) when restricted to diagrams $a$ whose $\alpha$-*total persistence*

$$\mathrm{Pers}_\alpha(a) = \sum_{u \in a} \mathrm{pers}(u)^\alpha \tag{4}$$

is bounded for some $\alpha > 0$ large enough, see [34] and [53]. Boundedness assumptions on the $\alpha$-total persistence also yield a version of the Bottleneck Stability Theorem with respect to the finer $\mathrm{OT}_p$ distance: if $\mathsf{A}_1$ and $\mathsf{A}_2$ are such that $\mathrm{Pers}_\alpha(\mathrm{dgm}_i(\mathsf{A}_k)) \leq M$ ($k = 1, 2$), then, for $p \geq \alpha$,

$$\mathrm{OT}_p^p(\mathrm{dgm}_i(\mathsf{A}_1), \mathrm{dgm}_i(\mathsf{A}_2)) \leq M\varepsilon^{p-\alpha}, \tag{5}$$

where $\varepsilon = d_H(\mathsf{A}_1, \mathsf{A}_2)$, see [27]. Hence, in addition to having intrinsic theoretical interest, controlling the total persistence and convergence with respect to the $\mathrm{OT}_p$ distance of PDs is crucial to ensure the soundness of most methods commonly used in TDA. This is the subject of this article.

**Contributions.** We provide a deeper understanding of the structure of Čech PDs in the specific case where the underlying set $\mathsf{A}$ is a compact subset of am $m$-dimensional manifold $\mathsf{M}$ in $\mathbb{R}^d$, focusing in particular on the total persistence of the PDs and their convergence to the PDs of $\mathsf{M}$ with respect to the $\mathrm{OT}_p$ distances. The importance of this case is supported by the *manifold hypothesis*, which often serves as a fundamental principle guiding the development of algorithms and models for data analysis [41, 76, 15]. Specifically, our main contributions are the following:

- **Theorem 2.2:** When $\mathsf{A} \subset \mathsf{M}$ is a compact set satisfying $d_H(\mathsf{A}, \mathsf{M}) \leq \varepsilon$ for $\varepsilon$ small enough, we provide a quadratic improvement upon the standard Bottleneck Stability Theorem (2). Namely, we show that there exists an optimal bottleneck matching $\gamma$ such that the distance between a coordinate of a point $u \in \mathrm{dgm}_i(\mathsf{A})$ and the coordinate of the matched point $\gamma(u) \in \mathrm{dgm}_i(\mathsf{M}) \cup \partial\Omega$ is of order $O(\varepsilon^2)$ whenever the coordinate of $u$ is larger than $2\varepsilon$.

- **Theorem 3.3:** In the case where the manifold $\mathsf{M}$ is generic and the set $\mathsf{A}$ is a $\delta$-sparse point cloud (i.e. $\min_{x \neq y \in \mathsf{A}} \|x - y\| \geq \delta$), we provide a finer analysis by showing that the $p$-total persistence $\mathrm{Pers}_p(\mathrm{dgm}_i(\mathsf{A}))$ remains bounded and the distance $\mathrm{OT}_p(\mathrm{dgm}_i(\mathsf{A}), \mathrm{dgm}_i(\mathsf{M}))$ converges to 0 for all $p > m$ whenever the ratio $\varepsilon/\delta$ is upper bounded.

- **Corollary 4.3:** We then focus on a random context, by assuming that $\mathsf{A} = \mathsf{A}_n$ is obtained by sampling $n$ i.i.d. random variables with positive bounded density $f$ on a generic manifold $\mathsf{M}$. We prove that $\mathrm{OT}_p(\mathrm{dgm}_i(\mathsf{A}_n), \mathrm{dgm}_i(\mathsf{M}))$ converges in expectation to 0 for $p > m$. Furthermore, we obtain a law of large numbers for the $\alpha$-total persistence of $\mathrm{dgm}_i(\mathsf{A}_n)$:

$$\mathrm{Pers}_\alpha(\mathrm{dgm}_i(\mathsf{A}_n)) = \mathrm{Pers}_\alpha(\mathrm{dgm}_i(\mathsf{M})) + C_i n^{1-\alpha/m} + o_{L^1}(n^{1-\alpha/m}) + O_{L^1}\left(\left(\frac{\log n}{n}\right)^{\frac{1}{m}}\right) \tag{6}$$

for all $\alpha > 0$, where $C_i$ is a constant that depends explicitly on $\mathsf{M}$ and $f$. In particular, for $0 \leq i < m$, $\mathrm{Pers}_\alpha(\mathrm{dgm}_i(\mathsf{A}_n))$ stays bounded if and only if $\alpha \geq m$.

Our contributions are to be compared to one of the only preexisting results regarding the $\alpha$-total persistence of a PD: in [27], the authors proved that for all $\alpha$ strictly greater than the ambient dimension $d$, the $\alpha$-total persistence of the Čech PD of a compact set $\mathsf{A} \subset B(0, R) \subset \mathbb{R}^d$ satisfies

$$\mathrm{Pers}_\alpha(\mathrm{dgm}_i(\mathsf{A})) \leq C_{\alpha,d} R^\alpha \tag{7}$$

for some constant $C_{\alpha,d}$ depending on $\alpha$ and $d$. In the two scenarios we considered (either $\delta$-sparse or random samples), the ambient dimension $d$ in the constraint $\alpha > d$ for the control of the $\alpha$-total persistence in (7) has been replaced by the smaller intrinsic dimension $m$ of the problem: the manifold hypothesis has been successfully exploited.

To summarize, our work sheds light on the behaviour of PDs, provides new guarantees for commonly used ML methods (see e.g. Corollary 4.4), and suggests new heuristics (see Section 5). We also perform various experiments to illustrate the validity of our results and their relevance to the classic TDA pipeline. All proofs are deferred to the Appendix.

**Related work.** This work is part of a long ongoing effort to understand simplicial complexes and PDs in a random context [10, 12, 13, 11, 49, 32, 59, 14, 45]. Closest to our work, Hiraoka, Shirai

and Trinh gave limit laws for Čech PDs for random points in the cube $[0,1]^d$ [47], while Goel, Trinh and Tsunoda gave similar asymptotics in the case of samples on manifolds [42]. Limit laws for the total persistence have been obtained by Divol and Polonik in the case of random samples in the cube [34]. Among other contributions, this work generalizes this result to submanifolds: unlike the cube, a manifold has a nontrivial topology; a fact which considerably complicates the situation, for we have to take into account the presence of large topological features in the Čech PDs in order to control the total persistence. Note also that tools other than persistent homology exist for studying the geometry of point clouds. For instance, the authors in [77, 52] consider complete isometric invariants for point clouds that are computable in polynomial time.

## 2 Čech persistence diagrams for subsets of submanifolds

Recall that a fundamental result in TDA, the Bottleneck Stability Theorem (2), states that Čech PDs are stable with respect to Hausdorff perturbations. Consider the particular setting where one has access to a set $A$, obtained as an approximation of an unknown shape of interest $S$ through some sampling procedure, with $A \subset S$ and $\sup_{x \in S} d_A(x) \leq \varepsilon$. The Bottleneck Stability Theorem ensures that $\mathrm{OT}_\infty(\mathrm{dgm}_i(A), \mathrm{dgm}_i(S)) \leq \varepsilon$ for any $i \geq 0$, a bound which cannot be improved in general. However, it turns out that a finer understanding of the proximity between $\mathrm{dgm}_i(A)$ and $\mathrm{dgm}_i(S)$ can be obtained if more regularity is assumed on the shape of interest $S$, namely in the situation where $S = M$ is a compact submanifold with positive reach.

Let us first set some notation. Let $M$ be an $m$-dimensional compact topological submanifold of $\mathbb{R}^d$; we always assume that the boundary of $M$ is empty. The orthogonal projection $\pi_M$ on $M$ is defined for $x$ close enough to $M$, and we define the *reach* $\tau(M)$ as the largest $r > 0$ such that the orthogonal projection $\pi_M$ is well (i.e. uniquely) defined for all $x \in \mathbb{R}^d$ at distance strictly less than $r$ from $M$. The reach is a key notion to quantify the regularity of a manifold, see e.g. [39] and [24] for more information.

Let $A \subset M$ be such that $d_H(A, M) \leq \varepsilon$ and let $z \in \mathbb{R}^d$. By definition of the Hausdorff distance, it holds that $|d_A(z) - d_M(z)| \leq \varepsilon$. However, this naive bound can be quadratically improved as long as $z$ stays far away from $M$.

**Lemma 2.1.** *Let $M \subset \mathbb{R}^d$ be a compact submanifold with positive reach and let $A \subset M$ be a compact set with $d_H(A, M) \leq \varepsilon$ for some $\varepsilon > 0$. Let $z \in \mathbb{R}^d \backslash M$. Then, $|d_M(z) - d_A(z)| \leq \frac{\varepsilon^2}{2d_M(z)} \left(1 + \frac{d_M(z)}{\tau(M)}\right)$.*

We can build upon this basic remark to obtain a very precise control of the behavior of the Čech PD of the set $A$. Namely, we identify three regions in the upper halfplane $\Omega$ (displayed in Figure 2) which contain all points in the PD $\mathrm{dgm}_i(A)$ (for some integer $i \geq 0$). In the first region, corresponding to microscopic topological features disappearing at scales smaller than $\varepsilon + \varepsilon^2/\tau(M)$, the Bottleneck Stability Theorem cannot be improved. However, there exists an optimal matching (i.e. a matching $\gamma : \mathrm{dgm}_i(A) \cup \partial\Omega \to \mathrm{dgm}_i(M) \cup \partial\Omega$ that realizes the bottleneck distance (1)) such that at least one of the coordinates of any point in the other two regions is larger than $\tau(M) - \varepsilon^2/\tau(M)$, and the proximity between a large coordinate of a point $u \in \mathrm{dgm}_i(A)$ and the coordinate of the matched point $\gamma(u) \in \mathrm{dgm}_i(M) \cup \partial\Omega$ is of order $O(\varepsilon^2)$. This yields a quadratic improvement upon the Bottleneck Stability Theorem.

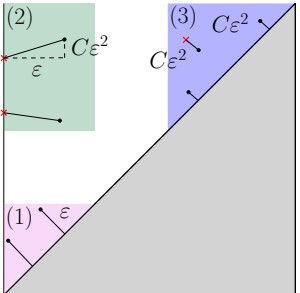

Figure 2: PDs of $M$ (red) and of $A$ (black).

**Theorem 2.2** (Improved Bottleneck Stability Theorem). *Let $M \subset \mathbb{R}^d$ be a compact submanifold with positive reach and let $A \subset M$ be a compact set such that $d_H(A, M) \leq \varepsilon < \tau(M)/4$. Let $i \geq 0$ be an integer. Then $\mathrm{dgm}_i(A)$ is the union of three regions $\mathrm{dgm}_i^{(1)}(A) := \mathrm{dgm}_i(A) \cap \{u_1, u_2 \leq \varepsilon + \frac{\varepsilon^2}{\tau(M)}\}$, $\mathrm{dgm}_i^{(2)}(A) := \mathrm{dgm}_i(A) \cap \{u_1 \leq \varepsilon, u_2 \geq \tau(M) - \frac{\varepsilon^2}{\tau(M)}\}$ and $\mathrm{dgm}_i^{(3)}(A) := \mathrm{dgm}_i(A) \cap \{u_1, u_2 \geq \tau(M) - \frac{\varepsilon^2}{\tau(M)}\}$.*

*Furthermore, let $C = \frac{2}{\tau(\mathsf{M})}\left(1 + \frac{R(\mathsf{M})}{\tau(\mathsf{M})}\right)$, where $R(\mathsf{M})$ is the radius of the smallest ball that contains* $\mathsf{M}$. *There exists an optimal matching* $\gamma : \mathrm{dgm}_i(\mathsf{A}) \cup \partial\Omega \to \mathrm{dgm}_i(\mathsf{M}) \cup \partial\Omega$ *for the bottleneck distance between* $\mathrm{dgm}_i(\mathsf{A})$ *and* $\mathrm{dgm}_i(\mathsf{M})$ *such that*

- *Region (1): If $u \in \mathrm{dgm}_i^{(1)}(\mathsf{A})$, then $\gamma(u) \in \partial\Omega$ and $\|u - \gamma(u)\|_\infty \leq \varepsilon$.*

- *Region (2): If $u \in \mathrm{dgm}_i^{(2)}(\mathsf{A})$, then $\gamma(u)$ is of the form $(0, v_2)$ and $|u_2 - v_2| \leq C\varepsilon^2$. The number of such points is finite and depends only on* $\mathsf{M}$.

- *Region (3): If $u \in \mathrm{dgm}_i^{(3)}(\mathsf{A})$, then $\|u - \gamma(u)\|_\infty \leq C\varepsilon^2$.*

Note that for any $i \geq d$, the $i$-th PDs of $\mathsf{M}$ and $\mathsf{A}$ are actually trivial.

## 3  Čech persistence diagrams for subsets of generic submanifolds

The improved Bottleneck Stability Theorem (Theorem 2.2) yields information relative to the location of points in the Čech PD of $\mathsf{A}$, but not about their numbers. However, both the $\alpha$-total persistence of $\mathrm{dgm}_i(\mathsf{A})$ and the distance $\mathrm{OT}_p(\mathrm{dgm}_i(\mathsf{A}), \mathrm{dgm}_i(\mathsf{M}))$ for $p < \infty$ crucially depend on the number of points in $\mathrm{dgm}_i(\mathsf{A})$ having small persistence.

Unfortunately, no control on, say, the total persistence, can exist without additional assumptions. Indeed, in general, even the $\alpha$-total persistence of the Čech PD of the submanifold $\mathsf{M}$ can be infinite.

**Example 3.1.** *Let $f : x \in \mathbb{R} \mapsto 1 + x^4 \sin(1/x)^2$. Consider the $C^2$ curve $\mathsf{M}$ in $\mathbb{R}^2$ defined as the union of the graphs of the functions $f$ and $-f$ on $[-2, 2]$. Being $C^2$, the curve has a positive reach [39]. For $i = 1$, the Čech PD of $\mathsf{M}$ contains a sequence of points $(1, \ell_n)$ for $n \geq 1$, where $\ell_n = 1 + \Theta(n^{-4})$. In particular, as $\sum_{n \geq 1}(\ell_n - 1)^\alpha = +\infty$ for $\alpha < 1/4$, the $\alpha$-total persistence of $\mathrm{dgm}_1(\mathsf{M})$ is infinite for such a value of $\alpha$. By considering the product $\mathsf{M}^m \subset \mathbb{R}^{2m}$, one can also build an $m$-dimensional $C^2$ compact submanifold without boundary such that $\mathrm{dgm}_1(\mathsf{M})$ has an infinite $\alpha$-total persistence for $\alpha < m/4$.*

The existence of such counterexamples is explained by the fact that the distance function $d_{\mathsf{M}}$ to a set $\mathsf{M}$ is not well-behaved in general, even when the set $\mathsf{M}$ is smooth. In contrast to this bleak general case, Song, Yim & Monod (in the case of surfaces in $\mathbb{R}^3$) and Arnal, Cohen-Steiner & Divol (in the general case) studied the distance function $d_{\mathsf{M}}$ to $\mathsf{M}$ when $\mathsf{M}$ is a *generic* submanifold [6, 71]. Their findings indicate that, although counterexamples such as the one presented in Example 3.1 exist, they are extremely uncommon in a sense which can be made precise.

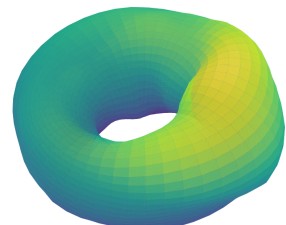

Figure 3: A generic torus.

Namely, given an *abstract* manifold $M$, Arnal, Cohen-Steiner and Divol show that the set of $C^2$ embeddings $\mathsf{M}$ of $M$ into $\mathbb{R}^d$ such that $d_{\mathsf{M}}$ satisfies some desirable regularity conditions (described in Appendix C) forms an open and dense set in the set of all $C^2$ embeddings equipped with the $C^2$-Whitney topology [6, Theorem 1.1]. In what follows, we will simply refer to a $C^2$ compact submanifold $\mathsf{M}$ such that $d_{\mathsf{M}}$ satisfies the regularity condition described in Appendix C as *generic*. See Figure 3 for an example.

**Proposition 3.2** (Čech PDs of generic submanifolds). *Let $\mathsf{M}$ be a generic submanifold of $\mathbb{R}^d$. Then, for any integer $i \geq 0$, the PD $\mathrm{dgm}_i(\mathsf{M})$ contains finitely many points. In particular, $\mathrm{Pers}_\alpha(\mathrm{dgm}_i(\mathsf{M})) < +\infty$ for all $\alpha > 0$.*

When $\mathsf{M}$ is a generic submanifold, it becomes a reasonable task to control the number of points in $\mathrm{dgm}_i(\mathsf{A})$ where $\mathsf{A} \subset \mathsf{M}$ is an approximation of $\mathsf{M}$ with $d_H(\mathsf{A}, \mathsf{M}) \leq \varepsilon$. The Bottleneck Stability Theorem implies that when $\varepsilon$ is small enough (compared to the smallest persistence of a point in $\mathrm{dgm}_i(\mathsf{M})$), every point of $\mathrm{dgm}_i(\mathsf{M})$ is mapped to a point in $\mathrm{dgm}_i(\mathsf{A})$ by the optimal bottleneck matching, leaving the points of $\mathrm{dgm}_i(\mathsf{A})$ at $\ell_\infty$-distance to $\partial\Omega$ less than $\varepsilon$ unmatched; those will be mapped to the diagonal $\partial\Omega$. The Improved Bottleneck Stability Theorem 2.2 (see Figure 2) shows that these points are of two kinds: those in Region (1), corresponding to small topological features in the set $\mathsf{A}$ (of size of order $O(\varepsilon)$), and those in Region (3), corresponding to large topological features. There are many points in Region (1) (in fact, our proofs show that when $\mathsf{A} = \mathsf{A}_n$ is a random

sample of $n$ points, the number of points in Region (1) is of order $O(n)$). In contrast, the genericity hypothesis allows us to show that the number of points in Region (3) is small under reasonable sampling assumptions.

We say that a point cloud $A \subset M$ is $(\delta, \varepsilon)$-*dense* in $M$ if $d_H(A, M) \leq \varepsilon$ and $\min_{x \neq y \in A} \|x - y\| \geq \delta$. Such point clouds naturally occur, e.g. as products of the farthest point sampling algorithm [2].

**Theorem 3.3.** *Let $M \subset \mathbb{R}^d$ be a generic compact submanifold and $A \subset M$ be a $(\delta, \varepsilon)$-dense set in $M$ for some $\varepsilon, \delta > 0$. Let $a \geq \varepsilon/\delta$ and let $i \geq 0$ be an integer. There exist $\varepsilon_0 > 0$ depending only on $M$ and $C_0, C_1, C_2, C_3$ depending only on $M$ and $a$ such that if $\varepsilon \leq \varepsilon_0$, then $\mathrm{dgm}_i^{(3)}(A)$ has at most $C_0$ points and for all $p \geq 1$, $\alpha \geq 0$,*

$$\begin{aligned} \mathrm{OT}_p^p(\mathrm{dgm}_i(A), \mathrm{dgm}_i(M)) &\leq C_1 \varepsilon^{p-m} \\ \mathrm{Pers}_\alpha(\mathrm{dgm}_i(A)) &\leq C_2(C_3^\alpha + \varepsilon^{\alpha-m}). \end{aligned} \tag{8}$$

In particular, as long as the ratio $\varepsilon/\delta$ is larger than some constant $a > 0$, the $\mathrm{OT}_p$ distance between $\mathrm{dgm}_i(A)$ and $\mathrm{dgm}_i(M)$ converges to 0 for all $p > m$ as $\varepsilon \to 0$, while the $p$-total persistence $\mathrm{Pers}_p(\mathrm{dgm}_i(A))$ stays bounded.

**Example 3.4.** *Consider two parallel line segments $M$ in $\mathbb{R}^2$, and a finite set $A$ consisting of two parallel grids of step $2\varepsilon$: the set $A$ is $(2\varepsilon, \varepsilon)$-dense in $M$. Then, there are $O(\varepsilon^{-1})$ points in $\mathrm{dgm}_1(A)$ with birth coordinates $u_1$ equal to $1/2$ and persistence of order $O(\varepsilon^2)$; they all belong to $\mathrm{dgm}_1^{(3)}(A)$, whose cardinality is thus not bounded by some $C_0 = C_0(M)$. This example can be easily modified to make $M$ a compact $C^2$ 1-dimensional manifold. This shows that the first conclusion of Theorem 3.3 cannot hold without a genericity assumption on $M$.*

# 4 Random samplings of submanifolds

We now turn to the case of random samplings of (non-generic and generic) submanifolds. They tend to adopt configurations that are more regular than what can be expected from e.g. a general $\varepsilon$-dense sampling, yet their randomness gives rise to new technical difficulties. Let $P$ be a probability measure having a density $f$ (with respect to the volume measure) on a compact submanifold $M$ of dimension $m \geq 1$. Let $A = A_n = \{X_1, \ldots, X_n\}$, where $X_1, \ldots, X_n$ is an i.i.d. sample from distribution $P$. Let $i \geq 0$ be an integer; we consider the three regions described in Figure 2 and in the statement of Theorem 2.2, and write again $\mathrm{dgm}_i^{(1)}(A_n)$, $\mathrm{dgm}_i^{(2)}(A_n)$ and $\mathrm{dgm}_i^{(3)}(A_n)$ for the three corresponding PDs. This section is devoted to the study of the probabilistic asymptotic behaviour of these three random PDs, which can be decomposed into two almost independent questions: $\mathrm{dgm}_i^{(1)}(A_n)$ only depends on small-scale phenomena and can essentially be reduced to the case of a cube, even if $M$ is non-generic, while $\mathrm{dgm}_i^{(2)}(A_n)$ and $\mathrm{dgm}_i^{(3)}(A_n)$ are tightly connected to the macroscopic geometry of the submanifold and can be further controlled using genericity assumptions on $M$.

Describing the limit behavior of the random PD $\mathrm{dgm}_i^{(1)}(A_n)$ requires extending the metric $\mathrm{OT}_p$ between PDs to more general Radon measures. Indeed, a PD can equivalently be seen as an integer-valued discrete Radon measure on $\Omega$, by identifying a multiset $a$ with the Radon measure $\sum_{u \in a} \delta_u$. Let $\mathcal{M}$ denote the space of Radon measures on $\Omega$, that is the space of Borel measures on $\Omega$ which give finite mass to every compact set $K \subset \Omega$.[4] The space of Radon measures is endowed with the *vague topology*, where a sequence $(\mu_n)_n$ of measures in $\mathcal{M}$ is said to converge vaguely to $\mu \in \mathcal{M}$ if $\int_\Omega \phi \mathrm{d}\mu_n \to \int_\Omega \phi \mathrm{d}\mu$ as $n \to \infty$ for all continuous functions $\phi : \Omega \to \mathbb{R}$ with compact support.

The $\alpha$-total persistence is defined for $\mu \in \mathcal{M}$ by $\mathrm{Pers}_\alpha(\mu) = \int_\Omega \mathrm{pers}(u)^\alpha \mathrm{d}\mu(u)$. For $p \geq 1$, we let $\mathcal{M}^p = \{\mu \in \mathcal{M} : \mathrm{Pers}_p(\mu) < +\infty\}$. The distance $\mathrm{OT}_p$, defined between PDs in (3), can be extended to the set $\mathcal{M}^p$, see [33]. The distance $\mathrm{OT}_p$ between Radon measures is a variation of the Wasserstein distance, with the important difference that the standard Wasserstein distance is only defined for measures having the same mass, while the distance $\mathrm{OT}_p$ is defined for measures having different (and even infinite) masses. We refer to Appendix F for the precise definition and the main properties of the $\mathrm{OT}_p$ distance on the space $\mathcal{M}_p$.

---

[4] A compact set $K \subset \Omega$ is at *positive distance* from the diagonal. Hence, a measure $\mu \in \mathcal{M}$ can have an accumulation of mass close to $\partial\Omega$.

For $q > 0$, given a function $f : \mathbb{N} \to \mathbb{R}$ and a sequence of (nonnecessarily measurable) real maps $(Y_n)_n$ defined on some probabilistic space, the notation $Y_n = O_{L^q}(f(n))$ means that $\mathbb{E}^*[|Y_n|^q] = O(f(n)^q)$, where $\mathbb{E}^*$ denotes the outer expectation [73, p.6] (and similarly for the little $o$ notation).

## 4.1 Region (1)

Consider the rescaled Radon measure $\mu_{n,i} = \frac{1}{n} \sum_{u \in \mathrm{dgm}_i^{(1)}(\mathsf{A}_n)} \delta_{n^{1/m}u}$, and note that $\mu_{n,i}$ is a random measure, owing to the randomness of the set $\mathsf{A}_n$. Goel, Trinh and Tsunoda studied the vague convergence of the sequence $(\mu_{n,i})_n$ [42, Remark 4.2]. Namely, assuming that the density $f$ satisfies $\int_{\mathsf{M}} f^j < \infty$ for all $j \geq 0$, they show that with probability 1 the sequence $(\mu_{n,i})_n$ converges vaguely to some (non-random) Radon measure $\mu_{f,i}$. The limit measure $\mu_{f,i}$ has support $\{0\} \times \mathbb{R}_+$ if $i = 0$ and $\Omega$ if $0 < i < m$; it is the zero measure if $i \geq m$. We can further describe it as follows: let $\mu_{\infty,i,m}$ be the limit of the sequence $(\mu_{n,i})$ in the case where the sample $\mathsf{A}_n$ is uniform on the unit cube $[0,1]^m$ (see [34]). Then, for any continuous function $\phi : \Omega \to \mathbb{R}$ with compact support,

$$\int_\Omega \phi(u)\mathrm{d}\mu_{f,i}(u) = \int_\Omega \int_{\mathsf{M}} f(x)\phi(f(x)^{-1/m}u)\mathrm{d}x\mathrm{d}\mu_{\infty,i,m}(u). \tag{9}$$

Note that the vague convergence of Radon measures is only defined with respect to compactly supported functions; as such, it is blind to phenomena located increasingly close to the diagonal $\partial\Omega$ as $n$ goes to infinity. In particular, and except in the case of the uniform distribution on the unit cube $[0,1]^m$ (see [34]), it was not known whether $\mu_{n,i}$ converges to $\mu_{f,i}$ for the $\mathrm{OT}_p$ distance as well, nor whether the sequences of total persistence $(\mathrm{Pers}_\alpha(\mu_{n,i}))$ converge. We close this gap with the following result.

**Theorem 4.1** (Law of large numbers). *Assume that $P$ has a density $f$ on $\mathsf{M}$ bounded away from $0$ and $\infty$. Let $i \geq 0$ be an integer and let $1 \leq p < \infty$. Then $\mu_{f,i} \in \mathcal{M}_p$ and $\mathbb{E}[\mathrm{OT}_p^p(\mu_{n,i}, \mu_{f,i})] \xrightarrow[n\to\infty]{} 0$.*

*Furthermore, for all $\alpha > 0$, $\mathrm{Pers}_\alpha(\mathrm{dgm}_i^{(1)}(\mathsf{A}_n))n^{\frac{\alpha}{m}-1} = \mathrm{Pers}_\alpha(\mu_{n,i}) = \mathrm{Pers}_\alpha(\mu_{f,i}) + o_{L^1}(1)$.*

## 4.2 Regions (2)-(3)

It is a well-known fact that the Hausdorff distance $\varepsilon = d_H(\mathsf{A}_n, \mathsf{M})$ between a random sample and $\mathsf{M}$ is of order $(\log n/n)^{1/m}$ whenever the underlying density $f$ is bounded away from zero and $\infty$ on $\mathsf{M}$, see e.g. [38]. Hence the PDs $\mathrm{dgm}_i^{(2)}(\mathsf{A}_n)$ and $\mathrm{dgm}_i^{(3)}(\mathsf{A}_n)$ can be described using Theorem 2.2. However, in the case where $\mathsf{M}$ is a generic submanifold, one can actually obtain tighter results. We let $\#E$ denote the cardinality of a multiset $E$.

**Proposition 4.2.** *Let $\mathsf{M}$ be a generic $m$-dimensional submanifold. Assume that $P$ has a density $f$ on $\mathsf{M}$ bounded away from $0$ and $\infty$. Let $i \geq 0$ be an integer. There exists an optimal matching $\gamma_n : \mathrm{dgm}_i(\mathsf{A}_n) \cup \partial\Omega \to \mathrm{dgm}_i(\mathsf{M}) \cup \partial\Omega$ for the bottleneck distance between $\mathrm{dgm}_i(\mathsf{A}_n)$ and $\mathrm{dgm}_i(\mathsf{M})$ such that for any $q \geq 1$:*

- *Region (2): It holds that $\max_{u \in \mathrm{dgm}_i^{(2)}(\mathsf{A}_n)} |u_2 - \gamma_n(u)_2| = O_{L^q}(n^{-2/m})$.*

- *Region (3): It holds that $\max_{u \in \mathrm{dgm}_i^{(3)}(\mathsf{A}_n)} \|u - \gamma_n(u)\|_\infty = O_{L^q}(n^{-2/m})$ and $\#(\mathrm{dgm}_i^{(3)}(\mathsf{A}_n)) = O_{L^q}(1)$.*

We remark that the same bounds can be obtained almost surely (e.g. "a.s. there exists $C > 0$ such that $\max_{u \in \mathrm{dgm}_i^{(3)}(\mathsf{A}_n)} \|u - \gamma_n(u)\|_\infty \leq Cn^{-2/m}$"), rather than in expectation, using similar arguments.

Proposition 4.2 yields two distinct improvements upon direct applications of Theorem 2.2 and Theorem 3.3 to the random case. First, we obtain bounds of order $n^{-2/m}$ instead of bounds of order $(\log n/n)^{2/m}$. Second, the random sample $\mathsf{A}_n$ is in general only $\delta$-sparse for $\delta$ of order $n^{-2/m}$. Hence, $\mathsf{A}_n$ is $(\delta, \varepsilon)$-dense in $\mathsf{M}$, but with a diverging ratio $\varepsilon/\delta$. Therefore, Theorem 3.3 cannot be applied to control the total number of points in $\mathrm{dgm}_i(\mathsf{A}_n)$ in Region (3).

## 4.3 Consequences for the Wasserstein convergence of persistence diagrams

As a simple consequence of Theorem 4.1 and Proposition 4.2, we obtain that for $i < m$, the $p$-Wasserstein convergence of $(\mathrm{dgm}_i(\mathsf{A}_n))$ to $\mathrm{dgm}_i(\mathsf{M})$ holds if and only if $p > m$, as well as precise asymptotics for the total persistence of $(\mathrm{dgm}_i(\mathsf{A}_n))$.

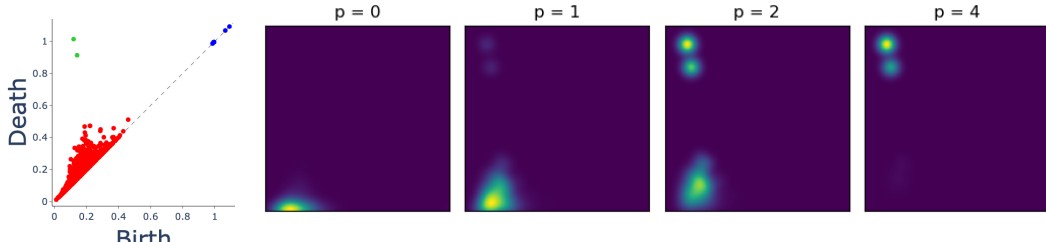

Figure 4: Left: the Čech PD $\mathrm{dgm}_1(\mathsf{A}_n)$ of a sample of $n = 10^4$ points sampled on a generic torus, with points in Regions (1), (2) and (3) highlighted in different colors. Right: the persistence images of $\mathrm{dgm}_1(\mathsf{A}_n)$ with weight $\mathrm{pers}^p$ for different values of $p$.

**Corollary 4.3.** *Let $p \geq 1$ and let $0 \leq i < d$ be an integer. Under the same assumptions as in Proposition 4.2, the following holds:*

- *If $p > m$, then $\mathbb{E}[\mathrm{OT}_p^p(\mathrm{dgm}_i(\mathsf{A}_n), \mathrm{dgm}_i(\mathsf{M}))] \to 0$ as $n \to \infty$.*

- *If $p = m$, $\mathbb{E}[\mathrm{OT}_p^p(\mathrm{dgm}_i(\mathsf{A}_n), \mathrm{dgm}_i(\mathsf{M}))] \to \mathrm{Pers}_p(\mu_{\infty,i,m})\mathrm{Vol}(\mathsf{M})$ as $n \to \infty$, where $\mathrm{Vol}(\mathsf{M})$ is the volume of $\mathsf{M}$.*

- *If $p < m$ and $i < m$, then $\mathbb{E}[\mathrm{OT}_p^p(\mathrm{dgm}_i(\mathsf{A}_n), \mathrm{dgm}_i(\mathsf{M}))] \to +\infty$ as $n \to \infty$.*

*Furthermore, for all $\alpha > 0$, $\mathrm{Pers}_\alpha(\mathrm{dgm}_i(\mathsf{A}_n))$ is equal to*

$$\mathrm{Pers}_\alpha(\mathrm{dgm}_i(\mathsf{M})) + n^{1-\frac{\alpha}{m}}\mathrm{Pers}_\alpha(\mu_{\infty,i,m})\int_{\mathsf{M}} f(x)^{1-\frac{\alpha}{m}}\mathrm{d}x + o_{L^1}(n^{1-\frac{\alpha}{m}}) + O_{L^1}\left(\left(\frac{\log n}{n}\right)^{\frac{1}{m}}\right).$$

As noted earlier, both $\mathrm{dgm}_i(\mathsf{A}_n)$ and $\mathrm{dgm}_i(\mathsf{M})$ are trivial if $i \geq d$.

This corollary gives a precise answer to the questions raised in the introduction. First, when $\mathsf{A}_n$ is a random subset of a $m$-dimensional generic manifold in $\mathbb{R}^d$, the $\alpha$-total persistence of $\mathrm{dgm}_i(\mathsf{A}_n)$ is not only bounded for $\alpha > d$ (as was shown by Cohen-Steiner & al. [27]), but for all $\alpha \geq m$. Moreover, the sequence $\mathrm{dgm}_i(\mathsf{A}_n)$ converges for the $\mathrm{OT}_p$ distance if $p > m$. A curious phenomenon can be observed in the case $p = m$: the sequence does not converge to $\mathrm{dgm}_i(\mathsf{M})$ as one would expect, but its distance to the power $p$ to $\mathrm{dgm}_i(\mathsf{M})$ converges to some constant–in that case, the cost to the power $p$ of matching all the points in Region (1) to the diagonal $\partial\Omega$ neither converges to 0 nor diverges, but is asymptotically equal to this constant.

Using these bounds on the total persistence, we obtain regularity guarantees for a large family of feature maps, called *linear feature maps*, which includes feature maps introduced in [23, 4, 60, 25, 54, 70, 19]. Let $(V, \|\cdot\|)$ be a normed vector space, and let $\phi : \Omega \to V$ be a continuous bounded map. For $\alpha \geq 0$, the linear feature map $\Phi_\alpha$ associated to $\phi$ and defined on the space $\mathcal{D}_f$ of PDs having a finite number of points is defined for all $a \in \mathcal{D}_f$ by $\Phi_\alpha(a) = \sum_{u \in a} \mathrm{pers}(u)^\alpha \phi(u) \in V$.

**Corollary 4.4.** *Let $\alpha \geq 1$ and let $0 \leq i < d$ be an integer. Under the same assumptions as in Proposition 4.2, it holds that $\Phi_\alpha(\mathrm{dgm}_i(\mathsf{A}_n))$ converges in probability to $\Phi_\alpha(\mathrm{dgm}_i(\mathsf{M}))$ whenever $\alpha > m$.*

Remark that other weighting schemes are possible. For instance, [53] argued for using linear feature maps of the form $\Phi_\alpha(a) = \sum_{u \in a} \arctan(\mathrm{pers}(u)^\alpha)\phi(u)$. Similar results would hold for such feature maps, as the map $u \mapsto \arctan(\mathrm{pers}(u)^\alpha)\phi(u)/\mathrm{pers}(u)^\alpha$ is continuous and bounded whenever $\phi$ is.

## 5 Numerical experiments

We illustrate our results with synthetic experiments, full details are given in Appendix H. We create a generic submanifold of dimension $m$ by applying a random diffeomorphism $\Psi$ to a given $m$-dimensional submanifold $\mathsf{M}_0$ (e.g. a torus). We then draw a sample of $n$ i.i.d. observations sampled according to the pushforward $P$ of the uniform distribution on $\mathsf{M}$ by $\Psi$.

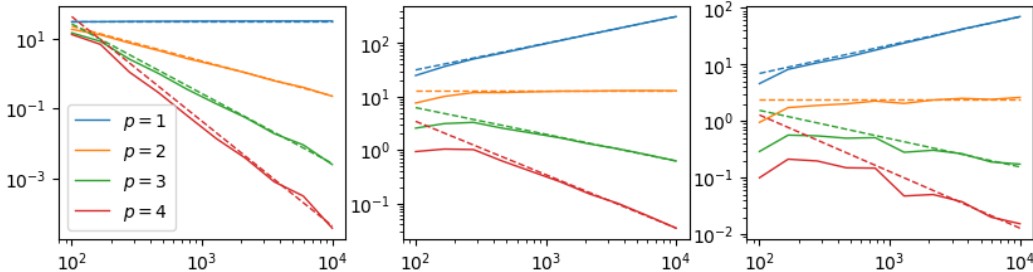

Figure 5: Plot in log-log scale of $\text{Pers}_p(\text{dgm}_i^{(1)}(\mathsf{A}_n))$ as a function of $n$ for points sampled on a circle, $i = 0$ (left), points sampled on a torus, $i = 0$ (center), points sampled on a torus, $i = 1$ (right). Dashed lines have slopes equal to $1 - p/m$.

**Continuity of feature maps.** As a first experiment, we test the continuity of a feature map, the persistence image [4]. In Figure 4, we plot the persistence image of $\text{dgm}_1(\mathsf{A}_n)$ where $\mathsf{A}_n$ is a sample of $n = 10^4$ points on a generic torus. We observe that the map is discontinuous for $p < 2$: the two points with large persistence corresponding to the PD of the underlying torus are nonapparent in the image. For $p > 2$, the two points are apparent, and the contribution of points with small persistence (close to the lower edge) has vanished. In the limit situation $p = 2$, we see the contribution of both points with large and small persistence. This phenomenon suggests the following heuristics: **when in presence of multiple datasets on $m$-dimensional objects whose global geometries need to be distinguished, feature maps with weights $\text{pers}^p$ with $p > m$ should be used; when the relevant information is the underlying density of the datasets, the choice $p < m$ should be preferred**.

**Convergence of total persistences.** We verify the rate of convergence of the total persistence predicted by Theorem 4.1. For values of $n$ ranging from $10^2$ to $10^4$, we compute $\text{Pers}_p(\text{dgm}_i^{(1)}(\mathsf{A}_n))$ in three scenarios: points sampled on a circle for $i = 0$, and points sampled on a torus for $i = 0$ and $i = 1$. The correct rates of convergence are observed on a log-log plot, see Figure 5. For $i = 1$, we remark that the asymptotic regime starts at larger values of $n$, above $n = 10^3$.

**Convergence of $\mu_{n,i}$.** We sample $n$ points on a torus by uniformly sampling the two angles $(\theta, \phi)$ parametrizing the torus. We obtain a (nonuniform) probability measure, having density $f$. We then compute, for various values of $n$, the measure $\mu_{n,1}$. The measure is approximated by kernel density estimation (see Figure 6). We approximate in a similar manner the measure $\mu_{\infty,1,2}$ by sampling $n = 10^5$ points on a square. We then apply the change of variable formula (9) to compute the theoretical limit $\mu_{f,1}$. The distance $\text{OT}_2(\mu_{n,1}, \mu_{f,1})$ is then computed by approximating the measures on a grid: the distance converges to 0 as predicted by Theorem 4.1. See also Figure 6.

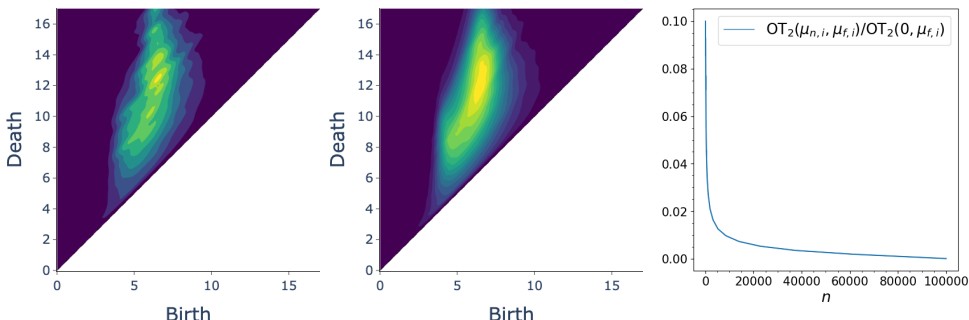

Figure 6: Left: Heatmap of $\mu_{n,1}$ for $n = 5 \cdot 10^4$ points sampled on the torus with density $f$. Center: Heatmap of $\mu_{f,1}$. Right: $\text{OT}_2$ distance between $\mu_{n,1}$ and $\mu_{f,1}$ (normalized by $\text{OT}_2(0, \mu_{f,i})$) for $n$ ranging from $10^2$ to $10^5$.

# 6   Conclusion

Under the manifold hypothesis, we have greatly refined earlier work regarding the persistent homology of subsamples of compact sets, with especially strong results when the sampling is either random or well-behaved. In particular, we have precisely described the PDs of such samplings, and provided new convergence guarantees w.r.t. the $p$-Wassertein distances, as well as detailed asymptotics for their total $\alpha$-persistence. This results in a deeper understanding of these objects, which play an important role in ML techniques applied to TDA. The main limitations of our work were the assumptions that the data is sampled from a submanifold, and without any noise. Relaxing those assumptions, as well as establishing similar guarantees for Vietoris-Rips complexes, could be the subject of future research. We also plan on exploring the consequences of our findings regarding the persistent homology dimension [3, 66, 9, 65] of submanifolds.

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

**Appendices**

## A  Outline of the appendix

We recall some basic definitions regarding persistent homology in Appendix B. In Appendix C, we define topological Morse functions and discuss some of the properties of distance functions to compact sets. We also detail the genericity conditions from [6], and prove Proposition 3.2. We prove Lemma 2.1 and Theorem 2.2 in Appendix D, and Theorem 3.3 in Appendix E. We rigorously define the partial optimal transport distance $\mathrm{OT}_p$ between Radon measures on $\Omega$ in Appendix F, before proving Theorem 4.1, Proposition 4.2, Corollary 4.3 and Corollary 4.4 in Appendix G. Finally, we provide some details on our experimental setup in Appendix H. Throughout the appendices, we write $C = C(a, b, \dots)$ to implicitly state that a newly introduced constant $C$ depends only on some objects $a, b, \dots$ (e.g. $C = C(\mathsf{M})$).

## B  Persistent homology

Although this work is only concerned with PDs with respect to the Čech filtration, it is more natural to define PDs for the sublevel sets of proper continuous functions that are bounded below. We refer to [22] for a thorough introduction to persistent homology and PDs in an even more general context.

Let $f : \mathbb{R}^d \to \mathbb{R}$ be a proper continuous function that is bounded below–e.g., the distance function to a compact set. For $t \in \mathbb{R}$, let $X_t = f^{-1}(-\infty, t]$ be the sublevel set of $f$ at level $t$. The collection $(X_t)_{t \in \mathbb{R}}$ is called a filtration. Let $i \geq 0$ be an integer. We let $H_i(X_t)$ be the homology group of degree $i$ with coefficients in any fixed field $\mathcal{F}$ (e.g. $\mathcal{F} = \mathbb{Z}/2\mathbb{Z}$ is a popular choice) of $X_t$. For $r < s$, the inclusion between the sublevel sets at levels $r$ and $s$ induces a map $\iota_{i,r,s}$ at the homology level. We define the persistent Betti number

$$\beta_{i,r,s}(f) = \mathrm{rk}(\iota_{i,r,s} : H_i(X_r) \to H_i(X_s)). \tag{10}$$

As shown in [22, Corollary 3.34], this number is finite. Informally, it represents the number of $i$th dimensional topological features present in the sublevel set at level $r$ that are still present at level $s$.

Define the extended half-plane $\Omega_\infty = \{u = (u_1, u_2) \in (\mathbb{R} \cup \{-\infty, +\infty\})^2 : -\infty \leq u_1 < u_2 \leq +\infty\}$ and $\Omega = \{u = (u_1, u_2) \in \mathbb{R}^2 : u_1 < u_2\}$. The collection of persistent Betti numbers $(\beta_{i,r,s}(f))_{r<s}$ defines a multiset of points in $\Omega_\infty$, called the persistence diagram $\mathrm{dgm}_i(f)$ of degree $i$ of $f$. See e.g. [22] for its precise definition.

Though persistence diagrams can have points with infinite coordinates, these will be of little interest in the cases considered in this article[5]. To simplify our notation and definitions, we let from now on

---

[5]The persistence diagram of degree $i$ of the distance function to a set has no point with infinite coordinates if $i > 0$, and a single such point if $i = 0$ whose coordinates are $(0, +\infty)$.

$\mathrm{dgm}_i(f)$ denote the finite part (i.e. the points whose coordinates are finite) of the diagram of degree $i$ of $f$, and we assume that every diagram considered henceforth has no point with infinite coordinates. In particular, they are all multisets of the half-plane $\Omega := \{u = (u_1, u_2) \in \mathbb{R}^2 : u_1 < u_2\}$, which leads to the following definition: the space $\mathcal{D}$ of persistence diagrams is the set of all multisets $a$ in $\Omega$ that contain a finite number of points[6] in every quadrant $Q_u$, $u \in \Omega$.

Note that while persistence diagrams capture key topological features in an interpretable fashion, they are not complete metric invariants: different sets can have equal persistence diagrams, see e.g. [69]. As such, they can be seen as a form of topology-centered dimensionality reduction technique. Other topological invariants have been proposed, some of which are complete, such as those from [77, 52].

## C   Distance functions, topological Morse functions and generic submanifolds

In this section, we consider a special class of PDs –those yielded by the distance function to a compact set $\mathsf{A} \subset \mathbb{R}^d$. We give a detailed description of such a PD when $\mathsf{A} = \mathsf{M}$ is a generic submanifold.

The theory of persistent homology was historically developed for Morse functions $f : \mathbb{R}^d \to \mathbb{R}$. A Morse function $f$ is a $C^2$ function whose critical points $x$ (points for which $d_x f = 0$) are non-degenerate (meaning that the Hessian of $f$ at $x$ is non-degenerate). The index of the critical point $x$ is equal to the number of negative eigenvalues of the corresponding Hessian. The changes of topology of the sublevel sets of such a function are perfectly understood. First, the isotopy lemma states that two sublevel sets $f^{-1}(-\infty, u_1]$ and $f^{-1}(-\infty, u_2]$ are isotopic if no critical values are found in the interval $[u_1, u_2]$ and if $f^{-1}[u_1, u_2]$ is compact. Second, if $f^{-1}[u_1, u_2]$ is compact and contains the critical points $x_1, \ldots, x_K$, then $f^{-1}(-\infty, u_2]$ has the homotopy type of $f^{-1}(-\infty, u_1]$ with cells $e_k$ of dimension equal to the index of $x_k$ attached along their boundaries (see e.g. [56] for a much more in-depth treatment).

In such a situation, the PD $\mathrm{dgm}_i(f)$ has a clear interpretation: the coordinates $(u_1, u_2)$ of a point $u \in \mathrm{dgm}_i(f)$ correspond to the critical value of a critical point of index $i$ and $i + 1$ respectively. Informally, the corresponding $i$-dimensional topological feature appears with the attachment of a $i$-dimensional cell at value $u_1$, and is "killed" by the attachment of a $(i + 1)$-dimensional cell at value $u_2$.

The notion of Morse function extends to continuous functions with the following definition.

**Definition C.1** (Topological Morse functions [58]). *Let $U \subset \mathbb{R}^d$ be an open set and let $f : U \to \mathbb{R}$ be a continuous function.*

- *A point $z \in U$ is said to be a topological regular point of $f$ if there is a homeomorphism $\phi : V_1 \to V_2$ between open neighborhoods $V_1$ of $0$ in $\mathbb{R}^d$ and $V_2$ of $U$ in $\mathbb{R}^d$ with $\phi(0) = z$ and such that for all $x = (x_1, \ldots, x_d) \in V_1$,*

$$f \circ \phi(x) = f(z) + x_d. \tag{11}$$

- *A point $z \in U$ is said to be a topological critical point of $f$ if it is not a topological regular point of $f$.*

- *A point $z \in U$ is said to be a non-degenerate topological critical point of $f$ of index $i$ if there exist an integer $0 \leq i \leq d$ and a homeomorphism $\phi : V_1 \to U_2$ between open neighborhoods $V_1$ of $0$ in $\mathbb{R}^d$ and $V_2$ of $U$ in $\mathbb{R}^d$ with $\phi(0) = z$ such that for all $x = (x_1, \ldots, x_d) \in V_1$,*

$$f \circ \phi(x) = f(z) - \sum_{j=1}^{i} x_j^2 + \sum_{j=i+1}^{d} x_j^2. \tag{12}$$

- *The function $f$ is said to be a topological Morse function if all its topological critical points are non-degenerate.*

For topological Morse functions, both the isotopy lemma and the handle attachment lemma stay valid:

---

[6]This corresponds to the set of diagrams of $q$-tame persistence modules as defined in [22].

**Lemma C.2** (Isotopy Lemma). *Let $f : \mathbb{R}^d \to \mathbb{R}$ be a proper topological Morse function. Let $a < b$ be such that $f^{-1}[a, b]$ contains no topological critical point. Then $f^{-1}(-\infty, a]$ is a deformation retract of $f^{-1}(-\infty, b]$.*

**Lemma C.3** (Handle Attachment Lemma). *Let $f : \mathbb{R}^d \to \mathbb{R}$ be a proper topological Morse function. Let $c \in \mathbb{R}$ and $\varepsilon > 0$ be such that $f^{-1}[c - \varepsilon, c + \varepsilon]$ contains no topological critical point except for $z_1, \ldots, z_k \in f^{-1}(c)$, with $z_j$ of index $i_j$. Then $f^{-1}(-\infty, c + \varepsilon]$ is homotopically equivalent to $f^{-1}(-\infty, c - \varepsilon]$ with cells $B^{i_1}, \ldots, B^{i_k}$ of dimension $i_1, \ldots, i_k$ attached, i.e.*

$$f^{-1}(-\infty, c + \varepsilon] \simeq f^{-1}(-\infty, c - \varepsilon] \cup B^{i_1} \cup \ldots \cup B^{i_k}.$$

Their proofs are roughly the same as for smooth Morse functions –see [71, Theorems 4 and 5] for details. As a consequence, the description of PDs for Morse functions also stays valid for topological Morse functions.

In this paper, we are interested in the PDs of the distance function $d_\mathsf{A}$ to various compact sets $\mathsf{A}$, called the Čech persistence diagram[7] of $\mathsf{A}$ and denoted by $\mathrm{dgm}_i(\mathsf{A})$. In general, such a function is not a topological Morse function. Instead, changes in the topology of its sublevels can be partially (though less completely than for a Morse function) described in terms of zeros of its generalized gradient, which is defined at $y \in \mathbb{R}^d \backslash \mathsf{A}$ as

$$\nabla d_\mathsf{A}(y) = \frac{y - c(\sigma_\mathsf{A}(y))}{d_\mathsf{A}(y)}, \tag{13}$$

where $\sigma_\mathsf{A}(y) = \{x \in \mathsf{A} : \|x - y\| = d_\mathsf{A}(y)\}$ is the set of projections of $y$ on $\mathsf{A}$ and $c(\tau)$ represents the center of the smallest enclosing ball of a set $\tau$. When $y \in \mathbb{R}^d \backslash \mathsf{A}$ satisfies $\nabla d_\mathsf{A}(y) = 0$, $y$ is called a *differential critical point* of $d_\mathsf{A}$. We let $\mathrm{Crit}(\mathsf{A})$ denote the set of differential critical points of $d_\mathsf{A}$. An adapted version of the Isotopy Lemma remains true, as shown in [43]:

**Lemma C.4** (Isotopy Lemma for Distance Functions). *If $0 < a < b$ are such that $d_\mathsf{A}^{-1}[a, b]$ contains no differential critical point of $d_\mathsf{A}$, then $d_\mathsf{A}^{-1}(-\infty, a]$ is a deformation retract of $d_\mathsf{A}^{-1}(-\infty, b]$. Consequently, any $(u_1, u_2) \in \mathrm{dgm}_i(\mathsf{A})$ is such that $u_1, u_2 \notin [a, b]$.*

Without further assumptions, little else can be said regarding the topology of the sublevels of $d_\mathsf{A}$; in particular, there is no equivalent to the Handle Attachment Lemma to control the changes occurring at critical values.

However, the distance function $d_\mathsf{M}$ to a compact $C^2$ submanifold $\mathsf{M} \subset \mathbb{R}^d$ turns out to be a topological Morse function in a "generic" sense, as was proven by Arnal, Cohen-Steiner and Divol.

**Theorem C.5** (Genericity Theorem [6]). *Let $M$ be a compact $C^2$ (abstract) manifold. Then the set of $C^2$ embeddings $i : M \to \mathbb{R}^d$ such that*

1. *the distance function $d_{i(M)} : \mathbb{R}^d \backslash i(M) \to \mathbb{R}$ is a topological Morse function,*

2. *for every $z \in \mathrm{Crit}(\mathsf{M})$, the projections $\sigma_\mathsf{M}(z)$ are the vertices of a non-degenerate simplex of $\mathbb{R}^d$ and $z$ belongs to its relative interior,*

3. *the set $\mathrm{Crit}(\mathsf{M})$ is finite,*

4. *for every $z \in \mathrm{Crit}(\mathsf{M})$ and every $x \in \sigma_\mathsf{M}(z)$, the sphere $S(z, d_\mathsf{M}(z))$ is non-osculating $\mathsf{M}$ at $x$, in the sense that there exist $\delta > 0$ and $\alpha > 0$ such that for all $y \in \mathsf{M} \cap B(x, \delta)$,*

$$\|y - z\|^2 \geq \|x - z\|^2 + \alpha \|y - x\|^2, \tag{14}$$

5. *there exist constants $C > 0$ and $\mu_0 \in (0, 1)$ such that for every $\mu \in [0, \mu_0)$, any point $x$ such that $\|\nabla d_\mathsf{M}(x)\| \leq \mu$ is at distance at most $C\mu$ from $\mathrm{Crit}(\mathsf{M})$,*

*is open and dense in the set of $C^2$ embeddings $M \to \mathbb{R}^d$ for the Whitney $C^2$-topology.*

When $\mathsf{M}$ is generic, the topological critical points of $d_\mathsf{M}$ coincide with its differential critical points (see [6, Theorem 1.8]), and the Čech PD $\mathrm{dgm}_i(\mathsf{M})$ can be related to the critical points of $d_\mathsf{M}$ in the same way as for smooth Morse function. We are now in position to prove Proposition 3.2, which we restate for the reader's convenience:

---

[7]The distance function $d_\mathsf{A}$ is proper due to the compacity of $\mathsf{A}$, hence its persistence module is $q$-tame and the associated persistence diagram is well-defined –see [22].

**Proposition 3.2** (Čech PDs of generic submanifolds). *Let* $M$ *be a generic submanifold of* $\mathbb{R}^d$. *Then, for any integer* $i \geq 0$*, the PD* $\mathrm{dgm}_i(M)$ *contains finitely many points. In particular,* $\mathrm{Pers}_\alpha(\mathrm{dgm}_i(M)) < +\infty$ *for all* $\alpha > 0$.

*Proof.* The proof is similar to that used in the case of smooth Morse functions, with the Isotopy Lemma and the Handle Attachment Lemma playing the same role; we briefly summarize it for completeness nonetheless. We do not distinguish between differential and topological critical values and points, as they coincide.

The set $\mathrm{Crit}(M)$ is finite: this is simply Condition 3. from the Genericity Theorem. The Isotopy Lemma shows that there is no change in homotopy type between $M^a$ and $M^b$ if $0 < a < b$ and $[a,b]$ contains no critical value. Similarly, $M$ has the same homotopy type as $M^a$ if $a > 0$ is small enough. Hence changes in homology in the offsets can only occur at 0, when the entire submanifold appears in the filtration, and at critical values of $d_M$, and there can be no birth or death of interval between them.

Let us now consider a critical value $c > 0$ and $0 < \varepsilon < c$ such that $d_M^{-1}[c - \varepsilon, c + \varepsilon]$ contains no critical point except for $z_1, \ldots, z_k \in d_M^{-1}(c)$, where $z_j$ is of index $i_j$. Then the Handle Attachment Lemma states that $M^{c+\varepsilon}$ is homotopically equivalent to $M^{c-\varepsilon}$ with cells $B^{i_1}, \ldots, B^{i_k}$ of dimension $i_1, \ldots, i_k$ attached, i.e.

$$M^{c+\varepsilon} \simeq M^{c-\varepsilon} \cup B^{i_1} \cup \ldots \cup B^{i_k}.$$

Let $i \geq 1$, and let $D_{i,b}$ be the dimension of the cokernel of $H_i(M^{c-\varepsilon}) \to H_i(M^{c+\varepsilon})$ (where the map is induced by the inclusion): it is precisely the number of births of intervals between $c - \varepsilon$ and $c + \varepsilon$ (hence precisely at $c$) in the persistence module of degree $i$ of the filtration. Similarly, the dimension $D_{i-1,d}$ of the kernel of $H_{i-1}(M^{c-\varepsilon}) \to H_{i-1}(M^{c+\varepsilon})$ is the number of deaths of intervals at $c$ in the persistence module of degree $i - 1$ of the filtration. A straightforward application of the Mayer-Vietoris exact sequence yields that $D_{i,b} + D_{i-1,d}$ is exactly equal to the number of $i$-dimensional cells among $B^{i_1}, \ldots, B^{i_k}$, meaning that each $i$-cell corresponds exactly either to the birth of an interval for the homology of degree $i$, or to the death of an interval for the homology of degree $i - 1$ (in particular, an $i - 1$-cell and an $i$-cell cannot "cancel each other out"). This proves that for any $i \geq 1$, the multiset of critical values $d_M(z)$ of critical points $z$ of $M$ of index $i$ is equal to the multiset

$$\{u_1 : \ (u_1, u_2) \in \mathrm{dgm}_i(M), \ u_1 \neq 0\} \cup \{u_2 : \ (u_1, u_2) \in \mathrm{dgm}_{i-1}(M)\}. \tag{15}$$

This fact proves in turn that $\mathrm{dgm}_i(M)$ is finite for any $i \geq 0$, which immediately implies that $\mathrm{Pers}_\alpha(\mathrm{dgm}_i(M)) < +\infty$ for all $\alpha > 0$. $\qquad\square$

# D   Proofs of Section 2

For $M$ a $m$-dimensional differential submanifold of $\mathbb{R}^d$ and $x \in M$, we let $T_x M$ be the tangent space of $M$ at $x$, which is identified with a linear subspace of $\mathbb{R}^d$. We denote by $\pi_x : \mathbb{R}^d \to T_x M$ the orthogonal projection on this subspace and let $\pi_x^\perp = id - \pi_x$ be the orthogonal projection on the normal space at $x$. A key property, that we will repeatedly used, is that the reach of $M$ controls the deviation of the manifold $M$ from its tangent space. Namely, [39, Theorem 4.18] states that for all $y \in M$,

$$\|\pi_x^\perp(x - y)\| \leq \frac{\|x - y\|^2}{2\tau(M)}. \tag{16}$$

We also define the *weak feature size of* $M$, denoted by $\mathrm{wfs}(M)$, as the minimal distance between a critical point of $M$ and $M$. As by definition, the projection is not unique at a critical point, we must have $\mathrm{wfs}(M) \geq \tau(M)$.

We first prove Lemma 2.1:

**Lemma 2.1.** *Let* $M \subset \mathbb{R}^d$ *be a compact submanifold with positive reach and let* $A \subset M$ *be a compact set with* $d_H(A, M) \leq \varepsilon$ *for some* $\varepsilon > 0$. *Let* $z \in \mathbb{R}^d \backslash M$. *Then,* $|d_M(z) - d_A(z)| \leq \frac{\varepsilon^2}{2d_M(z)}\left(1 + \frac{d_M(z)}{\tau(M)}\right)$.

*Proof.* As $A \subset M$, we have $d_M(z) \leq d_A(z)$. Let $x$ be a projection of $z$ onto $M$, and let $y \in A$ be a point at distance less than $\varepsilon$ from $x$. Then, using (16) and the fact that $z - x$ is orthogonal to $T_x M$, we obtain that

$$
\begin{aligned}
d_A(z)^2 &\leq \|z - y\|^2 = \|z - x\|^2 + \|x - y\|^2 + 2\langle z - x, x - y \rangle \\
&\leq d_M(z)^2 + \varepsilon^2 + 2\langle z - x, \pi_x^\perp(x - y) \rangle \\
&\leq d_M(z)^2 + \varepsilon^2 + \frac{d_M(z)\varepsilon^2}{\tau(M)}.
\end{aligned}
$$

Hence,

$$
d_A(z) - d_M(z) = \frac{d_A(z)^2 - d_M(z)^2}{d_A(z) + d_M(z)} \leq \frac{\varepsilon^2}{2d_M(z)}\left(1 + \frac{d_M(z)}{\tau(M)}\right).
$$

$\square$

We can now prove Theorem 2.2:

**Theorem 2.2** (Improved Bottleneck Stability Theorem). *Let $M \subset \mathbb{R}^d$ be a compact submanifold with positive reach and let $A \subset M$ be a compact set such that $d_H(A, M) \leq \varepsilon < \tau(M)/4$. Let $i \geq 0$ be an integer. Then $\mathrm{dgm}_i(A)$ is the union of three regions $\mathrm{dgm}_i^{(1)}(A) := \mathrm{dgm}_i(A) \cap \{u_1, u_2 \leq \varepsilon + \frac{\varepsilon^2}{\tau(M)}\}$, $\mathrm{dgm}_i^{(2)}(A) := \mathrm{dgm}_i(A) \cap \{u_1 \leq \varepsilon, u_2 \geq \tau(M) - \frac{\varepsilon^2}{\tau(M)}\}$ and $\mathrm{dgm}_i^{(3)}(A) := \mathrm{dgm}_i(A) \cap \{u_1, u_2 \geq \tau(M) - \frac{\varepsilon^2}{\tau(M)}\}$.*

*Furthermore, let $C = \frac{2}{\tau(M)}\left(1 + \frac{R(M)}{\tau(M)}\right)$, where $R(M)$ is the radius of the smallest ball that contains $M$. There exists an optimal matching $\gamma : \mathrm{dgm}_i(A) \cup \partial\Omega \to \mathrm{dgm}_i(M) \cup \partial\Omega$ for the bottleneck distance between $\mathrm{dgm}_i(A)$ and $\mathrm{dgm}_i(M)$ such that*

- *Region (1): If $u \in \mathrm{dgm}_i^{(1)}(A)$, then $\gamma(u) \in \partial\Omega$ and $\|u - \gamma(u)\|_\infty \leq \varepsilon$.*

- *Region (2): If $u \in \mathrm{dgm}_i^{(2)}(A)$, then $\gamma(u)$ is of the form $(0, v_2)$ and $|u_2 - v_2| \leq C\varepsilon^2$. The number of such points is finite and depends only on $M$.*

- *Region (3): If $u \in \mathrm{dgm}_i^{(3)}(A)$, then $\|u - \gamma(u)\|_\infty \leq C\varepsilon^2$.*

*Proof.* Proposition 5 from [7] states that if $\varepsilon = d_H(A, M) < (\sqrt{2} - 1)\tau(M)$, then the offset $A^r$ deformation-retracts onto $M$ for any

$$
r \in \left[\frac{1}{2}(\tau(M) + \varepsilon - \sqrt{\Delta}), \frac{1}{2}(\tau(M) + \varepsilon + \sqrt{\Delta})\right]
$$

and $\Delta = \tau(M)^2 - 2\varepsilon\tau(M) - \varepsilon^2$. Under the stronger assumption that $d_H(A, M) < \tau(M)/4$, and using elementary calculus, we find that the offset $A^r$ deformation-retracts onto $M$ for any

$$
r \in \left[\varepsilon + \frac{\varepsilon^2}{\tau(M)}, \tau(M) - \frac{\varepsilon^2}{\tau(M)}\right].
$$

This means in particular that the homology type, hence the homology, of $A^r$ does not change in that interval; as a result, there can be no birth or death of intervals in the Čech persistence diagrams of $A$ between $\varepsilon + \frac{\varepsilon^2}{\tau(M)}$ and $\tau(M) - \frac{\varepsilon^2}{\tau(M)}$, and all $(u_1, u_2) \in \mathrm{dgm}_i(A)$ must either be such that $u_1, u_2 \leq \varepsilon + \frac{\varepsilon^2}{\tau(M)}$, or $u_1 \leq \varepsilon + \frac{\varepsilon^2}{\tau(M)}, u_2 \geq \tau(M) - \frac{\varepsilon^2}{\tau(M)}$, or $u_1, u_2 \geq \tau(M) - \frac{\varepsilon^2}{\tau(M)}$. This almost proves that the partition of $\mathrm{dgm}_i(A)$ into $\mathrm{dgm}_i^{(1)}(A)$, $\mathrm{dgm}_i^{(2)}(A)$ and $\mathrm{dgm}_i^{(3)}(A)$ as defined in the statement is correct, except that the definition of $\mathrm{dgm}_i^{(2)}(A)$ requires that $u_1 \leq \varepsilon$, whereas we only have obtained that $u_1 \leq \varepsilon + \frac{\varepsilon^2}{\tau(M)}$.

Let $\gamma : \mathrm{dgm}_i(A) \cup \partial\Omega \to \mathrm{dgm}_i(M) \cup \partial\Omega$ be any optimal matching for the bottleneck distance: then the Bottleneck Stability Theorem states that any point $u = (u_1, u_2) \in \mathrm{dgm}_i(A)$ is such that $\|u - \gamma(u)\|_\infty \leq d_H(A, M) \leq \varepsilon$. Suppose that $u$ is such that $u_1 \leq \varepsilon + \frac{\varepsilon^2}{\tau(M)}$ and $u_2 \geq \tau(M) - \frac{\varepsilon^2}{\tau(M)}$.

Its distance in the infinity norm to the diagonal $\partial\Omega$ is equal to $(u_2 - u_1)/2 \geq (\tau(\mathsf{M}) - \varepsilon - 2\varepsilon^2/\tau(\mathsf{M}))/2 \geq \frac{5}{16}\tau(\mathsf{M}) > \varepsilon$, where we use the fact that $\varepsilon < \tau(\mathsf{M})/4$. Hence $\gamma(u)$ must belong to $\mathrm{dgm}_i(\mathsf{M})$. Let $(v_1, v_2)$ denote $\gamma(u)$. As in the proof of Proposition 3.2, the Isotopy Lemma shows that $\mathrm{dgm}_i(\mathsf{M})$ only contains two types of points: points of the shape $(0, w_2)$, which correspond to the homology of M itself, and points of the shape $(w_1, w_2)$, where in both cases $w_1, w_2$ are critical values of $d_\mathsf{M}$. In particular, both $w_1$ and $w_2$ must be greater than $\mathrm{wfs}(\mathsf{M})$. Let $\mathrm{dgm}_i^{(2)}(\mathsf{M})$ denote the multiset of all points of the first type, and $\mathrm{dgm}_i^{(3)}(\mathsf{M})$ denote the multiset of all points of the second type. If $v_1$ was non-zero, it would have to be greater than $\mathrm{wfs}(\mathsf{M}) \geq \tau(\mathsf{M})$, and we would have $\|u - \gamma(u)\|_\infty \geq |u_1 - v_1| \geq \tau(\mathsf{M}) - \varepsilon - \frac{\varepsilon^2}{\tau(\mathsf{M})} \geq \tau(\mathsf{M})/2 > \varepsilon$, which would be a contradiction (we once again use that $\varepsilon < \tau(\mathsf{M})/4$). Hence $v_1$ must be 0 (i.e. $\gamma(u) \in \mathrm{dgm}_i^{(2)}(\mathsf{M})$), and $u_1 = |u_1 - v_1| \leq \|u - \gamma(u)\|_\infty \leq \varepsilon$. This proves the correctness of the partition into regions from the statement.

Consider now $u = (u_1, u_2) \in \mathrm{dgm}_i^{(1)}(\mathsf{A})$. All $v = (v_1, v_2) \in \mathrm{dgm}_i(\mathsf{M})$ are such that

$$\|u - v\|_\infty \geq v_2 - u_2 \geq \mathrm{wfs}(\mathsf{M}) - \varepsilon - \frac{\varepsilon^2}{\tau(\mathsf{M})} \geq \tau(\mathsf{M})\left(1 - \frac{1}{4} - \frac{1}{16}\right) > \varepsilon \geq \|u - \gamma(u)\|_\infty,$$

hence $\gamma(u)$ must belong to $\partial\Omega$. This completes the proof of the first bullet point of the statement.

We have already shown that if $\gamma$ is an optimal matching and $u \in \mathrm{dgm}_i^{(2)}(\mathsf{A})$, then $\gamma(u) \in \mathrm{dgm}_i(\mathsf{M})$ is of the form $(0, v_2)$. As $\gamma$ maps at most a single point of $\mathrm{dgm}_i^{(2)}(\mathsf{A})$ to each point of $\mathrm{dgm}_i(\mathsf{M})$, the number of such points is upper bounded by the number of points of the form $(0, v_2)$ with $v_2 \geq \mathrm{wfs}(\mathsf{M})$ in $\mathrm{dgm}_i(\mathsf{M})$. Though $\mathrm{dgm}_i(\mathsf{M})$ need not be finite (M is not assumed to be generic), applying Corollary 3.34 from [22] to $d_\mathsf{M}$ shows that $\mathrm{dgm}_i(\mathsf{M})$ is q-tame, and in particular that it contains only a finite number $N$ of points $(v_1, v_2)$ with $v_1 \leq \mathrm{wfs}(\mathsf{M})/4$ and $v_2 \geq \mathrm{wfs}(\mathsf{M})/2$. Hence the cardinality of $\mathrm{dgm}_i^{(2)}(\mathsf{A})$ is bounded by $N$.

It only remains to show that $\gamma$ can be chosen such that if $u = (u_1, u_2) \in \mathrm{dgm}_i^{(2)}(\mathsf{A})$, respectively $u' \in \mathrm{dgm}_i^{(3)}(\mathsf{A})$, then $|u_2 - \gamma(u)_2| \leq C\varepsilon^2$, respectively $\|u' - \gamma(u')\|_\infty \leq C\varepsilon^2$. To that end, remember first that as shown above, our starting optimal matching $\gamma$ must be such that points in $\mathrm{dgm}_i^{(2)}(\mathsf{A})$ must be matched to points $\mathrm{dgm}_i^{(2)}(\mathsf{M})$. Conversely and for the same reasons, points in $\mathrm{dgm}_i^{(2)}(\mathsf{M})$ must be matched to points in $\mathrm{dgm}_i^{(2)}(\mathsf{A})$. Similarly, points $u \in \mathrm{dgm}_i^{(3)}(\mathsf{A})$ can only be matched to points in $\mathrm{dgm}_i^{(3)}(\mathsf{M})$ or to the diagonal $\partial\Omega$; otherwise, $\|u - \gamma(u)\|_\infty \geq |u_1 - \gamma(u)_1| = u_1$ would be too large. Likewise, points in $\mathrm{dgm}_i^{(3)}(\mathsf{M})$ can only be matched to points in $\mathrm{dgm}_i^{(3)}(\mathsf{A})$ or to the diagonal. Hence $\gamma$ defines disjoint submatchings

$$\gamma^{(2)} : \mathrm{dgm}_i^{(2)}(\mathsf{A}) \to \mathrm{dgm}_i^{(2)}(\mathsf{M})$$

and

$$\gamma^{(3)} : \mathrm{dgm}_i^{(3)}(\mathsf{A}) \cup \partial\Omega \to \mathrm{dgm}_i^{(3)}(\mathsf{M}) \cup \partial\Omega.$$

Now let $R(\mathsf{M})$ be the radius of the smallest ball that contains M, and consider the functions

$$a : \mathbb{R}^d \to \mathbb{R}, x \mapsto \min(\max(d_\mathsf{A}(x), \tau(\mathsf{M})/2), R(\mathsf{M}))$$

and

$$m : \mathbb{R}^d \to \mathbb{R}, x \mapsto \min(\max(d_\mathsf{M}(x), \tau(\mathsf{M})/2), R(\mathsf{M})).$$

Let us compare the persistence diagrams $\mathrm{dgm}_i(a)$ and $\mathrm{dgm}_i(m)$ of the sublevel sets filtration of $a$ and $m$ and the Čech persistence diagrams $\mathrm{dgm}_i(\mathsf{A})$ and $\mathrm{dgm}_i(\mathsf{M})$ respectively (which are by definition the persistence diagrams of the sublevel sets filtration of $d_\mathsf{A}$ and $d_\mathsf{M}$).

Note first that $d_\mathsf{A}$ and $d_\mathsf{M}$ can have no critical values strictly greater than $R(\mathsf{M})$, as a critical point must belong to the convex hull of its projections. Note also that for any $t \in [\tau(\mathsf{M})/2, R(\mathsf{M})]$, the sublevel set $a^{-1}(-\infty, t]$ is exactly equal to $d_\mathsf{A}^{-1}(-\infty, t] = \mathsf{A}^t$. Consequently, $\mathrm{dgm}_i(a)$ contains exactly two disjoint types of points. The first type are points of the form $(\tau(\mathsf{M})/2, u_2)$, which are in bijection with the points $(u_1, u_2) \in \mathrm{dgm}_i(\mathsf{A})$ with $u_1 \leq \tau(\mathsf{M})/2$ (the bijection maps $(u_1, u_2) \mapsto (\tau(\mathsf{M})/2, u_2)$); those are exactly the points in $\mathrm{dgm}_i^{(2)}(\mathsf{A})$. The second type are points of the form $(u_1, u_2)$ with $u_1 \geq \tau(\mathsf{M})/2$, which are in trivial bijection (the bijection is the identity) with the points in $\mathrm{dgm}_i(\mathsf{A})$

that satisfy the same condition; those are exactly the points of $\mathrm{dgm}_i^{(3)}(\mathsf{A})$. The points of $\mathrm{dgm}_i^{(1)}(\mathsf{A})$ cannot be "seen" in $\mathrm{dgm}_i(a)$. We will call $\mathrm{dgm}_i^{(2)}(a)$ the subdiagram comprised of the points of the first type, and $\mathrm{dgm}_i^{(3)}(a)$ the subdiagram of $\mathrm{dgm}_i(a)$ comprised of the points of the second type.

Similarly, $\mathrm{dgm}_i(m)$ contains two types of points: the first type are points of the form $(\tau(\mathsf{M})/2, v_2)$, which are in bijection with the points $(0, v_2) \in \mathrm{dgm}_i(\mathsf{M})$ (the bijection maps $(0, v_2) \mapsto (\tau(\mathsf{M})/2, v_2)$); those are exactly the points in $\mathrm{dgm}_i^{(2)}(\mathsf{M})$. The second type are points of the form $(v_1, v_2)$ with $v_1 \geq \tau(\mathsf{M})/2$, which are in trivial bijection (the bijection is the identity) with the points in $\mathrm{dgm}_i(\mathsf{M})$ that satisfy the same condition; those are exactly the points of $\mathrm{dgm}_i^{(3)}(\mathsf{M})$. We will call $\mathrm{dgm}_i^{(2)}(m)$ the subdiagram of $\mathrm{dgm}_i(m)$ comprised of the points of the first type, and $\mathrm{dgm}_i^{(3)}(m)$ the subdiagram comprised of the points of the second type.

Recall that Lemma 2.1 states that $|d_\mathsf{M}(z) - d_\mathsf{A}(z)| \leq \frac{\varepsilon^2}{2d_\mathsf{M}(z)}\left(1 + \frac{d_\mathsf{M}(z)}{\tau(\mathsf{M})}\right)$ for any $z \in \mathbb{R}^d \backslash \mathsf{M}$. If $z \in \mathbb{R}^d$ is such that $d_\mathsf{A}(z) \leq \tau(\mathsf{M})/2$, then $a(z) = m(z) = \tau(\mathsf{M})/2$; if it is such that $d_\mathsf{M}(z) \geq R(\mathsf{M})$, then $a(z) = m(z) = R(\mathsf{M})$. Otherwise, $d_\mathsf{M}(z) \geq d_\mathsf{A}(z) - d_H(\mathsf{A}, \mathsf{M}) \geq \tau(\mathsf{M})/4$ and $d_\mathsf{M}(z) \leq R(\mathsf{M})$, hence
$$|d_\mathsf{M}(z) - d_\mathsf{A}(z)| \leq \frac{\varepsilon^2}{2d_\mathsf{M}(z)}\left(1 + \frac{d_\mathsf{M}(z)}{\tau(\mathsf{M})}\right) \leq \frac{2\varepsilon^2}{\tau(\mathsf{M})}\left(1 + \frac{R(\mathsf{M})}{\tau(\mathsf{M})}\right) = C\varepsilon^2,$$
where $C$ is as defined in the proposition. This means that
$$\|m - a\|_\infty \leq C\varepsilon^2.$$
Due to the Bottleneck Stability Theorem, the diagrams $\mathrm{dgm}_i(a)$ and $\mathrm{dgm}_i(m)$ must be at bottleneck distance less than $C\varepsilon^2$.

Furthermore, let $\delta$ denote $\max_{u \in \mathrm{dgm}_i(\mathsf{A}) \cup \partial\Omega} \|u - \gamma(u)\|_\infty$. The matching $\gamma$ (and in particular the submatchings $\gamma^{(2)}$ and $\gamma^{(3)}$) also induces (through the correspondence detailed above between the points of $\mathrm{dgm}_i(a)$ and a subset of the points of $\mathrm{dgm}_i(\mathsf{A})$) a matching $\gamma'$ between $\mathrm{dgm}_i(a)$ and $\mathrm{dgm}_i(m)$ such that $\max_{u \in \mathrm{dgm}_i(a) \cup \partial\Omega} \|u - \gamma'(u)\|_\infty \leq \delta$. Hence the bottleneck distance between $\mathrm{dgm}_i(a)$ and $\mathrm{dgm}_i(m)$ is at most $\min(\delta, C\varepsilon^2)$.

Let $\beta : \mathrm{dgm}_i(a) \cup \partial\Omega \to \mathrm{dgm}_i(m) \cup \partial\Omega$ be an optimal matching for the bottleneck distance. For similar reasons as for $\gamma$, the matching $\beta$ can also be decomposed into two disjoint submatchings
$$\beta^{(2)} : \mathrm{dgm}_i^{(2)}(a) \to \mathrm{dgm}_i^{(2)}(m)$$
and
$$\beta^{(3)} : \mathrm{dgm}_i^{(3)}(a) \cup \partial\Omega \to \mathrm{dgm}_i^{(3)}(m) \cup \partial\Omega.$$
We can use the two matchings $\beta^{(2)}$ and $\beta^{(3)}$ to define a new optimal matching $\tilde\gamma : \mathrm{dgm}_i(\mathsf{A}) \cup \partial\Omega \to \mathrm{dgm}_i(\mathsf{M}) \cup \partial\Omega$ as follows:

- The points in $\mathrm{dgm}_i^{(1)}(\mathsf{A})$ are matched by $\tilde\gamma$ to $\partial\Omega$ as with $\gamma$.

- Given $u = (u_1, u_2) \in \mathrm{dgm}_i^{(2)}(\mathsf{A})$, let $u' = (\tau(\mathsf{M})/2, u_2)$ be the point of $\mathrm{dgm}_i^{(2)}(a)$ with which $u$ is in bijection. We let $\tilde\gamma$ match $u$ with the point $v = (0, v_2) \in \mathrm{dgm}_i^{(2)}(\mathsf{M})$ which is in bijection with $\beta^{(2)}(u') = (\tau(\mathsf{M})/2, v_2) \in \mathrm{dgm}_i^{(2)}(m)$. Then $|u_2 - v_2| \leq \min(\delta, C\varepsilon^2)$ due to the optimality of $\beta$, and this defines a bijective matching $\tilde\gamma^{(2)} : \mathrm{dgm}_i^{(2)}(\mathsf{A}) \to \mathrm{dgm}_i^{(2)}(\mathsf{M})$. Note also that $\max_{u \in \mathrm{dgm}_i^{(2)}(\mathsf{A})} |u_1 - \tilde\gamma^{(2)}(u)_1| = \max_{u \in \mathrm{dgm}_i^{(2)}(\mathsf{A})} u_1 = \max_{u \in \mathrm{dgm}_i^{(2)}(\mathsf{A})} |u_1 - \gamma^{(2)}(u)_1|$, hence $\max_{u \in \mathrm{dgm}_i^{(2)}(\mathsf{A})} \|u - \tilde\gamma^{(2)}(u)\|_\infty \leq \delta$.

- We have seen that $\mathrm{dgm}_i^{(3)}(a) = \mathrm{dgm}_i^{(3)}(\mathsf{A})$ and $\mathrm{dgm}_i^{(3)}(m) = \mathrm{dgm}_i^{(3)}(\mathsf{M})$. We simply define the restriction and corestriction $\tilde\gamma^{(3)} : \mathrm{dgm}_i^{(3)}(\mathsf{A}) \cup \partial\Omega \to \mathrm{dgm}_i^{(3)}(\mathsf{M}) \cup \partial\Omega$ of $\tilde\gamma$ as being equal to $\beta^{(3)} : \mathrm{dgm}_i^{(3)}(a) \cup \partial\Omega \to \mathrm{dgm}_i^{(3)}(m) \cup \partial\Omega$. The optimality of $\beta$ implies that $\max_{u \in \mathrm{dgm}_i^{(3)}(\mathsf{A})} \|u - \tilde\gamma(u)\|_\infty \leq \min(\delta, C\varepsilon^2)$.

Thus the global matching $\tilde\gamma : \mathrm{dgm}_i(\mathsf{A}) \cup \partial\Omega \to \mathrm{dgm}_i(\mathsf{M}) \cup \partial\Omega$ is well-defined, is optimal for the bottleneck distance, and satisfies the conditions stated in the proposition. This completes the proof. $\qquad\square$

# E Proof of Theorem 3.3

We prove Theorem 3.3, which we restate for the reader's convenience:

**Theorem 3.3.** *Let* $\mathsf{M} \subset \mathbb{R}^d$ *be a generic compact submanifold and* $\mathsf{A} \subset \mathsf{M}$ *be a* $(\delta, \varepsilon)$-*dense set in* $\mathsf{M}$ *for some* $\varepsilon, \delta > 0$. *Let* $a \geq \varepsilon/\delta$ *and let* $i \geq 0$ *be an integer. There exist* $\varepsilon_0 > 0$ *depending only on* $\mathsf{M}$ *and* $C_0, C_1, C_2, C_3$ *depending only on* $\mathsf{M}$ *and* $a$ *such that if* $\varepsilon \leq \varepsilon_0$, *then* $\mathrm{dgm}_i^{(3)}(\mathsf{A})$ *has at most* $C_0$ *points and for all* $p \geq 1$, $\alpha \geq 0$,

$$
\begin{aligned}
\mathrm{OT}_p^p(\mathrm{dgm}_i(\mathsf{A}), \mathrm{dgm}_i(\mathsf{M})) &\leq C_1 \varepsilon^{p-m} \\
\mathrm{Pers}_\alpha(\mathrm{dgm}_i(\mathsf{A})) &\leq C_2(C_3^\alpha + \varepsilon^{\alpha-m}).
\end{aligned}
\tag{8}
$$

*Proof.* Let us first prove the bound on the cardinality of $\mathrm{dgm}_i^{(3)}(\mathsf{A})$. As $\mathsf{M}$ is generic, Theorem 1.6 from [6] states that if $\varepsilon$ is smaller than some $\varepsilon_0 = \varepsilon_0(\mathsf{M})$, then each point in $\mathrm{Crit}(\mathsf{A})$ at distance more than $\tau(\mathsf{M})/2$ from $\mathsf{A}$ must be at distance at most $K_1\varepsilon$ from one of the finitely many points of $\mathrm{Crit}(\mathsf{M})$ for some $K_1 = K_1(\mathsf{M})$. Corollary 1.7 from the same article then states that the number of points in $\mathrm{Crit}(\mathsf{A})$ at distance less than $K_1\varepsilon$ from a given point of $\mathrm{Crit}(\mathsf{M})$ is upper bounded by some constant that depends on $\mathsf{M}$ and the ratio $\varepsilon/\delta$, and is decreasing in this ratio; hence it is upper bounded by some constant that depends on $\mathsf{M}$ and $a$. The proof of this corollary also shows that the maximum number of projections on $\mathsf{A}$ of each of these points of $\mathrm{Crit}(\mathsf{A})$ is also upper bounded by some constant that depends on $\mathsf{M}$ and $a$. Hence there exist constants $K_2 = K_2(\mathsf{M}, a)$ and $K_3 = K_3(\mathsf{M}, a)$ such that if $\varepsilon \leq \varepsilon_0$, then there are at most $K_2$ points in $\mathrm{Crit}(\mathsf{A})$ at distance more than $\tau(\mathsf{M})/2$ from $\mathsf{A}$, and each has at most $K_3$ projections on $\mathsf{A}$.

Lemma E.1 below, applied to the interval $[\tau(\mathsf{M})/2, \infty)$ and the set $\mathsf{A}$, then states that the number of points in $\mathrm{dgm}_i(\mathsf{A})$ such that at least one of their coordinates is greater than $\tau(\mathsf{M})/2$ is bounded by $K_2(\binom{K_3}{i+1} + \binom{K_3}{i+2})$. Hence there are at most $C_0 := K_2 2^{K_3} \geq K_2(\binom{K_3}{i+1} + \binom{K_3}{i+2})$ points in $\mathrm{dgm}_i^{(3)}(\mathsf{A})$ when $\varepsilon \leq \varepsilon_0$.

Now let us prove the bounds on $\mathrm{OT}_p^p(\mathrm{dgm}_i(\mathsf{A}), \mathrm{dgm}_i(\mathsf{M}))$ and $\mathrm{Pers}_\alpha(\mathrm{dgm}_i(\mathsf{A}))$. As stated in Theorem 2.2, which applies as $\varepsilon_0 < \tau(\mathsf{M})/4$, each point $u = (u_1, u_2) \in \mathrm{dgm}_i^{(1)}(\mathsf{A})$ is such that its coordinates satisfy $0 \leq u_1, u_2 \leq \varepsilon + \varepsilon^2/\tau(\mathsf{M}) \leq 2\varepsilon$. In particular, they must correspond to the birth or the death of an interval of the Čech persistence module of $\mathsf{A}$ that occurs before filtration time $2\varepsilon$. The homology of the offsets $\mathsf{A}^t$ can be computed using the Čech simplicial complex of $\mathsf{A}$ (see e.g. [37]). In particular, each change in the homology of the offsets, hence each birth or death in the Čech persistence module of $\mathsf{A}$, is induced by the apparition of some simplex $\sigma$ at the corresponding filtration value in the Čech complex, and each such apparition causes at most a single death or birth. If a simplex $\sigma$ appears before filtration time $2\varepsilon$, it is by definition contained in a ball of radius $2\varepsilon$, hence it is of diameter at most $4\varepsilon$. Let us assume from now on that $\varepsilon_0 \leq \tau(\mathsf{M})/16$. Consider $x \in \mathsf{A}$; then [2, Proposition 8.7] states that the intersection $\overline{B}(x, 4\varepsilon) \cap \mathsf{A}$ contains at most $K_4(\varepsilon/\delta)^m \leq K_4 a^m$ points for some constant $K_4 = K_4(\mathsf{M})$. Hence $x$ belongs to at most $2^{K_4 a^m}$ simplices that appear before $\varepsilon$, and there are at most $\#\mathsf{A} \cdot 2^{K_4 a^m}$ such simplices. As the cardinality $\#\mathsf{A}$ can be bounded by $K_5/\delta^m$ for some $K_5 = K_5(\mathsf{M})$, we find that $\#(\mathrm{dgm}_i^{(1)}(\mathsf{A})) \leq K_6/\delta^m$ for some $K_6 = K_6(\mathsf{M}, a)$.

Furthermore, when $\mathsf{M}$ is generic, Proposition 3.2 states that its PD $\mathrm{dgm}_i(\mathsf{M})$ has a finite number of points. Let $\gamma$ be an optimal matching between $\mathrm{dgm}_i(\mathsf{A})$ and $\mathrm{dgm}_i(\mathsf{M})$ for the bottleneck distance that satisfies the conclusions of Theorem 2.2. We find that any point $u \in \mathrm{dgm}_i^{(1)}(\mathsf{A})$ is matched to a point of $\partial\Omega$ at distance at most $\varepsilon$ from $u$. Moreover, the number of points in $\mathrm{dgm}_i^{(2)}(\mathsf{A}) \cup \mathrm{dgm}_i^{(3)}(\mathsf{A})$ is bounded by some constant $K_7 = K_7(\mathsf{M}, a)$, and they are all matched to a point of $\mathrm{dgm}_i(\mathsf{M})$ or $\partial\Omega$ at distance at most $\varepsilon$. In particular, these finitely many points are at distance at most $K_8 = K_8(\mathsf{M})$ from $\partial\Omega$. Furthermore, this matching is surjective, in the sense that $\gamma$ matches all points of $\mathrm{dgm}_i(\mathsf{M})$ to a point of $\mathrm{dgm}_i(\mathsf{A})$. As a result, for any $p \geq 1$, we find that

$$
\begin{aligned}
\mathrm{OT}_p^p(\mathrm{dgm}_i(\mathsf{A}), \mathrm{dgm}_i(\mathsf{M})) &\leq \sum_{u \in \mathrm{dgm}_i^{(1)}(\mathsf{A})} \|u - \gamma(u)\|_\infty^p + \sum_{u \in \mathrm{dgm}_i^{(2)}(\mathsf{A}) \cup \mathrm{dgm}_i^{(3)}(\mathsf{A})} \|u - \gamma(u)\|_\infty^p \\
&\leq K_6 \delta^{-m} \varepsilon^p + K_7 \varepsilon^p \leq K_6 a^m \varepsilon^{p-m} + K_7 \varepsilon^p \leq C_1 \varepsilon^{p-m}
\end{aligned}
$$

for some $C_1 = C_1(\mathsf{M}, a)$. Likewise, for any $\alpha \geq 0$, we have that

$$\mathrm{Pers}_\alpha(\mathrm{dgm}_i(\mathsf{A})) = \sum_{u \in \mathrm{dgm}_i^{(1)}(\mathsf{A})} \mathrm{pers}(u)^\alpha + \sum_{u \in \mathrm{dgm}_i^{(2)}(\mathsf{A}) \cup \mathrm{dgm}_i^{(3)}(\mathsf{A})} \mathrm{pers}(u)^\alpha$$

$$\leq K_6 \delta^{-m} \varepsilon^\alpha + K_7 K_8^\alpha \leq K_6 a^m \varepsilon^{\alpha-m} + K_7 K_8^\alpha \leq C_2(C_3^\alpha + \varepsilon^{\alpha-m})$$

for some $C_2 = C_2(\mathsf{M}, a), C_3 = C_3(\mathsf{M})$. This completes the proof. $\qquad\square$

**Lemma E.1.** *Let $\mathsf{A} \subset \mathbb{R}^d$ be a finite set, and let $a < b \in \mathbb{R} \cup \{-\infty, +\infty\}$ and $i \geq 0$. Let*

$$\bigcup_{j=1}^L C_j = \mathrm{Crit}(\mathsf{A}) \cap d_\mathsf{A}^{-1}[a, b]$$

*be a covering of the set of critical points of $d_\mathsf{A}$ whose critical value belongs to $[a, b]$,[8] and let*

$$N_j := \# \left( \bigcup_{z \in C_j} \sigma_\mathsf{A}(z) \right)$$

*be the cardinality of the union of the projections of the critical points of $C_j$. Then the number of points in $\mathrm{dgm}_i(\mathsf{A})$ such that at least one of their coordinates belongs to $[a, b]$ is upper-bounded by $\sum_{j=1}^L \binom{N_j}{i+1} + \binom{N_j}{i+2}$, hence by $\sum_{j=1}^L N_j^{i+2}$.*

*Proof.* It is shown in [5] that the distance function to any finite point cloud is a topological Morse function. Hence its topological critical points are in bijection (via their critical values) with the non-zero coordinates of the points in (the union over all degrees of) the Čech persistence diagrams of $\mathsf{A}$. The number of points in $\mathrm{dgm}_i(\mathsf{A})$ such that at least one of their coordinates belongs to $[a, b]$ must then be upper bounded by the number of topological critical points $z$ of topological Morse index $i$ or $i+1$ such that $d_\mathsf{A}(z) \in [a, b]$. Moreover, it is also shown that the topological critical points of $d_\mathsf{A}$ are a subset of $\mathsf{A} \cup \mathrm{Crit}(\mathsf{A})$, though not all differential critical points need be topologically critical. Hence any such $z$ belongs to $C_j$ for some $j \in \{1, \ldots, L\}$, and $\sigma_\mathsf{A}(z) \subset \bigcup_{z' \in C_j} \sigma_\mathsf{A}(z')$.

As shown in [5], if $z$ is of critical index $i$, then the linear span $\mathrm{Span}(\sigma_\mathsf{A}(z) - z)$ is of dimension $i$. By Carathéodory's theorem, there exists $i+1$ affinely independent points in $\sigma_\mathsf{A}(z) \subset \bigcup_{z' \in C_j} \sigma_\mathsf{A}(z')$ such that $z$ belongs to their convex hull. Those $i+1$ points uniquely identify $z$ among the critical points of $\mathsf{A}$, as it is the only point equidistant to them that belongs to their convex hull. Hence there is an injection from the topological critical points of $\mathsf{A}$ of index $i$ that belong to $C_j$ into the set of subsets of cardinality $i+1$ of $\bigcup_{z' \in C_j} \sigma_\mathsf{A}(z')$. Applying the same reasoning to the points of index $i+1$, we find that there are at most $\sum_{j=1}^n \binom{N_j}{i+1} + \binom{N_j}{i+2}$ points in $\mathrm{dgm}_i(\mathsf{A})$ with at least one of their coordinates in $[a, b]$, as desired. $\qquad\square$

# F   The $\mathrm{OT}_p$ distance between Radon measures

Let $\mathcal{M}$ denote the space of Radon measures on $\Omega$, and let us define $\bar{\Omega} := \{u = (u_1, u_2) \in \mathbb{R}^2 : u_1 \leq u_2\}$. We call $\pi$ an admissible transport plan between $\nu_1, \nu_2 \in \mathcal{M}$ if it is a Radon measure on $\bar{\Omega} \times \bar{\Omega}$ such that for all Borel sets $A, B \subset \Omega$,

$$\pi(A \times \bar{\Omega}) = \nu_1(A) \quad \text{and} \quad \pi(\bar{\Omega} \times B) = \nu_2(B). \tag{17}$$

For $p \in [1, +\infty)$, we define

$$\mathrm{OT}_p^p(\nu_1, \nu_2) = \inf_{\pi \in \mathrm{Adm}(\nu_1, \nu_2)} \iint \|u - v\|_\infty^p \mathrm{d}\pi(u, v) \in \mathbb{R} \cup \{+\infty\}, \tag{18}$$

where $\|u\|_\infty = \max(|u_1|, |u_2|)$ and $\mathrm{Adm}(\nu_1, \nu_2)$ is the set of all admissible transport plans between $\nu_1$ and $\nu_2$. We also define

$$\mathrm{OT}_\infty(\nu_1, \nu_2) = \inf_{\pi \in \mathrm{Adm}(\nu_1, \nu_2)} \sup\{\|u - v\|_\infty : (u, v) \in \mathrm{Support}(\pi)\} \tag{19}$$

---

[8]When $a = -\infty$, we commit a minor abuse of notation by writing $[a, b]$ rather than $(a, b]$, and similarly when $b = +\infty$.

For all $p \in [1, \infty]$, the infimum is in fact a minimum (see [33]). We call $\mathrm{OT}_p(\nu_1, \nu_2)$ the $p$-Wasserstein distance between $\nu_1$ and $\nu_2$, though it differs from the usual Wasserstein distance, which is defined between measures with equal finite mass, while $\mathrm{OT}_p$ is defined between measures that can have different (and even infinite) masses. Intuitively, $\mathrm{OT}_p$ allows for some of the mass of $\nu_1$ and $\nu_2$ to be transported to the diagonal $\partial\Omega := \{(u, u) \in \mathbb{R}^2\}$, which acts as an infinitely deep landfill. The $p$-Wasserstein distance is a distance on the space $\mathcal{M}_p = \{\nu \in \mathcal{M} : \mathrm{OT}_p(\nu, 0) < \infty\}$, where $0$ denotes the null measure.

As explained in Section 4, a PD $a$ can be identified with the Radon measure $\sum_{u \in a} \delta_u$. Let $\mathcal{D}_p$ be the set of PDs being in $\mathcal{M}_p$. Then, Divol and Lacombe show in [33] that the $\mathrm{OT}_p$ distance defined in (18) coincides with the $\mathrm{OT}_p$ distance defined between PDs in Equation (3), and likewise for the case $p = \infty$.

We require the following lemma from [33]:

**Lemma F.1.** *A sequence of measures $(\nu_n)_{n \geq 1}$ converges with respect to $\mathrm{OT}_p$ to some measure $\nu$ if and only if the sequence $(\nu_n)_{n \geq 1}$ converges vaguely towards $\nu$ and $\mathrm{Pers}_p(\nu_n) \to \mathrm{Pers}_p(\nu)$ as $n \to \infty$.*

## G Proofs of Section 4

We start with the proof Theorem 4.1, which is split into a series of lemmas, before proving Proposition 4.2, Corollary 4.3 and Corollary 4.4.

Let us restate Theorem 4.1 for the reader's convenience:

**Theorem 4.1** (Law of large numbers)**.** *Assume that $P$ has a density $f$ on $\mathsf{M}$ bounded away from $0$ and $\infty$. Let $i \geq 0$ be an integer and let $1 \leq p < \infty$. Then $\mu_{f,i} \in \mathcal{M}_p$ and $\mathbb{E}[\mathrm{OT}_p^p(\mu_{n,i}, \mu_{f,i})] \xrightarrow[n\to\infty]{} 0$.*

*Furthermore, for all $\alpha > 0$, $\mathrm{Pers}_\alpha(\mathrm{dgm}_i^{(1)}(\mathsf{A}_n)) n^{\frac{\alpha}{m}-1} = \mathrm{Pers}_\alpha(\mu_{n,i}) = \mathrm{Pers}_\alpha(\mu_{f,i}) + o_{L^1}(1)$.*

Before proving Theorem 4.1, we state a simple lemma which allows us to control the mass of balls on $\mathsf{M}$.

**Lemma G.1.** *Let $\mathsf{M}$ be a compact submanifold with positive reach. Let $P$ be a probability measure having a density $f$ on $\mathsf{M}$ satisfying $f_{\min} \leq f \leq f_{\max}$ for two strictly positive constants $f_{\min}, f_{\max}$. There exist constants $c_m, C_m$ depending only on $m$ such that for all $0 \leq r \leq \tau(\mathsf{M})/4$*

$$c_m f_{\min} r^m \leq P(\overline{B}(x, r)) \leq C_m f_{\max} r^m. \tag{20}$$

*Let $\mathsf{A}_n$ be a sample of $n$ i.i.d. observations of law $P$. Then, there exists $C = C(\mathsf{M})$ depending on $\mathsf{M}$ such that for all $x \in \mathsf{M}$ and all $r > 0$,*

$$\mathbb{P}(d(x, \mathsf{A}_n) \geq r) \leq \exp(-nCf_{\min}r^m). \tag{21}$$

*Proof.* For the first statement, see [2, Proposition 31]. Let us prove the second one. Remark that the probability is zero for $r > \mathrm{diam}(\mathsf{M})$. Hence, we can assume that $r \leq \mathrm{diam}(\mathsf{M})$. When $r \leq \tau(\mathsf{M})/4$, it holds that

$$\mathbb{P}(d(x, \mathsf{A}_n) \geq r) = (1 - P(\overline{B}(x, r)))^n \leq \exp(-nc_m f_{\min} r^m).$$

When $\tau(\mathsf{M})/4 \leq r \leq \mathrm{diam}(\mathsf{M})$, we write

$$\mathbb{P}(d(x, \mathsf{A}_n) \geq r) \leq \mathbb{P}(d(x, \mathsf{A}_n) \geq \tau(\mathsf{M})/4) \leq \exp(-nc_m f_{\min}(\tau(\mathsf{M})/4)^m)$$

$$\leq \exp(-nc_m f_{\min} \frac{(\tau(\mathsf{M})/4)^m}{\mathrm{diam}(\mathsf{M})^m} r^m).$$

Hence, the result holds with $C = c_m \min\left(1, \frac{(\tau(\mathsf{M})/4)^m}{\mathrm{diam}(\mathsf{M})^m}\right)$. $\qquad\square$

*Proof of Theorem 4.1.* Recall that Goel, Trinh and Tsunoda [42] have shown that almost surely, the sequence of Radon measures $(\mu_{n,i})_n$ vaguely converges to the Radon measure $\mu_{f,i}$. We start by showing that, using this vague convergence and Lemma F.1, it is enough to prove the convergence of the $p$-total persistence.

**Lemma G.2.** *Let $(\nu_n)_{n \geq 1}$ be a sequence of random measures in $\mathcal{M}_p$ that converges vaguely almost surely to a Radon measure $\nu \in \mathcal{M}_p$, and such that $\mathbb{E}[|\mathrm{Pers}_p(\nu_n) - \mathrm{Pers}_p(\nu)|] \to_{n \to \infty} 0$. Then, $\mathbb{E}[\mathrm{OT}_p^p(\nu_n, \nu)] \to_{n \to \infty} 0$.*

*Proof.* Let us first show that $(D_n)_{n \geq 1} = (\mathrm{OT}_p^p(\nu_n, \nu))_{n \geq 1}$ converges in probability to 0. We use the following standard result: if for every subsequence $(Z_{n_k})_{k \geq 1}$ of $(Z_n)_{n \geq 1}$, one can extract a subsequence $(Z_{n_{k_l}})_{l \geq 1}$ that converges almost surely to 0, then the sequence $(Z_n)_{n \geq 1}$ converges in probability to 0. Let $(D_{n_k})_{k \geq 1}$ be a subsequence of $(D_n)_{n \geq 1} = (\mathrm{OT}_p^p(\nu_n, \nu))_{n \geq 1}$. Then, as $(\mathrm{Pers}_p(\nu_n))_{n \geq 1}$ converges in $L^1$ to $\mathrm{Pers}_p(\nu)$, it also converges in probability. In particular, there exists a subsequence $(n_{k_l})_{l \geq 1}$ such that $(\mathrm{Pers}_p(\nu_{n_{k_l}}))_{n \geq 1}$ converges almost surely to $\mathrm{Pers}_p(\nu)$. When restricting ourselves to this subsequence, we have both vague convergence of the measures and convergence of the $p$-total persistence. Hence, according to Lemma F.1, we have $D_{n_{k_l}} = \mathrm{OT}_p^p(\nu_{n_{k_l}}, \nu) \to_{l \to \infty} 0$ almost surely, proving that we actually have that $(D_n)_{n \geq 1}$ converges in probability to 0. To prove that $\mathbb{E}[D_n] \to_{n \to \infty} 0$, it remains to show that the sequence $(D_n)_{n \geq 1}$ is uniformly integrable. By considering the trivial transport plan that sends all probability mass to $\partial\Omega$, we have for all $n \geq 1$

$$D_n \leq \mathrm{Pers}_p(\nu_n) + \mathrm{Pers}_p(\nu).$$

But the sequence $(\mathrm{Pers}_p(\nu_n))_{n \geq 1}$ is uniformly integrable, as it converges in $L^1$. Hence, so is the sequence $(D_n)_{n \geq 1}$, concluding the proof. $\square$

Using Lemma G.2, Theorem 4.1 would follow from the facts that $\mu_{f,i} \in \mathcal{M}_p$ and that $\mathbb{E}[|\mathrm{Pers}_p(\mu_{n,i}) - \mathrm{Pers}_p(\mu_{f,i})|]$ converges to 0.

Recall that $C_c(\Omega)$ is the set of continuous functions $f : \Omega \to \mathbb{R}$ with compact support (i.e. the support is bounded and at positive distance from $\partial\Omega$). For $s \geq 0$, let $T_s = \{(u_1, u_2) \in \Omega : u_2 \geq s\}$.

**Lemma G.3.** *Let $\alpha > 0$. Let $(\nu_n)_{n \geq 1}$ be a sequence of random measures in $\mathcal{M}_\alpha$ that converges vaguely almost surely to a Radon measure $\nu \in \mathcal{M}$. Assume that the sequence of random variables $(\nu_n(\Omega))_{n \geq 1}$ is uniformly integrable and that*

$$\sup_n \mathbb{E}[\mathrm{Pers}_\alpha(\nu_n)] < +\infty \text{ and } \lim_{s \to +\infty} \limsup_n \mathbb{E}\Big[\int_{T_s} \mathrm{pers}^\alpha(u) \mathrm{d}\nu_n(u)\Big] = 0.$$

*Then, $\nu \in \mathcal{M}_\alpha$ and $\mathbb{E}[|\mathrm{Pers}_\alpha(\nu_n) - \mathrm{Pers}_\alpha(\nu)|] \to_{n \to \infty} 0$.*

*Proof.* We divide the proof into several steps.

1. Let $\phi \in C_c(\Omega)$. We first show that $(\int \phi \mathrm{d}\nu_n)_{n \geq 1}$ converges in $L^1$ to $\int \phi \mathrm{d}\nu$. By assumption, the convergence holds almost surely. Furthermore, as $\phi$ is bounded and as the sequence $(\nu_n(\Omega))_{n \geq 1}$ is uniformly integrable, so is the sequence $(\int \phi \mathrm{d}\nu_n)_{n \geq 1}$. Hence, $\mathbb{E}[|\int \phi \mathrm{d}(\nu_n - \nu)|] \to_{n \to \infty} 0$.

2. Let $(\phi_k)_{k \geq 1}$ be an increasing sequence of functions in $C_c(\Omega)$ that converge pointwise to the function $\mathrm{pers}_\alpha$. Then, almost surely,

$$\int \phi_k \mathrm{d}\nu \leq \liminf_{n \to \infty} \int \phi_k \mathrm{d}\nu_n \leq \liminf_{n \to \infty} \int \mathrm{pers}_\alpha \mathrm{d}\nu_n.$$

   By Fatou's lemma, $\mathbb{E}[\liminf_{n \to \infty} \int \mathrm{pers}_\alpha \mathrm{d}\nu_n] \leq \liminf_{n \to \infty} \mathbb{E}[\mathrm{Pers}_\alpha(\nu_n)] = C < +\infty$ by assumption. Hence, by letting $k \to \infty$ and applying the monotone convergence theorem, we obtain $\mathrm{Pers}_\alpha(\nu) \leq C$, proving that $\nu \in \mathcal{M}_\alpha$.

3. The same argument can be applied to the constant function equal to 1, showing that $\nu(\Omega) < +\infty$.

4. Let $s \geq 1$. The function $\mathrm{pers}_\alpha$ can be decomposed into a sum of three positive continuous functions $\mathrm{pers}_\alpha = \phi_s^{(1)} + \phi_s^{(2)} + \phi_s^{(3)}$, where $\phi_s^{(1)}$ has compact support, the support of $\phi_s^{(2)}$

is included in the band $\{u \in \Omega : \text{pers}(u) \leq 1/s\}$ and the support of $\phi_s^{(3)}$ is included in $T_s$. Hence,

$$\limsup_{n \to +\infty} \mathbb{E}[|\text{Pers}_\alpha(\nu_n) - \text{Pers}_\alpha(\nu)|] \leq \limsup_{n \to +\infty} \mathbb{E}[|\int \phi_s^{(1)} \mathrm{d}(\nu_n - \nu)|]$$

$$+ \limsup_{n \to +\infty} \mathbb{E}[|\int \phi_s^{(2)} \mathrm{d}(\nu_n - \nu)|] + \limsup_{n \to +\infty} \mathbb{E}[|\int \phi_s^{(3)} \mathrm{d}(\nu_n - \nu)|].$$

The first term in the above sum is equal to zero because of the first item, the second one is smaller than $s^{-\alpha}(\sup_n \mathbb{E}[\nu_n(\Omega)] + \nu(\Omega))$, and the third one is smaller than

$$\limsup_n \mathbb{E}[\int_{T_s} \text{pers}_\alpha(u)\mathrm{d}\nu_n(u)] + \int_{T_s} \text{pers}_\alpha(u)\mathrm{d}\nu(u).$$

Using the hypotheses of the lemma, the second and the third term converges to $0$ as $s$ goes to $\infty$. We obtain that $\limsup_{n \to +\infty} \mathbb{E}[|\text{Pers}_\alpha(\nu_n) - \text{Pers}_\alpha(\nu)|] = 0$. $\qquad\square$

Our goal is to show that the conditions of Lemma G.3 holds for the sequence $(\mu_{n,i})_{n \geq 1}$ to conclude. Remark that for any Radon measure $\nu \in \mathcal{M}_\alpha$ and $s \geq 0$

$$\int_{T_s} \text{pers}^\alpha(u)\mathrm{d}\nu(u) = \alpha \int_0^\infty t^{\alpha-1}\nu(T_s \cap \{u : \text{pers}(u) \geq t\})\mathrm{d}t$$

$$\leq \alpha \int_{s/2}^\infty t^{\alpha-1}\nu(T_{2t})\mathrm{d}t + \alpha \int_0^{s/2} t^{\alpha-1}\nu(T_s)\mathrm{d}t \qquad (22)$$

$$\leq \alpha \int_{s/2}^\infty t^{\alpha-1}\nu(T_{2t})\mathrm{d}t + (s/2)^\alpha\nu(T_s),$$

where we use Fubini's theorem for the first equality and the fact that $\{u : \text{pers}(u) \geq t\} \subset T_{2t}$ for the first inequality. We also have $\nu(\Omega) = \nu(T_0)$. Hence, the different conditions of Lemma G.3 can all be obtained by controlling the random variable $\mu_{n,i}(T_s)$ for $s \geq 0$.

**Proposition G.4.** *Let* $\mathsf{M}$ *be a compact submanifold with positive reach. Assume that* $P$ *has a density* $f$ *on* $\mathsf{M}$ *satisfying* $f_{\min} \leq f \leq f_{\max}$ *for two positive constants* $f_{\min}, f_{\max}$. *Then there exist* $c, C > 0$ *that depend on* $\mathsf{M}$, $i$, $f_{\min}$ *and* $f_{\max}$ *such that for all integer* $n \geq 1$ *and all* $s \geq 0$,

$$\mathbb{E}[\mu_{n,i}(T_s)^2] \leq C \exp(-cs^m). \qquad (23)$$

Before proving Proposition G.4, let us show how to use it to conclude the proof of Theorem 4.1. First, it implies that the random variables $\mu_{n,i}(\Omega) = \mu_{n,i}(T_0)$ for $n \geq 1$ have a uniformly bounded second moment, and are therefore uniformly integrable. Second, we have $\mathbb{E}[\mu_{n,i}(T_s)] \leq \mathbb{E}[\mu_{n,i}(T_s)^2]^{1/2}$ using Hölder's inequality. Hence, (22) implies that for any $\alpha > 0$, we have

$$\mathbb{E}[\text{Pers}_\alpha(\mu_{n,i})] \leq \alpha \int_0^\infty t^{\alpha-1}\mathbb{E}[\mu_{n,i}(T_{2t})]\mathrm{d}t$$

$$\leq \alpha\sqrt{C} \int_0^\infty t^{\alpha-1}e^{-c2^{m-1}t^m}\mathrm{d}t.$$

In particular, $\sup_n \mathbb{E}[\text{Pers}_\alpha(\mu_{n,i})] < +\infty$. Likewise,

$$\sup_n \mathbb{E}\left[\int_{T_s} \text{pers}_\alpha(u)\mathrm{d}\mu_{n,i}(u)\right] \leq \alpha \int_{s/2}^\infty t^{\alpha-1}\mathbb{E}[\mu_{n,i}(T_{2t})]\mathrm{d}t + (s/2)^\alpha\mathbb{E}[\mu_{n,i}(T_s)]$$

$$\leq \alpha\sqrt{C} \int_0^\infty t^{\alpha-1}e^{-c2^{m-1}t^m}\mathrm{d}t + (s/2)^\alpha\sqrt{C}\exp(-c/2s^m)$$

so that $\lim_{s \to +\infty} \sup_n \mathbb{E}[\int_{T_s} \text{pers}_\alpha(u)\mathrm{d}\mu_{n,i}(u)] = 0$. We are therefore in position to apply Lemma G.3, proving the convergence of the $\alpha$-total persistence. Together with Lemma G.2 with $\alpha = p \geq 1$, we also obtain the $\text{OT}_p$-convergence of $\mu_{n,i}$. It remains to prove Proposition G.4.

*Proof of Proposition G.4.* Write $\varepsilon_n = d_H(\mathsf{A}_n, \mathsf{M})$. Let $s \geq 0$ and let $\varepsilon_0 < \tau(\mathsf{M})/2$ be a small parameter, to be fixed later. Recall that we write $\#S$ for the cardinality of a multiset $S$. Notice that $\mu_{n,i}(T_s) = 0$ if $sn^{-1/m} > \varepsilon_n + \varepsilon_n^2/\tau(\mathsf{M})$. In particular, we may assume without loss of generality that $s \leq n^{1/m}\mathrm{diam}(\mathsf{M})(1 + \mathrm{diam}(\mathsf{M})/\tau(\mathsf{M})) = n^{1/m}\varepsilon_{\max}$, for otherwise there is nothing to prove. Consider the event $E = \{\varepsilon_n + \varepsilon_n^2/\tau(\mathsf{M}) < \varepsilon_0\}$. By definition of Region (1), if $E$ is satisfied, then all the coordinates of points of the PD $\mathrm{dgm}_i^{(1)}(\mathsf{A}_n)$ are smaller than $\varepsilon_0$. Notice that the cardinality of $\mathrm{dgm}_i(\mathsf{A}_n)$ is smaller than the number of $i$-dimensional simplices in the Čech complex of $\mathsf{A}_n$, which is itself smaller than $n^{i+1}$ (as each simplex corresponds uniquely to a choice of $i + 1$ vertices of $\mathsf{A}_n$). Hence

$$\mathbb{E}[\mu_{n,i}(T_s)^2 \mathbf{1}\{E^c\}] \leq n^{-2}n^{2i+2}\mathbb{P}(\varepsilon_n + \varepsilon_n^2/\tau(\mathsf{M}) \geq \varepsilon_0).$$

We require the following lemma, which bounds the upper tail of the random variable $\varepsilon_n = d_H(\mathsf{A}_n, \mathsf{M})$.

**Lemma G.5.** *If $r \leq \tau(\mathsf{M})/2$, then*

$$\mathbb{P}(d_H(\mathsf{A}_n, \mathsf{M}) > r) \leq \frac{C_m}{f_{\min}r^m} \exp(-nc_m f_{\min} r^m) \tag{24}$$

*for two positive constants $c_m, C_m$ depending only on $m$. In particular, for any $q \geq 1$, $d_H(\mathsf{A}_n, \mathsf{M}) = O_{L^q}((\ln n/n)^{1/m})$.*

*Proof.* The bound $\mathbb{P}(d_H(\mathsf{A}_n, \mathsf{M}) > r)$ is given in [1, Lemma III.23]. Furthermore, [1, Lemma III.23] also states that for any $q > 0$, there exists $C_q$ depending on $f_{\min}$ and $m$ such that, with probability at least $1 - n^{-q/m}$, $d_H(\mathsf{A}_n, \mathsf{M}) \leq C_q \left(\frac{\ln n}{n}\right)^{1/m}$. In particular, we obtain that

$$\mathbb{E}[d_H(\mathsf{A}_n, \mathsf{M})^q] \leq C_q^q \left(\frac{\ln n}{n}\right)^{q/m} + \mathrm{diam}(\mathsf{M})^q n^{-q/m},$$

proving the second claim of the lemma. $\qquad\square$

Note that $\varepsilon_n \leq \mathrm{diam}(\mathsf{M})$, so $\mathbb{P}(\varepsilon_n + \varepsilon_n^2/\tau(\mathsf{M}) \geq \varepsilon_0) \leq \mathbb{P}(\varepsilon_n \geq c_0\varepsilon_0)$, with $c_0 = (1 + \mathrm{diam}(\mathsf{M})/\tau(\mathsf{M}))^{-1}$. Apply Lemma G.5 with $r = c_0\varepsilon_0$ to obtain that

$$\mathbb{E}[\mu_{n,i}(T_s)^2 \mathbf{1}\{E^c\}] \leq n^{2i}\frac{C_m}{f_{\min}r^m} \exp(-nc_m f_{\min} r^m) \leq C_0 \exp(-c_1 n)$$

for some positive constants $c_1, C_0$. Furthermore, recall that $s \leq n^{1/m}\varepsilon_{\max}$. Hence,

$$\mathbb{E}[\mu_{n,i}(T_s)^2 \mathbf{1}\{E^c\}] \leq C_0 \exp(-c_1\varepsilon_{\max}^{-m}s^m) = C_0 \exp(-c_2 s^m)$$

for some positive constant $c_2$.

It remains to bound $\mathbb{E}[\mu_{n,i}(T_s)^2 \mathbf{1}\{E\}]$. For each $j = 1, \ldots, n$, consider the set $\Xi_{n,s}^j$ of critical points $z \in \mathrm{Crit}(\mathsf{A}_n)$ such that $\sigma_{\mathsf{A}_n}(z)$ contains the point $X_i$ and $sn^{-1/m} \leq d_{\mathsf{A}_n}(z) \leq \varepsilon_n + \varepsilon_n^2/\tau(\mathsf{M})$.

**Lemma G.6.** *It holds that $\mu_{n,i}(T_s)$ is smaller than*

$$\frac{1}{n}\sum_{j=1}^{n} L_j^{i+2}, \tag{25}$$

*where $L_j$ is the cardinality of the set $\bigcup_{z \in \Xi_{n,s}^j} \sigma_{\mathsf{A}_n}(z)$.*

*Proof.* Lemma E.1 applied to a realization of $\mathsf{A}_n$ and the interval $[sn^{-1/m}, \varepsilon_n + \varepsilon_n^2/\tau(\mathsf{M})]$ yields that the number of points in $\mathrm{dgm}_i(\mathsf{A}_n)$ with at least one of their coordinates in $[sn^{-1/m}, \varepsilon_n + \varepsilon_n^2/\tau(\mathsf{M})]$ is upper bounded by $\sum_{j=1}^{n} L_j^{i+2}$. By definition, this means that $\mu_{n,i}(T_s) \leq \frac{1}{n}\sum_{j=1}^{n} L_j^{i+2}$, as desired. $\qquad\square$

One can show that the set $\Xi_{n,s}^j$ is localized, in the sense that it is included in a ball centered at $X_j$, with a radius depending on the sample $A_n$, that is small with high probability. Hence, the number $L_j$ is controlled by the number of points in $A_n$ found in a small (random) neighborhood of $X_j$. Let us make this idea rigorous.

For $r \geq 0$, define the shape

$$\mathcal{C}(r) = \{y = (y_m, y_{d-m}) \in \mathbb{R}^m \times \mathbb{R}^{d-m} : \|y\| \leq r, \|y_{d-m}\| \leq \frac{\|y\|^2}{2\tau(\mathsf{M})}\}. \tag{26}$$

We build a partition of $\mathcal{C}(r)$ in the following way. Consider a finite partition $\mathcal{W}$ of the unit sphere in $\mathbb{R}^m \times \{0\}^{d-m}$ into sets of diameters smaller than $\theta = \pi/4$ (for the geodesic distance on the sphere), which we fix for a given dimension $m$. Let $\mathcal{W}(r)$ be the partition of $\mathcal{C}(r)$ consisting of the sets

$$\{y = (y_m, y_{d-m}) \in \mathcal{C}(r) : y_m/\|y_m\| \in W\},$$

where $W$ is an element of the partition $\mathcal{W}$.

For $x \in \mathsf{M}$, consider an isometry $\iota_x : \mathbb{R}^d \to \mathbb{R}^d$ sending $\mathbb{R}^m \times \{0\}^{d-m}$ to $T_x\mathsf{M}$. Let $\mathcal{C}(x, r) = x + \iota_x(\mathcal{C}(r))$. Likewise, we define a partition $\mathcal{W}(x, r)$ by applying the affine transformation $W \mapsto x + \iota_x(W)$ to each $W \in \mathcal{W}(r)$.

For $j = 1, \ldots, n$, let $R_{jn}$ be the smallest radius $r \leq \varepsilon_0$ such that every $W \in \mathcal{W}(X_j, r)$ contains a point of $A_n$ other than $X_j$. By convention, we let $R_{jn} = \varepsilon_0$ if such a radius does not exist. $R_{jn}$ can be made measurable with a good choice of $x \mapsto \iota_x$; we assume it to be the case henceforth.

**Lemma G.7.** *Let $\varepsilon_0 \leq \tau(\mathsf{M})/\sqrt{2}$. For all $j = 1, \ldots, n$, if $z \in \mathrm{Crit}(A_n)$ is such that $X_j \in \sigma_{A_n}(z)$ and $d_{A_n}(z) \leq \varepsilon_0$, then $\|X_j - z\| \leq c_0 R_{jn}$ for some positive absolute constant $c_0$.*

*Proof.* Recall that for $x \in \mathsf{M}$, $\pi_x$ is the orthogonal projection on $T_x\mathsf{M}$ while $\pi_x^\perp$ is the orthogonal projection on the normal space at $x$. Let $j = 1, \ldots, n$ and let $z \in \mathrm{Crit}(A_n)$ be such that $X_j \in \sigma_{A_n}(z)$. The direction $e = \pi_{X_j}(z - X_j)/\|\pi_{X_j}(z - X_j)\|$ belongs to the unit sphere in $T_{X_j}\mathsf{M}$; note that $\pi_{X_j}(z - X_j) \neq 0$ due to Lemma G.8 below, which applies as $d_{A_n}(z) \leq \varepsilon_0 \leq \tau(\mathsf{M})/\sqrt{2}$. Hence, $\iota_x^{-1}(e)$ belongs to an element $W_0$ of the partition $\mathcal{W}$. Consider the corresponding element $W$ of the partition $\mathcal{W}(X_j, R_{jn})$.

If $R_{jn} = \varepsilon_0$, then the conclusion of the lemma holds (for $c_0 = 1$): indeed we have $\|X_j - z\| = d_{A_n}(z) \leq \varepsilon_0 \leq R_{jn}$. Otherwise, by assumption, there exists a point $X_k \in W$ for some $k \neq j$. As $X_j \in \sigma_{A_n}(z)$, it holds that $\|X_j - z\| \leq \|X_k - z\|$. Hence,

$$\|X_j - z\|^2 \leq \|X_k - z\|^2 = \|X_j - z\|^2 + \|X_j - X_k\|^2 + 2\langle X_k - X_j, X_j - z\rangle$$

and

$$\langle X_k - X_j, z - X_j\rangle \leq \frac{\|X_j - X_k\|^2}{2}. \tag{27}$$

We write

$$\langle X_k - X_j, z - X_j\rangle = \langle \pi_{X_j}(X_k - X_j), \pi_{X_j}(z - X_j)\rangle + \langle \pi_{X_j}^\perp(X_k - X_j), \pi_{X_j}^\perp(z - X_j)\rangle$$

By construction, as the diameter of $W_0$ is less than $\theta$, we have $\langle \pi_{X_j}(X_k - X_j), \pi_{X_j}(z - X_j)\rangle \geq \cos(\pi/4)\|\pi_{X_j}(X_k - X_j)\|\|\pi_{X_j}(z - X_j)\|$. On the other hand, according to [39, Theorem 4.18],

$$\|\pi_{X_j}^\perp(X_k - X_j)\| \leq \frac{\|X_k - X_j\|^2}{2\tau(\mathsf{M})}, \tag{28}$$

which also implies that $\|\pi_{X_j}(X_k - X_j)\| \geq \|X_j - X_k\|\sqrt{1 - \frac{\varepsilon_0^2}{4\tau(\mathsf{M})^2}}$. Similarly, Lemma G.8 below states that $\|\pi_{X_j}(z - X_j)\| \geq \|z - X_j\|/\sqrt{2}$ and $\|\pi_{X_j}^\perp(z - X_j)\| \leq \|z - X_j\|^2/\tau(\mathsf{M})$ (using our assumption that $d_{A_n}(z) \leq \varepsilon_0 \leq \tau(\mathsf{M})/\sqrt{2}$).

Hence we obtain that

$$\cos(\pi/4)\|X_j - X_k\|\|z - X_j\|\frac{\sqrt{1 - \frac{\varepsilon_0^2}{4\tau(\mathsf{M})^2}}}{\sqrt{2}} \leq \cos(\pi/4)\|\pi_{X_j}(X_k - X_j)\|\|\pi_{X_j}(z - X_j)\|$$

$$\leq \frac{\|X_k - X_j\|^2}{2} + \frac{\|X_k - X_j\|^2\|z - X_j\|^2}{2\tau(\mathsf{M})^2}$$

$$\leq \|X_k - X_j\|^2\left(\frac{1}{2} + \frac{\varepsilon_0^2}{2\tau(\mathsf{M})^2}\right).$$

Dividing by $\|X_j - X_k\|$ and using that $\|X_j - X_k\| \leq R_{jn}$ and that $\varepsilon_0 \leq \tau(\mathsf{M})/\sqrt{2}$, we see that $\|z - X_j\|$ is smaller than $R_{jn}$ up to an absolute multiplicative constant. $\qquad\square$

We now prove the lemma used above:

**Lemma G.8.** *Let* $\mathsf{A} \subset \mathsf{M}$, $z \in \mathrm{Crit}(\mathsf{A})$ *and* $x \in \sigma_{\mathsf{A}}(z)$. *Then* $\|\pi_x^\perp(z - x)\| \leq \|z - x\|^2/\tau(\mathsf{M})$. *Furthermore, if* $d_{\mathsf{A}}(z) \leq \tau(M)/\sqrt{2}$, *then* $\|\pi_x(z - x)\| \geq \|z - x\|/\sqrt{2}$.

*Proof.* The point $z$ can be written as a convex combination $z = \sum_k \lambda_k y_k$ where the points $y_k$ are in $\mathsf{A} \subset \mathsf{M}$. Then, using [39, Theorem 4.18],

$$\|\pi_x^\perp(z - x)\| \leq \sum_k \lambda_k \|\pi_x^\perp(y_k - x)\| \leq \sum_k \lambda_k \frac{\|y_k - x\|^2}{2\tau(\mathsf{M})}$$

$$= \sum_k \lambda_k \frac{\|y_k - z\|^2 + \|z - x\|^2 + 2\langle y_k - z, z - x\rangle}{2\tau(\mathsf{M})} = \frac{\|z - x\|^2}{\tau(\mathsf{M})},$$

as $\|y_k - z\|^2 = \|x - z\|^2$ and $\sum_k \lambda_k(y_k - z) = 0$. The second inequality $\|\pi_x(z - x)\| \geq \|z - x\|/\sqrt{2}$ from the statement follows from the first one through a direct computation. $\qquad\square$

Let us now show that the random variable $R_{jn}$ has controlled tails.

**Lemma G.9.** *For all* $x \in \mathsf{M}$, $r \geq 0$, $B(x, r) \cap \mathsf{M} \subset \mathcal{C}(x, r)$.

*Proof.* Let $y \in \mathsf{M}$ be such that $\|\pi_x(y - x)\| \leq r$. Then, according to [39, Theorem 4.18], $\|\pi_x^\perp(y - x)\| \leq \frac{\|y - x\|^2}{2\tau(\mathsf{M})}$. In particular, $B(x, r) \cap \mathsf{M} \subset \mathcal{C}(x, r)$. $\qquad\square$

**Lemma G.10.** *For* $0 < t \leq \tau(\mathsf{M})/4$ *and* $j = 1, \ldots, n$, *we have* $\mathbb{P}(R_{jn} > t) \leq C_m e^{-c_m f_{\min}(n-1)t^m}$ *for some positive constants* $c_m$, $C_m$.

*Proof.* If $R_{jn}$ is larger than $t$, then there exists at least one set $W \in \mathcal{W}(X_j, t)$ such that its intersection with $\mathsf{A}_n$ contains only $X_j$. Hence,

$$\mathbb{P}(R_{jn} > t|X_j) \leq \sum_{W \in \mathcal{W}(X_j, t)} \mathbb{P}(\mathsf{A}_n \cap W = X_j|X_j) = \sum_{W \in \mathcal{W}(X_j, t)} (1 - P(W))^{n-1}.$$

Let $\pi_0$ be the orthogonal projection from $\mathbb{R}^d$ to $\mathbb{R}^m \times \{0\}^{d-m}$. The image of a set $W \in \mathcal{W}(X_j, t)$ by the projection $y \mapsto \pi_{X_j}(y - X_j)$ is equal to $\iota_{X_j}(\pi_0(W_0)) \subset T_{X_j}\mathsf{M}$ for some $W_0 \in \mathcal{W}(t)$. For $t \leq \tau(\mathsf{M})/4$, the orthogonal projection $y \in B(X_j, t) \cap \mathsf{M} \mapsto \pi_{X_j}(y - X_j) \in T_{X_j}\mathsf{M}$ is a diffeomorphism on its image, with Jacobian lower bounded by a constant $c_m$ that depends only on $m$, see e.g. [31, Lemma 2.2]. According to Lemma G.9, the preimage of $\iota_{X_j}(\pi_0(W_0)) \subset T_{X_j}\mathsf{M}$ by this diffeomorphism is equal to $W \cap M$. Hence, by a change of variable,

$$P(W) = P(W \cap M) \geq f_{\min}c_m\mathrm{Vol}_m(W_0) \geq f_{\min}c'_m t^m$$

for some $c'_m > 0$. Hence,

$$\mathbb{P}(R_{jn} > t|X_j) \leq \#\mathcal{W}e^{-f_{\min}c'_m(n-1)t^m}.$$

We conclude by taking the expectation. $\qquad\square$

Let us now control the number of points found in a ball $B(X_j, \kappa R_{jn})$ for some $\kappa \geq 0$. Let $\rho_0 > 0$ be small enough such that $\int_{B(x,\rho_0) \cap \mathsf{M}} f < 1/2$ for any $x \in \mathsf{M}$.

**Lemma G.11.** *Let $l \geq 0, \kappa > 0$ be such that $\kappa \varepsilon_0 \leq \min(\rho_0, \tau(\mathsf{M})/4)$. For $j = 1, \ldots, n$, let $K_{jn}(\kappa)$ be the number of elements of $\mathsf{A}_n$ found in $B(X_j, \kappa R_{jn})$. Then, $\mathbb{E}[K_{jn}(\kappa)^l] \leq C_{m,l}(1 + \left(\frac{f_{\max}}{f_{\min}}\right)^l \kappa^{lm})$ for some constant $C_{m,l}$ which depends on $m$ and $l$.*

*Proof.* Let us write $\{W_1, \ldots, W_K\} = \mathcal{W}(X_j, \kappa R_{jn})$. Without loss of generality, we can assume that $n \geq K + 1$, as otherwise the bound is trivial. As in [34, Lemma 5], we remark that there is at least one sample point in every $W_i$, and that there is (almost surely) one single element $W_{i^*}$ of the partition with exactly one sample point on its boundary. Let $N_i$ be the cardinality of $(W_i \cap \mathsf{A}_n) \backslash \{X_j\}$, and $N_{-1}$ be the cardinality of $\mathsf{A}_n \backslash B(X_j, \kappa R_{jn})$. Define

$$\tilde{\alpha}_i := \int_{W_i} f$$

and $\alpha_i := \frac{\tilde{\alpha}_i}{1 - \tilde{\alpha}_{i^*}}$ for all $i \neq i^*$, as well as $\alpha_{-1} = \frac{1 - \sum_{i=1}^{K} \tilde{\alpha}_i}{1 - \tilde{\alpha}_{i^*}}$. Note that as $\kappa R_{jn} \leq \rho_0$, we have $\tilde{\alpha}_i < 1/2$ for all $i = 1, \ldots, K$. As $\kappa R_{jn} \leq \kappa \varepsilon_0 \leq \tau(\mathsf{M})/4$, we may use once again that the orthogonal projection $y \in B(X_j, \kappa R_{jn}) \cap \mathsf{M} \mapsto \pi_{X_j}(y - X_j) \in T_{X_j}\mathsf{M}$ is a diffeomorphism on its image, with Jacobian upper and lower bounded by constants depending only on $m$ (see [31, Lemma 2.2]) to also obtain that

$$cf_{\min}(\kappa R_{jn})^m \leq \tilde{\alpha}_i \leq Cf_{\max}(\kappa R_{jn})^m$$

for all $i = 1, \ldots, K$ and some constants $c = c(m), C = C(m)$. Hence there exists $C' = C'(m) > 0$ such that

$$cf_{\min}(\kappa R_{jn})^m \leq \alpha_i \leq C'f_{\max}(\kappa R_{jn})^m$$

for all $i = 1, \ldots, K$. Consider a multinomial random variable $L = (L_1, \ldots, \widehat{L_{i^*}}, \ldots, L_K, L_{-1})$ of parameters $n - 2$ and $(\alpha_1, \ldots, \widehat{\alpha_{i^*}}, \ldots, \alpha_K, \alpha_{-1})$, and let $E$ denote the event

$$\{L_i \geq 1 \; \forall i \in \{1, \ldots, K\} \backslash \{i^*\}\}.$$

Then conditionally on $X_j$, $R_{jn}$ and $i^*$, the variable $N = (N_1, \ldots, \widehat{N_{i^*}}, \ldots, N_K, N_{-1})$ follows the same distribution as $L \mid E$. Thus, conditionally on $X_j$, $R_{jn}$ and $W_{i^*}$, the variable $K_{jn}(\kappa) = 1 + \sum_{i=1}^{K} N_i = 2 + \sum_{i=1, i \neq i^*}^{K} N_i$ (where the initial 1 comes from $X_j$) has the same distribution as

$$2 + \sum_{i=1, i \neq i^*}^{K} L_i \mid E.$$

Note that as $\kappa R_{jn} \leq \rho_0$, we have $\alpha_{-1} \geq 1/2$.

As a result, Lemma G.12 below yields that

$$\mathbb{E}[K_{jn}(\kappa)^l | X_j, R_{jn}, W_{i^*}] = \mathbb{E}[(2 + \sum_{i=1, i \neq i^*}^{K} L_i)^l \mid E] \leq 2^l(2^l + \mathbb{E}[(\sum_{i=1, i \neq i^*}^{K} L_i)^l \mid E])$$

$$\leq 2^l(2^l + C_{K,l}(1 + (n - 2)\sum_{i=1, i \neq i^*}^{K} \alpha_i)^l)$$

$$\leq C'_{K,l}(1 + (n\sum_{i=1, i \neq i^*}^{K} \alpha_i)^l) \leq C'_{K,l}(1 + (nKC'f_{\max}(\kappa R_{jn})^m)^l)$$

$$\leq C_{m,l}(1 + \kappa^{ml} f_{\max}^l n^l R_{jn}^{ml})$$

where $C_{K,l}, C'_{K,l}$ and $C_{m,l}$ are constants whose dependencies are indicated by their indices (remember that $K = \#\mathcal{W}$ depends only on $m$). We can now conclude by considering

$$\mathbb{E}[K_{jn}(\kappa)^l] = \mathbb{E}_{X_j, R_{jn}, W_{i^*}}[\mathbb{E}[K_{jn}(\kappa)^l | X_j, R_{jn}, W_{i^*}]] \leq \mathbb{E}_{R_{jn}}[C_{m,l}(1 + \kappa^{ml} f_{\max}^l n^l R_{jn}^{ml})]$$

The quantity $n^l \mathbb{E}[R_{jn}^{ml}]$ is bounded by a constant, which is proved by integrating the tail bound found in Lemma G.10. Indeed, Lemma G.10 implies that the random variable $nR_{jn}^m$ is subexponential with a subexponential norm $m$ independent of $n$, of order $O(1/f_{\min})$; the moment of order $l$ of such a random variable is bounded by $C_l m^l$, see [74, Section 2.7]. $\qquad\square$

Let us now prove the technical lemma used above.

**Lemma G.12.** *Let $K \geq 1$ and $L = (L_1, \ldots, L_K, L_{K+1})$ be a random multinomial variable of parameters $n$ and $\alpha_1, \ldots, \alpha_K, \alpha_{K+1}$. Then there exists $C = C(K, l)$ such that*

$$\mathbb{E}[(\sum_{i=1}^{K} L_i)^l | L_i \geq 1 \ \forall i = 1, \ldots, K] \leq C_l (1 + (n \sum_{i=1}^{K} \alpha_i)^l).$$

*Proof.* Let $X_1, \ldots, X_n$ be i.i.d. categorical variables of parameters $\alpha_1, \ldots, \alpha_K, \alpha_{K+1}$. Define $L_k(p) = \sum_{r \leq p} \mathbf{1}_{X_r = k}$; then $(L_1(n), \ldots, L_K(n), L_{K+1}(n))$ has the same distribution as $(L_1, \ldots, L_K, L_{K+1})$, and we identify the two in our notations. For a fixed $n$, and for any injective function $\iota : \{1, \ldots, k\} \to \{1, \ldots, n\}$, consider the event

$$E_\iota := \{L_1(\iota(1)) = 1, \ldots, L_1(\iota(K)) = 1\},$$

i.e. $\iota(i)$ is the first appearance of $i$ among the variables $X_1, \ldots, X_n$. Note that $E := \{L_1, \ldots, L_K \geq 1\} = \bigsqcup_\iota E_\iota$ where the sum is taken over all such injective functions. Then

$$\mathbb{E}[(\sum_{i=1}^{K} L_i)^l | E] = \sum_\iota \mathbb{P}(E_\iota | E) \mathbb{E}[(\sum_{i=1}^{K} L_i)^l | E_\iota].$$

Fix a function $\iota$, and assume without loss of generality that $\iota(1) < \iota(2) < \ldots < \iota(K)$. Conditioned by $A_\iota$, the variable $Y_i := \sum_{r=\iota(i)+1}^{\iota(i+1)-1} \mathbf{1}_{X_r \neq K+1}$ is a binomial variable of parameters $\iota(i+1) - \iota(i) - 2 \leq n$ and $\frac{\sum_{i=1}^{i} \alpha_i}{\alpha_1 + \ldots + \alpha_i + \alpha_{K+1}} \leq \sum_{i=1}^{K} \alpha_i$. Hence $\mathbb{E}[Y_i^l | E_\iota] \leq C_1(l)(n \sum_{i=1}^{K} \alpha_i)^l$ using classical bounds on the $l$-th moment of a binomial variable, and we see that

$$\mathbb{E}[(\sum_{i=1}^{K} L_i)^l | E_\iota] = \mathbb{E}[(\sum_{r=1}^{n} \mathbf{1}_{X_r \neq K+1})^l | E_\iota] = \mathbb{E}[(K + \sum_{i=0}^{K} \sum_{r=\iota(i)+1}^{\iota(i+1)-1} \mathbf{1}_{X_r \neq K+1})^l | E_\iota]$$

$$\leq C_2(K, l)(K^l + \sum_{i=0}^{K} \mathbb{E}[Y_i^l | E_\iota]) \leq C_3(K, l)(1 + (n \sum_{i=1}^{K} \alpha_i)^l)$$

where we write $\iota(0) = 0$ and $\iota(K+1) = n$ to simplify notations.

$\qquad\square$

Let us wrap things up. Recall Lemma G.6: it holds that

$$\mu_{n,i}(T_s) \leq \frac{1}{n} \sum_{j=1}^{n} L_j^{i+2}, \tag{29}$$

where $L_j$ is the cardinality of the set $\bigcup_{z \in \Xi_{n,s}^j} \sigma_{A_n}(z)$. But according to Lemma G.7, if $z \in \Xi_{n,s}^j$, then $\|X_j - z\| \leq c_0 R_{jn}$. In particular, $d_{A_n}(z) \leq c_0 R_{jn}$, and any point $y \in \sigma_{A_n}(z)$ is at distance less than $2c_0 R_{jn}$ from $X_j$. Hence, $L_j$ is smaller than $K_{jn}(\kappa)$ for $\kappa = 2c_0$. Choose $\varepsilon_0$ so that $\kappa \varepsilon_0 \leq \kappa \varepsilon_0 \leq \min(\rho_0, \tau(\mathsf{M})/4)$. We are in position to apply Lemma G.11. We further remark that Lemma G.7 implies that $\Xi_{n,s}^j$ is empty if $c_0 R_{jn} < s n^{-1/m}$.

Hence, by Jensen's inequality,

$$
\begin{aligned}
\mathbb{E}[\mu_{n,i}(T_s)^2 \mathbf{1}\{E\}] &\leq \mathbb{E}\left[\left(\frac{1}{n}\sum_{j=1}^{n}\mathbf{1}\{2c_0 R_{jn} \geq sn^{-1/m}\}K_{jn}(2c_0)^{i+2}\right)^2\right] \\
&\leq \mathbb{E}\left[\mathbf{1}\{2c_0 R_{1n} \geq sn^{-1/m}\}K_{1n}(2c_0)^{2i+4}\right] \\
&\leq \sqrt{\mathbb{P}(2c_0 R_{jn} \geq sn^{-1/m})\mathbb{E}[K_{1n}(2c_0)^{4i+8}]} \\
&\leq C_{m,i}\exp(-c_m f_{\min}s^m)
\end{aligned}
$$

for some constants $c_m, C_{m,i} > 0$, where we apply Lemma G.10 and Lemma G.11 at the last line. $\square$

This completes the proof of Theorem 4.1. $\square$

We now prove Proposition 4.2:

**Proposition 4.2.** *Let* $\mathsf{M}$ *be a generic $m$-dimensional submanifold. Assume that $P$ has a density $f$ on $\mathsf{M}$ bounded away from $0$ and $\infty$. Let $i \geq 0$ be an integer. There exists an optimal matching $\gamma_n : \mathrm{dgm}_i(\mathsf{A}_n) \cup \partial\Omega \to \mathrm{dgm}_i(\mathsf{M}) \cup \partial\Omega$ for the bottleneck distance between $\mathrm{dgm}_i(\mathsf{A}_n)$ and $\mathrm{dgm}_i(\mathsf{M})$ such that for any $q \geq 1$:*

- *Region (2): It holds that $\max_{u \in \mathrm{dgm}_i^{(2)}(\mathsf{A}_n)} |u_2 - \gamma_n(u)_2| = O_{L^q}(n^{-2/m})$.*

- *Region (3): It holds that $\max_{u \in \mathrm{dgm}_i^{(3)}(\mathsf{A}_n)} \|u - \gamma_n(u)\|_\infty = O_{L^q}(n^{-2/m})$ and $\#(\mathrm{dgm}_i^{(3)}(\mathsf{A}_n)) = O_{L^q}(1)$.*

*Proof.* Let $\varepsilon_n = d_H(\mathsf{A}_n, \mathsf{M})$. Let $\Pi_\mathsf{M} := \{x \in \sigma_\mathsf{M}(z) : z \in \mathrm{Crit}(\mathsf{M})\}$ be the (finite) set of projections of critical points $z \in \mathrm{Crit}(\mathsf{M})$. The proof of Theorem 3.3 relied on the use of Theorem 1.6 in [6]. This theorem states roughly that both critical points $z \in \mathrm{Crit}(\mathsf{A}_n)$ far from $\mathsf{M}$ and their projections $x \in \sigma_{\mathsf{A}_n}(z)$ are stable with respect to the Hausdorff distance, meaning that every such point $z$ is at distance $O(\varepsilon_n)$ from a critical point $z' \in \mathrm{Crit}(\mathsf{M})$, with $x$ being at distance $O(\varepsilon_n)$ from a point $x' \in \Pi_\mathsf{M}$. Thus, the number of critical points of $\mathsf{A}_n$ located close to a given $z' \in \mathrm{Crit}(\mathsf{M})$ is crudely upper bounded by the number of subsets which can be formed by selecting elements in neighborhoods of size $O(\varepsilon_n)$ around $x' \in \sigma_\mathsf{M}(z')$. We used the same idea to bound the cardinality of $\mathrm{dgm}_i^{(3)}(\mathsf{A})$ in Theorem 3.3.

In a random setting, the distance $\varepsilon_n$ is of order $(\ln n/n)^{1/m}$, as suggested by Lemma G.5, while the number of points found in a ball of radius $r$ is typically of order $nr^m$, yielding a logarithmic number of elements in a neighborhood of size $O(\varepsilon_n)$ around a given point $x \in \mathsf{M}$. Hence, our earlier strategy is not tight enough to bound in expectation the cardinality of $\mathrm{dgm}_i^{(3)}(\mathsf{A})$ by a constant.

We improve upon this strategy with the following intuition: the maximal distance between a point $x$ of $\mathsf{M}$ and the point cloud $\mathsf{A}_n$ does not really matter in [6, Theorem 1.6], but only the density of the point cloud $\mathsf{A}_n$ *around* the (finitely many) projections $x \in \Pi_\mathsf{M}$. Although $\varepsilon_n$ is of order $(\ln n/n)^{1/m}$, the distance between a *fixed* point $x \in \mathsf{M}$ and $\mathsf{A}_n$ is known to be of order $n^{-1/m}$ (see (21)). This remark explains how we can intuitively replace neighborhoods of radii $(\ln n/n)^{1/m}$ by radii of size $n^{-1/m}$ in the previous arguments.

We now make these ideas rigorous. Let $\mathrm{Crit}_>(\mathsf{A}_n) = \{z \in \mathrm{Crit}(\mathsf{A}_n) : d_{\mathsf{A}_n}(z) \geq \tau(\mathsf{M})/2\}$ denote the set of critical points of $\mathsf{A}_n$ "far" from $\mathsf{M}$ and let $\Pi_{\mathsf{A}_n} := \{x \in \sigma_{\mathsf{A}_n}(z) : z \in \mathrm{Crit}_>(\mathsf{A}_n)\}$ be the corresponding set of projections. We let $\varepsilon_0$ be a small constant to be fixed later. Theorem 1.6 from [6] states that, thanks to the genericity of $\mathsf{M}$, there exist $K_1, K_2 > 0$ such that for $\varepsilon_0$ small enough, if $\varepsilon_n < \varepsilon_0$:

- there exists a map $\phi : \mathrm{Crit}_>(\mathsf{A}_n) \to \mathrm{Crit}(\mathsf{M})$ such that $d(z, \phi(z)) \leq K_1\varepsilon_n$ for all $z \in \mathrm{Crit}_>(\mathsf{A}_n)$, and

- there exists a map $\psi : \Pi_{\mathsf{A}_n} \to \Pi_\mathsf{M}$ such that $d(x, \psi(x)) \leq K_2\varepsilon_n$ for all $x \in \Pi_{\mathsf{A}_n}$.

Furthermore, the map $\psi$ is such that for each $z \in \mathrm{Crit}_>(\mathsf{A}_n)$ and each $x' \in \sigma_\mathsf{M}(\phi(z))$, there exists $x \in \sigma_{\mathsf{A}_n}(z)$ such that $\psi(x) = x'$ (if $\varepsilon_0$ is chosen small enough). Indeed, due to the genericity of M, each critical point $z' \in \mathrm{Crit}(\mathsf{M})$ belongs to the relative interior of $\sigma_\mathsf{M}(z')$ (as stated in Appendix C). In particular, it cannot belong to the convex hull of any strict subset of $\sigma_\mathsf{M}(z')$. By continuity of the convex hull for the Hausdorff distance, as soon as $z \in \mathrm{Crit}_>(\mathsf{A}_n)$ is close enough to $\phi(z)$ and its projections $x \in \sigma_{\mathsf{A}_n}(z)$ are close enough to $\sigma_\mathsf{M}(\phi(z))$ (i.e. as soon as $\varepsilon_n$ is small enough), the point $z$ cannot belong to the convex hull of any subset $S \subset \sigma_{\mathsf{A}_n}(z)$ that does not contain for each $x' \in \sigma_\mathsf{M}(\phi(z))$ at least one point $x$ such that $\psi(x) = x'$. As $z$ must belong to the convex hull of $\sigma_{\mathsf{A}_n}(z)$, we conclude that each $x' \in \sigma_\mathsf{M}(\phi(z))$ has at least one preimage by $\psi$ in $\sigma_{\mathsf{A}_n}(z)$. As $\mathrm{Crit}(\mathsf{M})$ is finite, taking the minimum distance such that this property holds for $z'$ over all $z' \in \mathrm{Crit}(\mathsf{M})$ proves the claim.

We introduce the random function defined as

$$\forall r \geq 0, \ E(r) := \min(\sup\{d_{\mathsf{A}_n}(y) : \ y \in \mathsf{M} \cap \Pi_\mathsf{M}^r\}, \varepsilon_0) =: \min(F(r), \varepsilon_0), \tag{30}$$

where $\Pi_\mathsf{M}^r$ is as usual the $r$-offset of $\Pi_\mathsf{M}$. This random variable measures the density of the point cloud $\mathsf{A}_n$ in neighborhoods of size $r$ of points $x' \in \Pi_\mathsf{M}$.

Let

$$\rho_n := \sup\{d_{\Pi_\mathsf{M}}(x) : x \in \Pi_{\mathsf{A}_n}\} \tag{31}$$

give (when $\varepsilon_n < \varepsilon_0$ for $\varepsilon_0$ small enough) the largest distance between a projection $x \in \Pi_{\mathsf{A}_n}$ and the corresponding projection $\psi(x) \in \Pi_\mathsf{M}$. We also define

$$\eta_n := \sup\{d_{\Pi_\mathsf{M}}(x) : \ x = \pi_\mathsf{M}(z), \ z \in \mathrm{Crit}_>(\mathsf{A}_n)\}. \tag{32}$$

We require the following controls on the random variables $\rho_n$ and $\eta_n$.

**Lemma G.13.** *There exist positive constants* $\varepsilon_0 = \varepsilon_0(\mathsf{M})$, $K_3 = K_3(\mathsf{M})$, $K_4 = K_4(\mathsf{M})$ *and* $K_5 = K_5(\mathsf{M})$ *such that if* $\varepsilon_n < \varepsilon_0$, *it holds that for any* $z \in \mathrm{Crit}_>(\mathsf{A}_n)$

$$|d_{\mathsf{A}_n}(z) - d_\mathsf{M}(\phi(z))| \leq K_3 E(\eta_n)^2, \tag{33}$$
$$\eta_n \leq K_4 E(\eta_n), \tag{34}$$
$$\rho_n \leq K_5 E(\eta_n). \tag{35}$$

*Proof.* The lemma follows from a careful read of the proof of Theorem 1.6 in [6]. First, remark that in the proof of Lemma 5.1 in [6] (applied with $r = \tau(\mathsf{M})/2$, $R = \mathrm{diam}(\mathsf{M})$), the Hausdorff distance $\varepsilon = d_H(\mathsf{A}_n, \mathsf{M})$ can be replaced by $E(\eta_n)$. Hence, if $z \in \mathrm{Crit}_>(\mathsf{A}_n)$, there exists a $\mu$-critical point $z'$ of M at distance less than $E(\eta_n)$, with $\mu \leq L_1(\mathsf{M})E(\eta_n)$. By genericity, for a choice of $\varepsilon_0$ small enough, this point $z'$ is at distance $L_2(\mathsf{M})\mu$ from a critical point $z_0 \in \mathrm{Crit}(\mathsf{M})$. For $\varepsilon_0$ small enough, this point $z_0$ is necessarily equal to $\phi(z)$, with $\|z - \phi(z)\| \leq (1 + L_2 L_1)E(\eta_n)$. Due to the Lipschitz property of the projection onto M around $z_0$ (see the arguments found at the bottom of p. 19 in [6]), any point $x \in \pi_\mathsf{M}(z)$ is such that $\|x - x'\| \leq L_3(\mathsf{M})\|z - \phi(z)\|$ for some $x' \in \sigma_\mathsf{M}(\phi(z))$. By taking the supremum over all such points $x$, we obtain that

$$\eta_n \leq L_3(1 + L_2 L_1)E(\eta_n),$$

proving (34).

One can also check that $\varepsilon_n$ can be replaced by $E(\eta_n)$ in the end of the proof of Theorem 1.6 in [6], so that (35) holds.

Let us now prove (33). Let $z \in \mathrm{Crit}_>(\mathsf{A}_n)$. Consider $x_{\mathsf{A}_n} \in \sigma_{\mathsf{A}_n}(z)$; we know that $\|\psi(x_{\mathsf{A}_n}) - x_{\mathsf{A}_n}\| \leq K_5 E(\eta_n)$ using (33). As $\phi(z) - \psi(x_{\mathsf{A}_n})$ is orthogonal to $T_{\psi(x_{\mathsf{A}_n})}\mathsf{M}$, [39] states that $\|\pi^\perp_{\psi(x_{\mathsf{A}_n})}(x_{\mathsf{A}_n} - \psi(x_{\mathsf{A}_n}))\| \leq \frac{\|x_{\mathsf{A}_n} - \psi(x_{\mathsf{A}_n})\|^2}{2\tau(\mathsf{M})} \leq \frac{K_5^2 E(\eta_n)^2}{2\tau(\mathsf{M})}$, hence

$$\|\phi(z) - x_{\mathsf{A}_n}\|^2 = \|\phi(z) - \psi(x_{\mathsf{A}_n})\|^2 + 2\langle \phi(z) - \psi(x_{\mathsf{A}_n}), \psi(x_{\mathsf{A}_n}) - x_{\mathsf{A}_n}\rangle + \|\psi(x_{\mathsf{A}_n}) - x_{\mathsf{A}_n}\|^2$$

$$\leq d_\mathsf{M}(\phi(z))^2 + \frac{K_5^2 E(\eta_n)^2}{\tau(\mathsf{M})} d_\mathsf{M}(\phi(z)) + K_5^2 E(\eta_n)^2 \leq d_\mathsf{M}(\phi(z))^2 + E(\eta_n)^2 L_4,$$

where $L_4 = K_5^2 \left(\frac{R(\mathsf{M})}{\tau(\mathsf{M})} + 1\right)$, $R(\mathsf{M})$ is the radius of the smallest closed ball that contains M, and $d_\mathsf{M}(\phi(z)) \leq R(\mathsf{M})$ because a critical point must belong to the convex hull of its projections. As

the same bound applies to each of the projections $x'_{A_n} \in \sigma_{A_n}(z)$, we find that the closed ball $\overline{B}\left(\phi(z), \sqrt{d_M(\phi(z))^2 + E(\eta_n)^2 L_4}\right)$ contains $\sigma_{A_n}(z)$. Since $z$ is the center of the smallest ball that contains $\sigma_{A_n}(z)$, whose radius is $d_{A_n}(z)$, we find that

$$d_{A_n}(z)^2 \leq d_M(\phi(z))^2 + E(\eta_n)^2 L_4. \tag{36}$$

Similarly, consider now $x_M \in \sigma_M(\phi(z))$, and let $x_{A_n} \in \psi^{-1}(x_M) \cap \sigma_{A_n}(z)$ (we have seen below the definition of $\phi$ and $\psi$ that its existence is guaranteed). We have shown earlier that there exists $x \in \sigma_M(z)$ such that $\|x - x_M\|, \|x - x_{A_n}\| \leq O(E(\eta_n))$, and the same reasoning as above yields that $\|z - x_{A_n}\|^2 = \|z - x\|^2 + O(E(\eta_n)^2)$ and $\|z - x_M\|^2 = \|z - x\|^2 + O(E(\eta_n)^2)$ (with all big $O$ constants depending only on M), hence that $\|z - x_M\|^2 \leq \|z - x_{A_n}\|^2 + E(\eta_n)^2 L_5(M) = d_{A_n}(z)^2 + E(\eta_n)^2 L_5$.

As before, this shows that the closed ball $\overline{B}\left(z, \sqrt{d_{A_n}(z)^2 + E(\eta_n)^2 L_5}\right)$ contains $\sigma_M(\phi(z))$, hence that

$$d_M(\phi(z))^2 \leq d_{A_n}(z)^2 + E(\eta_n)^2 L_5. \tag{37}$$

This, together with (36) and the fact that $d_{A_n}(z) \geq \tau(M)/2$, shows that $|d_{A_n}(z) - d_M(\phi(z))| \leq K_3(M)E(\eta_n)^2$. □

Consider a (random) optimal matching $\gamma_n : \mathrm{dgm}_i(A_n) \cup \partial\Omega \to \mathrm{dgm}_i(M) \cup \partial\Omega$ such that $\max_{u \in \mathrm{dgm}_i^{(2)}(A_n)} |u_2 - \gamma_n(u)_2| \leq C\varepsilon_n^2$ and $\max_{u \in \mathrm{dgm}_i^{(3)}(A_n)} \|u - \gamma_n(u)\|_\infty \leq C\varepsilon_n^2$ whose existence is guaranteed by Theorem 2.2 (for some $C = C(M) > 0$). For $r \geq 0$, we let $N(r)$ be the number of points of $A_n$ found in $\Pi_M^r$ (i.e. at distance less than $r$ from a point in $\Pi_M$).

**Lemma G.14.** *There exists $\varepsilon_0 = \varepsilon_0(M)$ such that if $\varepsilon_n \leq \varepsilon_0$, then*

$$\max_{u \in \mathrm{dgm}_i^{(2)}(A_n)} |u_2 - \gamma_n(u)_2| \leq K_3 E(\eta_n)^2,$$

$$\max_{u \in \mathrm{dgm}_i^{(3)}(A_n)} \|u - \gamma_n(u)\|_\infty \leq K_3 E(\eta_n)^2.$$

*Furthermore, $\#(\mathrm{dgm}_i^{(3)}(A_n))$ is smaller than $N(\rho_n)^{i+2}$.*

*Proof.* Let us assume from now on that $\varepsilon_0$ is small enough that $2(K_3 + C)\varepsilon_0^2$ is smaller than the smallest difference between two distinct critical values of $d_M$.

Consider a point $(u_1, u_2) \in \mathrm{dgm}_i^{(3)}(A_n)$ that is mapped by $\gamma_n$ to the diagonal. The coordinates $u_1$ and $u_2$ differ by at most $C\varepsilon_n^2$, hence they are critical values of $d_{A_n}$ that correspond to critical points that are mapped by $\phi$ to two critical points of M that have the same critical value (due to $2(K_3 + C)\varepsilon^2$ being smaller than the smallest difference between two distinct critical values of $d_M$). But as stated in Equation (33) of Lemma G.13, these critical values must then be $K_3 E(\eta_n)^2$-close to that of the two critical points of M, hence $\|u - \gamma_n(u)\|_\infty = d(u, \partial\Omega) = (u_2 - u_1)/2 \leq K_3 E(\eta_n)^2$, as desired.

Likewise, consider a point $(u_1, u_2)$ of $\mathrm{dgm}_i^{(3)}(A_n)$ that is mapped by $\gamma_n$ to a point $(v_1, v_2)$ of $\mathrm{dgm}_i^{(3)}(M)$. The birth coordinates $u_1$ and $v_1$ differ by at most $C\varepsilon^2$, hence the associated critical values must correspond to a critical point $z$ of $A_n$ and a critical point $z'$ of M such that $\phi(z)$ has the same filtration value as $z'$ (again due to $2(K_3 + C)\varepsilon^2$ being smaller than the smallest difference between two distinct critical values of $d_M$). But as above, we have $|u_1 - v_1| = |d_{A_n}(z) - d_M(z')| = |d_{A_n}(z) - d_M(\phi(z))| \leq K_3 E(\eta_n)^2$. As the same applies to $u_2$ and $v_2$, we find that $\|u - \gamma_n(u)\|_\infty \leq K_3 EE(\eta_n)^2$ and $\max_{u \in \mathrm{dgm}_i^{(3)}(A_n)} \|u - \gamma_n(u)\|_\infty \leq K_3 E(\eta_n)^2$. The same reasoning shows that $\max_{u \in \mathrm{dgm}_i^{(2)}(A_n)} |u_2 - \gamma_n(u)_2| \leq K_3 E(\eta_n)^2$.

Finally, and by definition, $\Pi_{A_n} = \{x \in \sigma_{A_n}(z) : z \in \mathrm{Crit}(A_n), d_{A_n}(z) \geq \tau(M)/2\}$ is included in $A_n \cap \Pi_M^{\rho_n}$, hence $\#(\Pi_{A_n}) \leq N(\rho_n)$. Lemma E.1 applied to $A_n$ and the interval $[\tau(M)/2, +\infty)$ then yields that the number of points in $\mathrm{dgm}_i^{(3)}(A_n)$ is smaller than $N(\rho_n)^{i+2}$. □

**Lemma G.15.** *Let $\varepsilon_0$ be the parameter defined in Lemma G.14. It holds that for all $q \geq 1$, $E(\eta_n)\mathbf{1}\{\varepsilon_n \leq \varepsilon_0\} = O_{L^q}(n^{-1/m})$ and $N(\rho_n)\mathbf{1}\{\varepsilon_n \leq \varepsilon_0\} = O_{L^q}(1)$.*

Assume for a moment that Lemma G.15 holds. Then, we write (for a choice of $\varepsilon_0$ small enough)

$$\max_{u \in \mathrm{dgm}_i^{(2)}(\mathsf{A}_n)} |u_2 - \gamma_n(u)_2| \leq K_3 E(\eta_n)^2 \mathbf{1}\{\varepsilon_n \leq \varepsilon_0\} + R(\mathsf{M})\mathbf{1}\{\varepsilon_n > \varepsilon_0\}$$

$$\leq K_3 E(\eta_n)^2 \mathbf{1}\{\varepsilon_n \leq \varepsilon_0\} + R(\mathsf{M})\mathbf{1}\{\varepsilon_n > \varepsilon_0\}. \tag{38}$$

where we recall that $R(\mathsf{M})$ is the radius of the smallest enclosing ball of $\mathsf{M}$. Because of Lemma G.5, the random variable $\mathbf{1}\{\varepsilon_n > \varepsilon_0\}$ is a $O_{L^q}(n^{-2/m})$ for all $q \geq 1$. But then, because of Lemma G.15, we obtain that the right-hand side in (38) is a $O_{L^q}(n^{-2/m})$ for all $q \geq 1$. Likewise, we obtain that $\max_{u \in \mathrm{dgm}_i^{(3)}(\mathsf{A}_n)} \|u - \gamma_n(u)\|_\infty = O_{L^q}(n^{-2/m})$ for all $q \geq 1$. At last, we have

$$\#(\mathrm{dgm}_i^{(3)}(\mathsf{A}_n)) \leq N(\rho_n)^{i+2}\mathbf{1}\{\varepsilon_n \leq \varepsilon_0\} + n^{i+2}\mathbf{1}\{\varepsilon_n > \varepsilon_0\}.$$

The first term is a $O_{L^q}(1)$ for all $q \geq 1$ because of Lemma G.15, while the second one is a $O_{L^q}(1)$ for all $q \geq 1$ because of Lemma G.5. This concludes the proof of Proposition 4.2. Let us now prove Lemma G.15. $\qquad\square$

*Proof of Lemma G.15.* Let $t > 0$. The key observation to obtain Lemma G.15 is that (34) implies that if $\eta_n > t$, then there exists $r > t$ with $r \leq K_4 E(r)$.

**Lemma G.16.** *For all $\lambda > 0$, there exist positive constants $C_\lambda$, $c_\lambda$ (depending on $\lambda$, $\mathsf{M}$ and $f_{\min}$) such that for all $r > 0$, $\mathbb{P}(r \leq \lambda E(r)) \leq C_\lambda \exp(-c_\lambda n r^m)$.*

*Proof.* Remark that if $r \leq \lambda E(r)$, then $r \leq \lambda F(r)$. The set $\mathsf{M} \cap \Pi_{\mathsf{M}}^r$ is the union of a finite number of balls of radius $r$. Hence, it can be covered by $C_\lambda = C_\lambda(\mathsf{M})$ open balls of radius $r/(2\lambda)$, with centers $x_1, \ldots, x_{C_\lambda} \in \mathsf{M}$. Note that if all these balls intersect $\mathsf{A}_n$, then for all $y \in \mathsf{M} \cap \Pi_{\mathsf{M}}^r$, $d_{\mathsf{A}_n}(y) < r/\lambda$. Hence, if $\lambda F(r) \geq r$, then the intersection of one of these balls with $\mathsf{A}_n$ is empty. Hence, according to Lemma G.1,

$$\mathbb{P}(r \leq \lambda F(r)) \leq \sum_{k=1}^{C_\lambda} \mathbb{P}(d_{\mathsf{A}_n}(x_k) \geq r/(2\lambda)) \leq C_\lambda \exp(-C(\mathsf{M}) f_{\min} n (r/\lambda)^m). \qquad\square$$

Remark that the fonction $E$ is nondecreasing and 1-Lipschitz continuous: we have for $r < s$, $E(r) \leq E(s) \leq E(r) + (r - s)$. Fix $t > 0$ and consider the sequence $t_k = a^k t$ for some $a > 1$ to fix. Assume that $\eta_n > t$ and that $\varepsilon_n \leq \varepsilon_0$. Then $\eta_n$ is between two values $t_k < t_{k+1}$. But then, according to Lemma G.14,

$$\frac{E(t_k)}{t_k} \geq \frac{E(\eta_n) - (\eta_n - t_k)}{t_k} \geq \frac{1}{a}\frac{E(\eta_n)}{\eta_n}\frac{t_{k+1}}{t_k} - (a-1) \geq \frac{1}{K_4} - (a-1)$$

$$\frac{E(t_{k+1})}{t_{k+1}} \geq \frac{1}{a}\frac{E(\eta_n)}{t_k} \geq \frac{1}{a}\frac{E(\eta_n)}{\eta_n} \geq \frac{1}{K_4 a}.$$

Choose $a > 1$ such that $\frac{1}{K_4} - (a-1) > 0$, and let

$$\lambda = \min\left(\frac{1}{K_4} - (a-1), \frac{1}{K_4 a}\right) > 0$$

We have proven that if $\varepsilon_n \leq \varepsilon_0$ and $\eta_n > t$, then there exists $k \geq 0$ with $E(t_k) \geq \lambda t_k$.

Hence,

$$\mathbb{P}(\eta_n > t, \varepsilon_n \leq \varepsilon_0) \leq C_\lambda \sum_{k \geq 0} \exp(-c_\lambda a^{km} n t^m).$$

A standard comparison between this sum and an integral shows that this sum is at most of order

$$K_6 (n t^m)^{-1} \exp(-K_7 n t^m). \tag{39}$$

for two positive constants $K_6$, $K_7$ depending on $\mathsf{M}$ and $f_{\min}$. But when $n t^m \leq 1$, we can simply use the bound $\mathbb{P}(\eta_n > t, \varepsilon_n \leq \varepsilon_0) \leq 1$. Hence,

$$\mathbb{P}(\eta_n > t, \varepsilon_n \leq \varepsilon_0) \leq \min(1, K_6 (n t^m)^{-1} \exp(-K_7 n t^m)) \leq K_8 \exp(-K_9 n t^m)) \tag{40}$$

for two other constants $K_8$, $K_9$.

To summarize, we have shown that the random variable $n\eta_n^m \mathbf{1}\{\varepsilon_n \le \varepsilon_0\}$ is subexponential, with subexponential norm depending only on M and $f_{\min}$, see e.g. [74, Section 2.7]. We will now simply say that a random variable is subexponential to indicate that it is subexponential with a norm depending only on M and $f_{\min}$.

Lemma G.1 shows that the random variable $nd_{\mathsf{A}_n}(x)^m$ for a *fixed* $x \in \mathsf{M}$ is also subexponential. Thus, so is $nE(0)^m = n\max_{x\in\Pi_\mathsf{M}} d_{\mathsf{A}_n}(x)^m$, as a maximum of a finite number of subexponential random variables. As $nE(\eta_n)^m \le n(\eta_n + E(0))^m \le 2^{m-1}(n\eta_n^m + nE(0)^m)$, the random variable $nE(\eta_n)^m \mathbf{1}\{\varepsilon_n \le \varepsilon_0\}$ is also subexponential. In particular, we have $E(\eta_n)\mathbf{1}\{\varepsilon_n \le \varepsilon_0\} = O_{L^q}(n^{-1/m})$ for all $q \ge 1$.

It remains to bound $N(\rho_n)$. First, because of (35), the random variable $n\rho_n^m \mathbf{1}\{\varepsilon_n \le \varepsilon_0\}$ is subexponential. Also, for a fixed $t$, $N(t)$ follows a binomial distribution of parameter $n$ and $P(\Pi_\mathsf{M}^t)$. As long as $t \le \tau(\mathsf{M})/4$, a ball of radius $t$ is of mass smaller than $C_m f_{\max} t^m$ (see Lemma G.1). Let $0 \le k \le n$ be an integer. For any $t \le \tau(\mathsf{M})/4$, we bound

$$\mathbb{P}(N(\rho_n) \ge k, \varepsilon_n \le \varepsilon_0) \le \mathbb{P}(N(t) \ge k, \rho_n \le t) + \mathbb{P}(\rho_n > t, \varepsilon_n \le \varepsilon_0)$$
$$\le \mathbb{P}(N(t) \ge k) + 2\exp(-K_{10}nt^m),$$

where $K_{10}$ is proportional to the subexponential norm of $n\rho_n^m \mathbf{1}\{\varepsilon_n \le \varepsilon_0\}$, see [74, Section 2.7]. Let $t = (k/n)^{1/m}\min(\tau(\mathsf{M})/4, 1/(\#\Pi_\mathsf{M} \cdot 2C_m f_{\max})^{1/m})$. This choice of $t$ ensures that $t \le \tau(\mathsf{M})/4$ and that $\mathbb{E}[N(t)] \le n(\#\Pi_\mathsf{M})C_m f_{\max} t^m \le k/2$. Then, by Bernstein's inequality [74, Theorem 2.8.4],

$$\mathbb{P}(N(t) \ge k) \le \mathbb{P}(N(t) - \mathbb{E}[N(t)] \ge k/2) \le \exp\left(-\frac{k^2/8}{n(\#\Pi_\mathsf{M})C_m f_{\max} t^m + k/6}\right)$$
$$\le \exp\left(-K_{11}k\right)$$

for some constant $K_{11}$ depending on $m$, M and $f_{\max}$. We have proven that for all $k \ge 0$,

$$\mathbb{P}(N(\rho_n) \ge k, \varepsilon_n \le \varepsilon_0) \le 2\exp\left(-K_{11}k\right) + 2\exp(-K_{10}nt^m)$$
$$\le 2\exp\left(-K_{11}k\right) + 2\exp(-K_{12}k)$$

for some constant $K_{12}$ depending on $m$, M, $f_{\min}$ and $f_{\max}$. Hence, the random variable $N(\rho_n)\mathbf{1}\{\varepsilon_n \le \varepsilon_0\}$ is subexponential, with a subexponential norm depending on M, $f_{\min}$ and $f_{\max}$. This implies in particular that $N(\rho_n)\mathbf{1}\{\varepsilon_n \le \varepsilon_0\} = O_{L^q}(1)$ for all $q \ge 1$. $\qquad\square$

Corollary 4.3 is a simple consequence of Proposition 4.2:

**Corollary 4.3.** *Let $p \ge 1$ and let $0 \le i < d$ be an integer. Under the same assumptions as in Proposition 4.2, the following holds:*

- *If $p > m$, then $\mathbb{E}[\mathrm{OT}_p^p(\mathrm{dgm}_i(\mathsf{A}_n), \mathrm{dgm}_i(\mathsf{M}))] \to 0$ as $n \to \infty$.*

- *If $p = m$, $\mathbb{E}[\mathrm{OT}_p^p(\mathrm{dgm}_i(\mathsf{A}_n), \mathrm{dgm}_i(\mathsf{M}))] \to \mathrm{Pers}_p(\mu_{\infty,i,m})\mathrm{Vol}(\mathsf{M})$ as $n \to \infty$, where $\mathrm{Vol}(\mathsf{M})$ is the volume of $\mathsf{M}$.*

- *If $p < m$ and $i < m$, then $\mathbb{E}[\mathrm{OT}_p^p(\mathrm{dgm}_i(\mathsf{A}_n), \mathrm{dgm}_i(\mathsf{M}))] \to +\infty$ as $n \to \infty$.*

*Furthermore, for all $\alpha > 0$, $\mathrm{Pers}_\alpha(\mathrm{dgm}_i(\mathsf{A}_n))$ is equal to*

$$\mathrm{Pers}_\alpha(\mathrm{dgm}_i(\mathsf{M})) + n^{1-\frac{\alpha}{m}}\mathrm{Pers}_\alpha(\mu_{\infty,i,m})\int_\mathsf{M} f(x)^{1-\frac{\alpha}{m}}\mathrm{d}x + o_{L^1}(n^{1-\frac{\alpha}{m}}) + O_{L^1}\left(\left(\frac{\log n}{n}\right)^{\frac{1}{m}}\right).$$

*Proof.* Let $\varepsilon_n = d_H(\mathsf{A}_n, \mathsf{M})$. Consider the event $E = \{\varepsilon_n < \tau(M)/2\}$. According to Theorem 2.2, on $E$, the number of points of $\mathrm{dgm}_i^{(2)}(\mathsf{A}_n)$ is bounded by some constant $N_0$ depending only on M.

- Consider the optimal matching $\gamma_n$ given by Proposition 4.2. On the event $E$, this optimal matching sends all the points of $\mathrm{dgm}_i^{(1)}(\mathsf{A}_n)$ to the diagonal $\partial\Omega$. Thus, we have (when $E$ is

satisfied)

$$|\mathrm{OT}_p^p(\mathrm{dgm}_i(\mathsf{A}_n), \mathrm{dgm}_i(\mathsf{M})) - \mathrm{Pers}_p(\mathrm{dgm}_i^{(1)}(\mathsf{A}_n))|$$

$$\leq N_0 \max_{u \in \mathrm{dgm}_i^{(2)}(\mathsf{A}_n)} \|u - \gamma_n(u)\|_\infty^p + \#(\mathrm{dgm}_i^{(3)}(\mathsf{A}_n)) \max_{u \in \mathrm{dgm}_i^{(3)}(\mathsf{A}_n)} \|u - \gamma_n(u)\|_\infty^p. \quad (41)$$

Furthermore, note that

$$\max_{u \in \mathrm{dgm}_i^{(2)}(\mathsf{A}_n)} \|u - \gamma_n(u)\|_\infty^p \leq \max \left( \max_{(u_1,u_2) \in \mathrm{dgm}_i^{(2)}(\mathsf{A}_n)} |u_2 - \gamma_n(u)_2|^p, \varepsilon_n^p \right)$$

$$\leq \max_{(u_1,u_2) \in \mathrm{dgm}_i^{(2)}(\mathsf{A}_n)} |u_2 - \gamma_n(u)_2|^p + \varepsilon_n^p.$$

We take the expectation and apply the Cauchy-Schwarz inequality to obtain (together with Theorem 4.1, Proposition 4.2 and Lemma G.5):

$$\mathbb{E}[\mathrm{OT}_p^p(\mathrm{dgm}_i(\mathsf{A}_n), \mathrm{dgm}_i(\mathsf{M}))\mathbf{1}\{E\}] \leq n^{1-p/m}\mathrm{Pers}_p(\mu_{f,i}) + o(1).$$

When $E$ is not realized, we crudely bound $\mathrm{OT}_p^p(\mathrm{dgm}_i(\mathsf{A}_n), \mathrm{dgm}_i(\mathsf{M}))$ by considering the matching sending every point to the diagonal. The cost of this matching is bounded by $C_\mathsf{M}(1 + n^{i+1})$, where $n^{i+1}$ is an upper bound on the number of points in $\mathrm{dgm}_i(\mathsf{A}_n)$ and $C_\mathsf{M}$ is some constant depending on $\mathsf{M}$. Hence,

$$\mathbb{E}[\mathrm{OT}_p^p(\mathrm{dgm}_i(\mathsf{A}_n), \mathrm{dgm}_i(\mathsf{M}))\mathbf{1}\{E^c\}] \leq C_\mathsf{M}(1 + n^{i+1})\mathbb{P}(\varepsilon_n > \tau(\mathsf{M})/2) = o(1),$$

according to Lemma G.5. This proves the first bullet point.

- The second bullet point is proved likewise from (41). Indeed, in that case, it holds that

$$\mathbb{E}[|\mathrm{OT}_p^p(\mathrm{dgm}_i(\mathsf{A}_n), \mathrm{dgm}_i(\mathsf{M})) - \mathrm{Pers}_p(\mathrm{dgm}_i^{(1)}(\mathsf{A}_n))|\mathbf{1}\{E\}] = o(1). \quad (42)$$

Also, using the same crude bound, we obtain that

$$\mathbb{E}[|\mathrm{OT}_p^p(\mathrm{dgm}_i(\mathsf{A}_n), \mathrm{dgm}_i(\mathsf{M})) - \mathrm{Pers}_p(\mathrm{dgm}_i^{(1)}(\mathsf{A}_n))|\mathbf{1}\{E^c\}] = o(1). \quad (43)$$

So far, we have proved that $\mathrm{OT}_p^p(\mathrm{dgm}_i(\mathsf{A}_n), \mathrm{dgm}_i(\mathsf{M})) = \mathrm{Pers}_p(\mathrm{dgm}_i^{(1)}(\mathsf{A}_n)) + o_{L^1}(1)$. According to Theorem 4.1, it holds that $\mathrm{Pers}_p(\mathrm{dgm}_i^{(1)}(\mathsf{A}_n)) = \mathrm{Pers}_p(\mu_{f,i}) + o_{L^1}(1)$ for $p = m$, proving the second bullet point.

- For the third bullet point, we use that on the event $E$,

$$\mathrm{OT}_p^p(\mathrm{dgm}_i(\mathsf{A}_n), \mathrm{dgm}_i(\mathsf{M})) \geq \mathrm{Pers}_p(\mathrm{dgm}_i^{(1)}(\mathsf{A}_n)). \quad (44)$$

The latter is equal to $n^{1-p/m}\mathrm{Pers}_p(\mu_{f,i}) + o_{L^1}(n^{1-p/m})$, with $\mathrm{Pers}_p(\mu_{f,i}) > 0$: indeed, the support of the measure $\mu_{f,i}$ is nontrivial for $i < m$, see [47, 42]. Hence,

$$\mathbb{E}[\mathrm{OT}_p^p(\mathrm{dgm}_i(\mathsf{A}_n), \mathrm{dgm}_i(\mathsf{M}))] \geq \mathbb{E}[\mathrm{OT}_p^p(\mathrm{dgm}_i(\mathsf{A}_n), \mathrm{dgm}_i(\mathsf{M}))\mathbf{1}\{E\}]$$

$$\geq \mathbb{E}[\mathrm{Pers}_p(\mathrm{dgm}_i^{(1)}(\mathsf{A}_n))] - \mathbb{E}[\mathrm{Pers}_p(\mathrm{dgm}_i^{(1)}(\mathsf{A}_n))\mathbf{1}\{E^c\}].$$

The second term goes to zero (use the crude bound $\mathrm{Pers}_p(\mathrm{dgm}_i^{(1)}(\mathsf{A}_n)) \leq C_\mathsf{M}(1 + n^{i+1})$), while the first one diverges. This proves the third bullet point.

At last, we prove the formula for the asymptotic expansion of

$$\mathrm{Pers}_\alpha(\mathrm{dgm}_i(\mathsf{A}_n)) = \mathrm{Pers}_\alpha(\mathrm{dgm}_i^{(1)}(\mathsf{A}_n)) + \mathrm{Pers}_\alpha(\mathrm{dgm}_i^{(2)}(\mathsf{A}_n)) + \mathrm{Pers}_\alpha(\mathrm{dgm}_i^{(3)}(\mathsf{A}_n)). \quad (45)$$

The first term is equal to $n^{1-\alpha/m}\mathrm{Pers}_\alpha(\mu_{\infty,i,m})\int_\mathsf{M} f(x)^{1-\frac{\alpha}{m}}\,\mathrm{d}x + o_{L^1}(n^{1-\alpha/m})$ according to Theorem 4.1. Remark that for $u, v \in \Omega$, $|\mathrm{pers}_\alpha(u) - \mathrm{pers}_\alpha(v)| \leq 2\alpha(\mathrm{pers}_\alpha(u) + \mathrm{pers}_\alpha(v))\|u - v\|_\infty$. Hence, on the event $E = \{\varepsilon_n < \tau(\mathsf{M})/2\}$,

$$|\mathrm{Pers}_\alpha(\mathrm{dgm}_i^{(2)}(\mathsf{A}_n)) + \mathrm{Pers}_\alpha(\mathrm{dgm}_i^{(3)}(\mathsf{A}_n)) - \mathrm{Pers}_\alpha(\mathrm{dgm}_i(\mathsf{M}))|$$

$$\leq 2\alpha\varepsilon_n(\mathrm{Pers}_\alpha(\mathrm{dgm}_i^{(2)}(\mathsf{A}_n)) + \mathrm{Pers}_\alpha(\mathrm{dgm}_i^{(3)}(\mathsf{A}_n)) + \mathrm{Pers}_\alpha(\mathrm{dgm}_i(\mathsf{M}))).$$

We crudely bound the persistence of a point in $\mathrm{dgm}_i(\mathsf{A}_n)$ by $R(\mathsf{M})$, the radius of the smallest enclosing ball of $\mathsf{M}$, to obtain that

$$|\mathrm{Pers}_\alpha(\mathrm{dgm}_i^{(2)}(\mathsf{A}_n)) + \mathrm{Pers}_\alpha(\mathrm{dgm}_i^{(3)}(\mathsf{A}_n)) - \mathrm{Pers}_\alpha(\mathrm{dgm}_i(\mathsf{M}))|$$
$$\leq 2\alpha\varepsilon_n(R(\mathsf{M})^\alpha + 1)(\#(\mathrm{dgm}_i^{(2)}(\mathsf{A}_n)) + \#(\mathrm{dgm}_i^{(3)}(\mathsf{A}_n)) + \mathrm{Pers}_\alpha(\mathrm{dgm}_i(\mathsf{M}))).$$

We have already established that $\#(\mathrm{dgm}_i^{(2)}(\mathsf{A}_n)) = O_{L^2}(1)$ (actually, it is larger than $N_0$ with only exponentially small probability). Furthermore, Proposition 4.2 states that $\#(\mathrm{dgm}_i^{(3)}(\mathsf{A}_n)) = O_{L^2}(1)$. Lemma G.5 also states that $\varepsilon_n = O_{L^2}(((\log n)/n)^{1/m})$. Hence,

$$|\mathrm{Pers}_\alpha(\mathrm{dgm}_i^{(2)}(\mathsf{A}_n)) + \mathrm{Pers}_\alpha(\mathrm{dgm}_i^{(3)}(\mathsf{A}_n)) - \mathrm{Pers}_\alpha(\mathrm{dgm}_i(\mathsf{M}))| = O_{L^1}(((\log n)/n)^{1/m}).$$

This concludes the proof. $\qquad\square$

Finally, remember that we defined linear feature maps as follows (at the end of Section 4): we let $(V, \|\cdot\|)$ be a normed vector space and $\phi : \Omega \to V$ be Lipschitz continuous. For $\alpha \geq 0$, the linear feature map $\Phi_\alpha$ associated to $\phi$ and defined on the space $\mathcal{D}_f$ of finite PDs is

$$\forall a \in \mathcal{D}_f, \ \Phi_\alpha(a) = \sum_{u \in a} \mathrm{pers}(u)^\alpha \phi(u) \in V. \tag{46}$$

Let us prove the associated Corollary 4.4:

**Corollary 4.4.** *Let $\alpha \geq 1$ and let $0 \leq i < d$ be an integer. Under the same assumptions as in Proposition 4.2, it holds that $\Phi_\alpha(\mathrm{dgm}_i(\mathsf{A}_n))$ converges in probability to $\Phi_\alpha(\mathrm{dgm}_i(\mathsf{M}))$ whenever $\alpha > m$.*

*Proof.* According to [33, Proposition 5.1], the feature map $\Phi_\alpha$ is continuous with respect to the $\mathrm{OT}_\alpha$ distance. But we have shown in Corollary 4.3 that $\mathbb{E}[\mathrm{OT}_\alpha^\alpha(\mathrm{dgm}_i(\mathsf{A}_n), \mathrm{dgm}_i(\mathsf{M}))] \to 0$ as $n \to \infty$. In particular, $\mathrm{OT}_\alpha(\mathrm{dgm}_i(\mathsf{A}_n), \mathrm{dgm}_i(\mathsf{M}))$ converges in probability to 0. By continuity, so does $\|\Phi_\alpha(\mathrm{dgm}_i(\mathsf{A}_n)) - \Phi_\alpha(\mathrm{dgm}_i(\mathsf{M}))\|$. $\qquad\square$

# H  Details on numerical experiments

We provide in this section additional details on the numerical experiments conducted in Section 5. PDs are computed using `GUDHI` [62]. PDs are plotted using the `giotto-tda` library [72] and persistence images using `scikit-tda` [64]. All experiments can be easily run on a standard office laptop over a few hours.

**Simulation of generic tori.** Let $R > r$ be two positive numbers. The (standard) torus $\mathsf{M}_0$ of major radius $R$ and minor radius $r$ is given as the image of $[0, 2\pi]^2$ by the map

$$F : (\theta, \phi) \in [0, 2\pi]^2 \mapsto \begin{pmatrix} (R + r\cos(\theta))\cos(\phi) \\ (R + r\cos(\theta))\sin(\phi) \\ r\sin(\theta) \end{pmatrix}. \tag{47}$$

Consider a list $y_1, \ldots, y_k$ of pairs of angles in $[0, 2\pi]^2$, together with positive numbers $a_1, \ldots, a_K$. We let $\mathbf{r}(y) = r + \sum_{k=1}^{K} a_k \psi(d(y, y_k)/\sigma)$, where $\psi(t) = \mathbf{1}\{t < 1\}\exp(1/(t^2 - 1))$, $0 < \sigma < 2\pi$ and $d$ is the Euclidean distance on the flat torus. We then define

$$\tilde{F} : (\theta, \phi) \in [0, 2\pi]^2 \mapsto \begin{pmatrix} (R + \mathbf{r}(\theta, \phi)\cos(\theta))\cos(\phi) \\ (R + \mathbf{r}(\theta, \phi)\cos(\theta))\sin(\phi) \\ \mathbf{r}(\theta, \phi)\sin(\theta) \end{pmatrix}. \tag{48}$$

**Lemma H.1.** *Assume that $\max_{(\theta, \phi) \in [0, 2\pi]^2} \mathbf{r}(\theta, \phi) < R - r$. Then, the image of $\tilde{F}$ is a compact smooth submanifold, homeomorphic to a torus.*

*Proof.* The condition on the function $\mathbf{r}$ ensures that the function $\tilde{F}$ is injective and $(2\pi)$-periodic in each of its variables. Let $(\theta, \phi) \in [0, 2\pi]^2$. We write $\mathbf{r}$ as a shorthand for $\mathbf{r}(\theta, \phi)$. Let $S = R + \mathbf{r}\cos(\theta)$ and $U = -\mathbf{r}\sin(\theta) + \partial_\theta\mathbf{r}\cos(\theta)$. The differential of $\tilde{F}$ is equal to

$$d\tilde{F}(\theta, \phi) = \begin{pmatrix} U\cos(\phi) & -S\sin(\phi) + \partial_\phi\mathbf{r}\cos(\theta)\cos(\phi) \\ U\sin(\phi) & S\cos(\phi) + \partial_\phi\mathbf{r}\cos(\theta)\sin(\phi) \\ \mathbf{r}\cos(\theta) + \partial_\theta\mathbf{r}\sin(\theta) & \partial_\phi\mathbf{r}\sin(\theta) \end{pmatrix} \tag{49}$$

The determinant of the two first rows is equal to $SU$, and is therefore only equal to 0 if $U = 0$. But in that case,

$$d\tilde{F}(\theta, \phi) = \begin{pmatrix} 0 & -S\sin(\phi) + \partial_\phi\mathbf{r}\cos(\theta)\cos(\phi) \\ 0 & S\cos(\phi) + \partial_\phi\mathbf{r}\cos(\theta)\sin(\phi) \\ \mathbf{r}\cos(\theta) + \partial_\theta\mathbf{r}\sin(\theta) & \partial_\phi\mathbf{r}\sin(\theta) \end{pmatrix} \tag{50}$$

is a rank 2 matrix. Indeed, note that $U$ is equal to the dot product of $e_\theta = (-\sin(\theta), \cos(\theta))$ with the (nonzero) vector $(\mathbf{r}, \partial_\theta\mathbf{r})$. Hence, if $U = 0$, then the dot product of $(\mathbf{r}, \partial_\theta\mathbf{r})$ with the vector $(\cos(\theta), \sin(\theta))$ (that is perpendicular to $e_\theta$) is nonzero. Furthermore, one of the two top entries in the second column is also nonzero. Indeed, this is clear if $\partial_\phi\mathbf{r} = 0$ or if $\sin(\phi) = 0$ (we have $S > 0$ by assumption). Otherwise, having the two entries equal to zero would imply that $S^2 = -(\partial_\phi\mathbf{r})\cos^2(\theta)$, a contradiction.

In all cases, $d\tilde{F}(\theta, \phi)$ is of maximal rank. This implies the conclusion. $\square$

Let $R = 2.4$, $r = 0.8$, $\sigma = 2$ and $K = 30$. We create a random torus by sampling pairs of angles $\mathsf{Y} = \{y_1, \ldots, y_K\}$ uniformly at random, conditioned on the fact that $\max_{y \in [0,2\pi]^2} d_\mathsf{Y}(y) < \sigma$. We then draw the numbers $a_1, \ldots, a_K$ as exponential random variables of scale $0.5$, conditioned on the fact that $\max_{\phi \in [0,2\pi]} \mathbf{r}(\phi) < R - r$. We let $\mathsf{M}$ be the image of $[0, 2\pi]^2$ by $\tilde{F}$. Although we do not rigorously prove that the manifold $\mathsf{M}$ is almost surely generic, we conjecture that it is the case.

A probability measure on $P$ on $\mathsf{M}$ is obtained in the following way. Sample $(\theta, \phi) \in [0, 2\pi]^2$ so that $F(\theta, \phi)$ is uniform on the torus. Then, let $x = \tilde{F}(\theta, \phi)$. To put it another way, $P$ is the pushforward of the uniform measure on the torus by $\tilde{F} \circ F^{-1}$.

**Continuity of feature maps** We sample $n = 10^4$ points according to $P$, and compute the Čech PDs of the corresponding set $\mathsf{A}_n$. For $i = 1$, we plot the persistence images with weights $\mathrm{pers}^p$, $p = 0, 1, 2, 4$, with both birth and persistence values ranging between 0 and 1, and grid step equal to 0.002. See Figure 4.

**Convergence of the total persistences.** We run three experiments. First, we compute the Čech PD for $i = 0$ of $n$ uniform points on a (nongeneric) circle. Second, we compute the Čech PD for $i = 0$ of $n$ uniform points on a (nongeneric) torus. Third, we compute the Čech PD for $i = 1$ of $n$ uniform points on a (nongeneric) torus. Each experiment is ran for values of $n$ ranging from $n = 10^2$ to $n = 10^4$. We observe in Figure 5 the convergence rate of the total persistence of $\mathrm{dgm}_i^{(1)}(\mathsf{A}_n)$ predicted by Theorem 4.1. We then repeat each experiment 10 times, with similar rates of convergence observed for each run, although with a large relative standard deviation (around $30\%$) for very small values of $\mathrm{Pers}_p(\mathrm{dgm}_i^{(1)}(\mathsf{A}_n))$ (the relative standard deviation being defined by the ratio between the standard deviation and the mean).

**Convergence of $\mu_{n,i}$.** Let $i = 1$. Let $Q$ be the probability distribution obtained as the pushforward of the uniform distribution on $[0, 2\pi]^2$ by the map $F$ defined above. Using Equation (49), one can show that the density $f$ of $Q$ is given by

$$\forall x \in \mathsf{M}_0, \ f(x) = \frac{1}{(2\pi)^2} \frac{1}{r(R + r\cos(\theta))}, \tag{51}$$

where $(\theta, \phi) = F^{-1}(x)$. Assume that the Radon measure $\mu_{\infty,i,2}$ has a density $g_0$ on $\Omega$ (this fact is not proven, but conjectured and strongly supported by experiments). Then, using (9), one can make a change of variables and observe that $\mu_{f,i}$ has a density $g_f$, given by

$$\forall v \in \Omega, \ g_f(v) = \frac{1}{(2\pi)^3} \int_0^{2\pi} \frac{1}{r(R + r\cos(\theta))} g_0 \left( \frac{1}{2\pi} \frac{v}{\sqrt{r(R + r\cos(\theta))}} \right) \mathrm{d}\theta. \tag{52}$$

We sample a set $\mathsf{A}_n$ of $n = 10^5$ points on a square and consider the convolution of the measure $\frac{1}{n} \sum_{u \in \mathrm{dgm}_i(\mathsf{A}_n)} \delta_{n^{1/m}u}$ with a Gaussian kernel, so that we obtain an estimation $\hat{g}_0$ of the density $g_0$. We then define an estimator $\hat{g}_f$ of $g_f$ by approximating the integral over a regular grid of 100 points on $[0, 2\pi]$.

We then place a grid of size $100 \times 100$ over the square $[0, 17]^2$. The measure $\mu_{f,i}$ is approximated by the measure $\hat{\mu}_{f,i}$ with piecewise constant density on each square of the grid, with density in the square centered at $x$ given by $\hat{g}_f(x)$. The corresponding measure is displayed in the center of Figure 6.

For a given value of $n$, we sample a set $A_n$ of $n$ points according to $Q$, and compute the corresponding measure $\mu_{n,i}$. The same transformations are applied to $\mu_{n,i}$, so that we obtain a piecewise constant measure $\hat{\mu}_{n,i}$ on the same $100 \times 100$ grid. The heatmap of this measure is shown on the left of Figure 6.

At last, we approximate the distance $\mathrm{OT}_2(\mu_{n,i}, \mu_{f,i})$ by computing the distance $\mathrm{OT}_2(\hat{\mu}_{n,i}, \hat{\mu}_{f,i})$ using the POT library, see [40, 55]. The distance is then normalized by $\mathrm{OT}_2(0, \hat{\mu}_{f,i}) = \sqrt{\mathrm{Pers}_2(\hat{\mu}_{f,i})}$. The evolution of the normalized distance with respect to $n$ is plotted in Figure 6. As predicted by Theorem 4.1, the $\mathrm{OT}_2$ distance converges to 0 as $n$ gets larger. We then repeat each experiment 10 times, with a convergence to 0 observed each time, the relative standard deviation being always smaller than 7%.

