# OpenReview forum: "Wasserstein convergence of Cech persistence diagrams for samplings of submanifolds"
_NeurIPS.cc/2024/Conference — NeurIPS 2024 poster_

### Official Review · Reviewer_hrBD · 2024-06-24

**Soundness:** 4
**Presentation:** 4
**Contribution:** 3
**Rating:** 9
**Confidence:** 4

**Summary:**

The paper provides a more fine-grained analysis of stability results for persistent homology for data sampled from manifolds, proving new theoretical guarantees for methods in topological data analysis.

**Strengths:**

The results of the paper are in this reviewers opinion strong and interesting. Topological methods, in particular persistent homology, has become a widely used and powerful tool for the analysis of data, in particular data zhat is sampled from an underlying manifold. In many cases, the use of persistent homology can add an additional layer of explainability. For this, mathematical guarantees are fundamental. The new guarantees proven in this paper are significantly stronger than previous ones. In this reviewers opinion the proofs are correct and the precise arguments as well as the underlying ideas are nicely presented.

**Weaknesses:**

As the authors note, the proofs require some assumptions, which are not always guaranteed: they assume the manifold hypothesis and that the points are sampled without noise. Also, their analysis is based on Cech persistence and not Vietoris-Rips persistence, which is used more often in practice. In this reviewers opinion it is however still reasonable to make these assumptions, as they provide the necessary structure to thoroughly mathematically analyze the used tool (in this case persistent homology) in an „ideal world setting“, as is often done for mathematical guarantees.

**Questions:**

-Did you perform any experiments on data with noise or with Vietoris-Rips persistence? If so, how did the results compare to the ones you present in the paper?

**Limitations:**

The limitations (that are also mentioned above) are addressed and explained at multiple points in the paper:
-in line 169 the authors argue why some assumptions are always needed.
-in line 216 they explain why the genericity assumption is needed for their Theorem 3.3.
- in line 353 they again mention the assumptions as the main limitation of their work.

---

> ### Author Rebuttal · Authors · 2024-08-06
>
> Dear reviewer,
>
> Thank you very much for your interest in our work.
> We address the issue of noisy data in our response to all reviewers, and we agree that similar guarantees for the Vietoris-Rips complex would be valuable; in fact, we are already working towards proving such results.
>
>
> Regarding experiments in particular, preliminary investigations suggest that comparable results should hold for the Vietoris-Rips persistence, and also seem to support what we explain regarding noisy data in our general response.

---

> > ### Comment · Reviewer_hrBD · 2024-08-08
> >
> > I would like to thank the authors for their convincing rebuttal.

---

### Official Review · Reviewer_xmoJ · 2024-07-05

**Soundness:** 4
**Presentation:** 4
**Contribution:** 4
**Rating:** 4
**Confidence:** 3

**Summary:**

The paper describes three new theorems on the stability of persistence diagrams with respect to Bottleneck and Wasserstein distance under certain additional assumptions.

**Strengths:**

All three results are interesting fundamental contributions to the field of topological data analysis

**Weaknesses:**

I am not sure about how much the results are in scope for a machine learning conference. I do agree that a solid foundational layer is important but my feeling is that the connection could be described in more length (Sec 4.4 seems to serve this purpose partially).

The authors also admit themselves in the conclusion that the results rely on the manifold hypothesis and the absence of noise. I guess one could argue in length about the manifold hypothesis and how realistic it is in practice. But my hunch is that not allowing any noise is a more serious limitation.

**Questions:**

Why do you think that the paper is an adequate fit for Neurips, as opposed to, say, a journal specialized on TDA?

**Limitations:**

As mentioned, they address the limitations of their results adequately

---

> ### Author Rebuttal · Authors · 2024-08-06
>
> Dear reviewer,
>
> Thank you for your appreciation of the overall quality of our work.\
> We discuss the soundness and importance of our noiseless data hypothesis, as well as the relevance of our work for the ML community, in our response to all reviewers.\
> Regarding specifically the question *Why NeurIPS rather than a journal specialized in TDA?*, our understanding is that any high-quality contribution to a subfield of Machine Learning (e.g. TDA, bandits, causality, fairness, etc.) is welcome at NeurIPS, though such a contribution could also be submitted to a more specialized journal or conference.

---

> > ### Comment · Reviewer_xmoJ · 2024-08-13
> >
> > Thanks for the answer. I guess it is up to the area chair, or somebody higher up to decide whether a theory TDA paper is suitable for the conference. My low score reflects my expectations that a Neurips paper should have a more direct connection to machine learning, not because I find the paper weak. In that sense, I am satisfied with the author's response that clarifies that the paper's contribution is on the theory side.

---

### Official Review · Reviewer_rgww · 2024-07-11

**Soundness:** 4
**Presentation:** 4
**Contribution:** 3
**Rating:** 8
**Confidence:** 4

**Summary:**

The $p$-optimal transport convergence of the Cech persistence diagram of a sample of a closed embedded manifold is studied, both in a deterministic and a probabilistic setting. The paper has three main results:
1. An improvement of the classical Cech bottleneck stability result for sufficiently good samples of closed embedded manifolds.
2. A bound for the total $p$-persistence, and a $p$-optimal transport-convergence result, for the Cech persistence diagram of sufficiently good (deterministic) samples of generically embedded closed manifolds; in the case $p$ is strictly larger than the dimension of the manifold.
3. A probabilistic convergence result for the Cech persistence diagram, and a law of large numbers for its total persistence. The probabilistic convergence result says in particular that $p$-optimal transport convergence of the Cech persistence diagram occurs if and only if $p$ is strictly larger than the manifold dimension.

**Strengths:**

- The paper is very well written: it is clear and easy to follow, it contains many useful and relevant references, and the results are clear and well abstracted.
- The results are (to the best of my understanding) novel, as well as relevant for both theoretical and applied purposes. The results are connected to a lot of previous literature, so they should be of great interest to researchers in stochastic topology as well as the Topological Data Analysis public in general.
- The probabilistic result identifies precisely the choices of $p$ that lead to convergence.

**Weaknesses:**

- The intersection of Topological Data Analysis and Machine Learning is non-trivial but not huge, restricting the audience of the paper to some extent.
- The results have several restrictive hypothesis: strict manifold hypothesis (data lies exactly on a manifold), genericity of the manifold, no noise of any kind.

**Questions:**

1. Is there any hope that (at least part of) the main probabilistic result (Corollary 4.3) still holds for non-generic manifolds? Could the genericity assumption be removed if we instead assume that the data is sampled from a manifold plus noise?
2. Theorem 3.3 fails for non-generic manifolds. Is it possible to quantify (say, with a number or function) how generic a manifold is, in such a way that the constants of Theorem 3.3 only depend on the level of genericity of the manifold?
3. In your summary of Corollary 4.3, in the introduction, why don't you say that optimal transport convergence occurs if and only if $p > m$?

**Limitations:**

The manifold being generic also seems like a limitation to me. It seems genericity could fail in highly structured scenarios (data sampled from a configuration space or a simulation of a dynamical system). Please correct me if I am wrong.

---

> ### Author Rebuttal · Authors · 2024-08-06
>
> Dear reviewer,
>
> Thank you for your appreciation of our work and your interesting questions.
> We discuss noisy data in general in our response to all reviewers.
> To answer your questions more specifically:
>
> *Is there any hope that (at least part of) the main probabilistic result
> (Corollary 4.3) still holds for non-generic manifolds?* :
>
> No, one can build strong counter-examples (e.g. with a well-chosen union of circles in a high-dimensional space). In fact, such configurations will be studied at length in our next work on persistent homology dimension.
>
> *Could the genericity assumption be removed if we instead assume that the data is sampled from a manifold plus noise?*
>
> This is a very interesting question, whose answer seems to depend on the kind of noise considered. See also our discussion of noisy data in our answer to all reviewers.
>
> *Theorem 3.3 fails for non-generic manifolds. Is it possible to quantify (say, with a number or function) how generic a manifold is, in such a way that the constants of Theorem 3.3 only depend on the level of genericity of the manifold?*
>
> It might be doable, but it would require a lot of additional (and somewhat tedious) work. Indeed, the proofs of [ACSD 23] rely on nonquantitative compactness arguments; quantitative versions of these arguments would need to be developed to achieve such a result.
>
> *In your summary of Corollary 4.3, in the introduction, why don't you say that optimal transport convergence occurs if and only if $p>m$?*
>
> Convergence also occurs if  $i\geq m$ and $p=m$. Furthermore, we do not know under which precise conditions it occurs when $p<m$ and $i\geq m$.
>
>
> *The manifold being generic also seems like a limitation to me. It seems genericity could fail in highly structured scenarios (data sampled from a configuration space or a simulation of a dynamical system). Please correct me if I am wrong.*
>
> You are right to point out that we cannot expect *all* submanifolds to be generic.
> However, one can easily build counter-examples to most of our results if the genericity assumption is dropped - as such, the need for genericity is not really a limitation of our work in itself, but rather an unavoidable constraint.
> Note also that not all highly structured scenarios are hopeless: in particular, it has been shown that among real algebraic submanifolds (a highly non-generic and structured subset of all submanifolds), a generic subset satisfies conditions similar to ours (see e.g. https://arxiv.org/abs/2402.08639).

---

> > ### Comment · Reviewer_rgww · 2024-08-08
> >
> > I am satisfied with the authors' responses.

---

### Official Review · Reviewer_G8s6 · 2024-07-12

**Soundness:** 3
**Presentation:** 3
**Contribution:** 2
**Rating:** 5
**Confidence:** 2

**Summary:**

The authors study the behaviour of persistent homology under subsampling of compact sets. They have provided new convergence guarantees with respect to the p-Wasserstein distance and asymptotic results for their $\alpha$-persistence.

**Strengths:**

(S1)	The paper addresses a relevant problem in TDA, and proves a number of significant results

(S2)	The paper is comprehensive and well-written

**Weaknesses:**

(W1) While the main results of this paper are theoretically solid and of strong interest to the TDA community, I do not see direct applications of this work to AI/ML. The authors have referenced a few works where their work might be applied (persistent homology dimension), but they have not elaborated enough on how exactly their method can be used, what kind of significant questions it can answer, and how successful that will be.

(W2) The experiments are very limited and do not demonstrate the implications of the proved results in ML

**Questions:**

(Q1) Are there any results about real-world applications of the theory?

**Limitations:**

Yes, all limitations are addressed adequately by the authors.

---

> ### Author Rebuttal · Authors · 2024-08-06
>
> Dear reviewer,
>
> Thank you for your feedback. We address your concerns and your question in our response to all reviewers.

---

> > ### Comment · Reviewer_G8s6 · 2024-08-13
> >
> > I thank the authors for their response. I have read the response and would like to stick with my score.

---

### Comment · Area_Chair_b6Ns · 2024-07-31
**Comments by the area chair**

Question. Do all experiments considered only uniformly sampled points on a square and a 2-dimensional torus?

Limitations. The major limitation of persistence is its weakness as an isometry invariant. For all standard filtrations (including Cech complexes considered in this paper) on a point cloud, any persistence diagram is invariant under all distance-preserving transformations and hence should be compared with the already known isometry invariants of point clouds.

Already in 2004, Boutin and Kemper proved that the distribution of pairwise distances distinguishes all generic clouds of unordered points in any Euclidean space, see  https://www.sciencedirect.com/science/article/pii/S0196885803001015.

Combined with the computability in quadratic time and Lipschitz continuity, this result makes distance distributions much more practical than persistence. These distributions were extended to the complete invariants for all point clouds (including singular ones) by Widdowson et al in the paper at https://openaccess.thecvf.com/content/CVPR2023/html/Widdowson_Recognizing_Rigid_Patterns_of_Unlabeled_Point_Clouds_by_Complete_and_CVPR_2023_paper.html

More recently, persistence was proved to be identical (and even empty) for generic non-isometric point clouds by Smith et al in the paper at https://link.springer.com/article/10.1007/s41468-024-00177-6. All stability results for persistence provided only upper (not lower) bounds for distances on persistence diagrams, but these distances can often be zeros for generic clouds of unordered points.

Though the Cech complex is too slow for real data, the proofs in the paper (if correct) can still be valuable in the hope that they can be later extended to more practical constructions. However, it is important to embrace a bigger picture as advocated in the book "The Scientific Truth, the Whole Truth and Nothing but the Truth", https://www.taylorfrancis.com/books/mono/10.1201/9781003405399/scientific-truth-whole-truth-nothing-truth-john-helliwell

If the authors are committed to the highest level of scientific integrity, the paper should discuss the *whole* truth about persistence including its weakness as an isometry invariant vs easier, faster, and stronger invariants.

The authors and reviewers are encouraged to consult the guidance at https://neurips.cc/Conferences/2024/ReviewerGuidelines and focus on justified comments, questions, and answers, without putting weight on numerical scores because scientific decisions should be made not by popular votes but through rigorous arguments.

---

> ### Author Response · Authors · 2024-08-06
> **Response to the area chair**
>
> We thank the AC for taking the time to write a detailed comment. To summarize, the AC is not specifically criticizing our work, but rather the use of persistent homology (PH) as a method in data analysis in general. \
>     Their criticisms focus on two arguments: (1) PH is a weak isometry invariant, in particular when compared to the distribution of pairwise distances and (2) when used with the Cech filtration, it is too slow to compute PH for time-critical applications. Before addressing these two concerns, let us remark that it is true that we did not compare PH to other methods in our work, nor discussed about  computational concerns of PH-based methods. This is due to a very simple reason: this is not the topic of our paper, which is focused on providing an extensive theoretical study of a basic method in Topological Data Analysis (computing the Cech persistence diagram of a set of points) under the manifold hypothesis. Other papers that compare PH-based methods to other methods in machine learning already exist, and have even been published at NeurIPS, see e.g. [1]. They show that PH achieve state-of-the-art performance on a variety of tasks, which explains its use by an active community of researchers, see e.g. the various applied papers we cite in the introduction. Let us now respond to the two concerns:
>
> * _PH is a weak isometry invariant._ This is true! We thoroughly agree with the AC: PH should not be used for isometry detection. The problem of testing whether two sets of points are (approximately) isometric is a delicate question topic, which has been addressed from several different perspectives. Beyond the papers cited by the AC, one could  also consider other approaches, such as the one proposed by Brécheteau, who builds isometry tests with prescribed asymptotic levels, see [2].
>
>     In applications, PH is typically not used for isometry detection, but as a complementary tool to other methods. As explained by Turkes \& al. in [1], PH is useful to detect geometric or topological quantities such as the number of holes, the presence of curvature, or convexity, and has the advantage of producing easily interpretable features. This argument is the general argument used in works on persistence and not something specific to this (theoretical) paper, which is why we did not repeat it.
>
>     _To summarize, building complete isometric invariants is an interesting research topic, but it is not related to PH, and it is not the subject of our work._
>
>
> * _Cech persistence diagrams are ``too slow for real data''._ Known upper bounds for the time complexity for computing  \v Cech persistence diagrams for a set of $n$ points in dimension $d$ are of order $O(n^{3d/2})$. Note that this is only a rough upper bound in a worst case scenario. In practice, computing the  Cech persistence diagram of a set of $n=10^4$ points for $d=3$ takes only a couple of seconds on a standard laptop. Still, as the AC rightfully indicates, _the computation of  Cech persistence diagrams notoriously suffers from the curse of dimensionality_, and Cech persistence diagrams are used for low-dimensional problems in practice (say $d\leq 4$). In higher dimensional settings, a variant of Cech persistence diagrams, called Rips persistence diagrams are used. Rips persistence diagrams are only based on the distance matrix of the data points, and can therefore be computed in high-dimensional settings. Moreover, several research teams have designed fast algorithms to compute them, see [3,4,5]. To give an order of magnitude, the Rips PD of a point cloud of size $10^4$ in dimension $100$ can be computed in roughly $10$ seconds on a 2017 Macbook Pro. Rips persistence diagrams are generally believed to behave in a qualitatively similar way to  Cech persistence diagrams in most situations. Still, we choose to focus on  Cech persistence diagrams in this article, their connections to critical points of the distance function to a set making them tailored to our study. We ran additional numerical experiments with Rips PDs instead of  Cech PDs, with similar behaviors observed. Proving the corresponding asymptotics for Rips PDs constitutes an exciting open problem.
>
> Regarding the AC's specific question _Do all experiments considered only uniformly sampled points on a square and a 2-dimensional torus?_: We consider non-uniform samplings on ``deformed’’ two-dimensional tori such as the one displayed in Figure 3.
>
> [1] https://proceedings.neurips.cc/paper_files/paper/2022/file/e637029c42aa593850eeebf46616444d-Paper-Conference.pdf
>
>  [2] https://inria.hal.science/hal-01426331v3/document
>
> [3] GUDHI
>
>  [4] Ripser
>
>  [5] Ripser++

---

> > ### Comment · Area_Chair_b6Ns · 2024-08-07
> > **Thank you for the detailed reply**
> >
> > The authors are right that the previous comment didn't criticize the theorems proved in the submission but mentioned the limitations, which were missed in other reviews, but should be included according to https://neurips.cc/Conferences/2024/ReviewerGuidelines : "You are encouraged to think through whether any critical points are missing and provide these as feedback for the authors."
> >
> > >it is true that we did not compare PH to other methods in our work, nor discussed about computational concerns of PH-based methods.
> >
> > This honest statement could have been used by many NeurIPS reviewers as an easy reason for rejection.
> >
> > >Other papers that compare PH-based methods to other methods in machine learning already exist, and have even been published at NeurIPS, see e.g. [1]. They show that PH achieve state-of-the-art performance on a variety of tasks
> >
> > The quoted variety of tasks in [1] involves prediction of simple isometry invariants, so persistence as an isometry invariant is expected to outperform all non-invariant methods:
> >
> > (1) counting 0, 1, 2, 4 or 9 holes for samples of 20 shapes, without any guarantees, which were proved in "A fast and robust algorithm to count topologically persistent holes in noisy clouds" (CVPR 2014);
> >
> > (2) prediction of a *constant* curvature by combining persistence with SVM without guarantees;
> >
> > (3) convexity detection for samples of 8 shapes based on Theorem 1 saying that "X is convex if and only if for every line in R^d the persistence diagram in degree 0 with respect to the tubular filtration contains exactly one interval", whose proof wasn't found online but seems to immediately follow from the definition saying that a convex body, for any two of its points, should contain the straight-line segment connecting them. More importantly, Theorem 1 requires infinitely many checks for all possible lines.
> >
> > The limitations of [1] honestly accept that "we do not have an extensive comparison of the state-of-the-art for the given problems", so the quotes can be more exact.
> >
> > Reference [2] includes stronger and more relevant results, which deserve detailed explanations. Already the abstract talks about "a pseudo-metric between metric-measure spaces, that is bounded above by the Gromov-Wasserstein distance". Such a pseudo-metric with upper bounds can be theoretically zero for many non-equivalent objects but Propositions 3.7-3.9 prove some lower bounds under certain hypothesis, though still without claiming a proper metric on isomorphism classes.
> >
> > >research teams have designed fast algorithms to compute them, see [3,4,5]. To give an order of magnitude, the Rips PD of a point cloud of size 10^4 in dimension 100 can be computed in roughly 10 seconds on a 2017 Macbook Pro.
> >
> > All pairwise distances between 10^4 points in any Euclidean dimension might be computed (at least without storing them) even faster and are much more valuable due to their generic completeness under isometry. This fast computational time hints that the outputs can be small and often even empty as formally proved now for generic clouds with trivial 1D persistence.
> >
> > The more important problem is to provide theoretical guarantees. Otherwise, any quick output can be claimed the fastest "state-of-the-art". Hence your guarantees for the Cech filtration are valuable before computation becomes relevant. The only advice is to accept the known limitations and carefully quote the relevant past results.

---

> > > ### Author Response · Authors · 2024-08-08
> > >
> > > Dear AC,
> > >
> > > We share your commitment to scientific truth. Would you find it satisfactory if we added something  like the following to the Related Works section
> > >
> > > *"Additionally, it should also be noted that other invariants, such as the empirical distribution of pairwise distances, have been used to study similar questions, for instance in [...]",*
> > >
> > > as well as a subsection in the Appendix (where we describe Cech persistence diagrams) that mentions and compares alternative methods?

---

> > > > ### Comment · Area_Chair_b6Ns · 2024-08-08
> > > > **references to past work**
> > > >
> > > > Dear authors, because of the page limit, it's enough to briefly include the discussed references about isometry invariants within a review of past work or limitations in the main paper, so feel free to include more details in appendices. If you would like to save space, the words "Additionally, it should also be noted that other invariants, such as the empirical distribution" may not be essential, because the mentioned distributions of distances are explicitly defined invariants, not empirical outputs. Thank you.

---

### Author Rebuttal · Authors · 2024-08-06

We thank all reviewers for their time, effort, and valuable feedback. We are grateful for the overall positive reception of our work.
The most common questions pertained 1) to the practical applications of our results, and 2) to our assumption that the data is noiseless.
We address these two points below, and respond to each reviewer individually regarding their more specific questions.

* **Applications and relevance to the ML community:** (particularly in response to reviewers G8s6 and xmoJ)

    Though our paper does provide some new heuristics regarding the optimal choice of parameter for the Wasserstein distance (see Section 5), its main goal is to offer strong theoretical guarantees and a better understanding of techniques and objects routinely used by the TDA community, which is part of the wider ML community.
    As such, it is a study of existing ML methods rather than the description of a new method, similar e.g. to http://proceedings.mlr.press/v139/carriere21a.html (for an example within the field of TDA).
    Hence the main real world applications of our results are to be found in more experimental papers that apply the techniques that we examine, and for which we provide new guarantees.
    Of course, we also hope that a deeper understanding of those methods will in turn result in new methods, but those are beyond the scope of this (already rather long) article.

    This explains the relatively small number of experiments as well (which was commented upon by reviewer G8s6); those are not meant to showcase the power of some new method, nor to prove that our results are correct (as the mathematical proof is enough), but rather to illustrate them and to show that our asymptotic results are already observable with a reasonably small number of points.



* **Noise:** (particularly in response to reviewers rgww, xmoJ and hrBD): \
    Being limited by space constraints, we chose to focus on the noiseless case, but we agree that statements that allow for some noise would be a welcome addition to our results. \
Depending on the nature of the noise considered, some of our results extend seamlessly to the noisy case, while others would require additional work:
for example, if we let the maximum amplitude of the noise be small compared to the density $\epsilon = d_H(A,M)$ of the sampling $A$ in the manifold (before the addition of the noise), i.e. of order $\epsilon^2$, then Theorem 2.2 still holds (with modified constants) thanks to the Bottleneck stability theorem, and Region 3 of the diagram still has a finite number of points in expectation, as in Proposition 4.2. \
If we assume that the noise is normal to the submanifold and uniform (or at least that its density is lower and upper bounded) and that its amplitude $l$ is fixed and independent from the point cloud, then the situation is equivalent to sampling from an upper and lower bounded distribution on the open set $M^l$ (the offset of the submanifold by the amplitude of the noise). Its boundary is smooth or at least very regular, depending on the amplitude of the noise.
Regarding the small cycles (i.e. the points in Region 1 of the diagram), everything should work roughly as in the case of the $d$-dimensional cube, i.e. the limit distribution of $\mu_{n,i}$ should be some integral of $\mu_{\infty,i,d}$ over the open set. The behavior of the large cycles, i.e. of the points in Regions 2 and 3, should be dictated by the shape of the boundary of $M^l$. In particular, $\partial M^l$ should be generic enough for a generic choice of $M$ and $l$ that slightly modified of most of our results still hold, in particular the finite expected number of points in Region and the Wasserstein convergence of the diagrams  (except that $p$ needs to be greater than the ambient dimension $d$, rather than the intrinsic dimension $m$).
These questions might be explored in detail in future work.

---

### Author Response · Authors · 2024-08-13

Thanks again to all reviewers and to the AC for their feedback.

---

### Decision · Program_Chairs · 2024-09-25

**Decision:**

Accept (poster)

**Comment:**

The paper proved upper bounds for convergence of persistent homology when data points are sampled from a submanifold in a Euclidean space. All reviewers agreed that theoretical results are worth publishing. The authors acknowledged the limitations of persistence as an isometry invariant for generic point clouds in comparison with easier, faster and stronger isometry invariants, and also in the cases when persistence can be empty and hence the proved upper bounds certainly hold.